# Mapping the world's inland surface waters: an upgrade to the Global Lakes and Wetlands Database (GLWD v2)

Bernhard Lehner[1], Mira Anand[1], Etienne Fluet-Chouinard[2], Florence Tan[1,3], Filipe Aires[4], George H. Allen[5], Philippe Bousquet[6], Josep G. Canadell[7], Nick Davidson[8,9], C. Max Finlayson[9,10], Thomas Gumbricht[11], Lammert Hilarides[12], Gustaf Hugelius[13], Robert B. Jackson[14], Maartje C. Korver[1], Liangyun Liu[15], Peter B. McIntyre[16], Szabolcs Nagy[12], David Olefeldt[17], Tamlin M. Pavelsky[18], Jean-Francois Pekel[19], Benjamin Poulter[20], Catherine Prigent[4], Jida Wang[21,22], Thomas A. Worthington[23], Dai Yamazaki[24], Xiao Zhang[15], Michele Thieme[25]

[1] Department of Geography, McGill University, Montreal, Quebec H3A 0B9, Canada
[2] Earth System Science Division, Pacific Northwest National Laboratory, Richland, WA, USA
[3] Department of Earth and Environmental Sciences, KU Leuven, Leuven, Belgium
[4] Observatoire de Paris, LERMA, CNRS, France
[5] Department of Geosciences, Virginia Polytechnic Institute and State University, Blacksburg, VA, USA
[6] Laboratoire des Sciences du Climat et de l'Environnement, Université Paris-Saclay, CEA, CNRS, UVSQ, Gif sur Yvette, France
[7] Global Carbon Project, CSIRO Environment, Canberra, ACT 2601, Australia
[8] Nick Davidson Environmental, Queens House, Ford Street, Wigmore HR6 9UN, UK
[9] Gulbali Institute for Agriculture, Water & Environment, Charles Sturt University, Albury, NSW, Australia
[10] Centre for Ecosystem Science, School of Biological, Earth and Environmental Sciences, UNSW Sydney, Sydney, NSW, Australia
[11] Department of Physical Geography, Stockholm University, 10691 Stockholm, Sweden
[12] Wetlands International, 6700 AL Wageningen, The Netherlands
[13] Department of Physical Geography and Bolin Centre for Climate Research, Stockholm University, SE 106 91 Stockholm, Sweden
[14] Department of Earth System Science, Woods Institute for the Environment, and Precourt Institute for Energy, Stanford University, Stanford, CA
[15] Aerospace Information Research Institute, Chinese Academy of Sciences, Beijing 100094, China
[16] Department of Natural Resources and the Environment, Cornell University, Ithaca, NY 14853
[17] Department of Renewable Resources, University of Alberta, Alberta T6G 2G7, Canada
[18] Department of Earth, Marine and Environmental Sciences, University of North Carolina, Chapel Hill, NC 27599, USA
[19] European Commission, Joint Research Centre (JRC) Ispra, Italy
[20] NASA Goddard Space Flight Center, Biospheric Sciences Lab., Greenbelt, MD 20771, USA
[21] Department of Geography and Geographic Information Science, University of Illinois Urbana-Champaign, Illinois 61801, USA
[22] Department of Geography and Geospatial Sciences, Kansas State University, Kansas 66506, USA
[23] Conservation Science Group, Department of Zoology, University of Cambridge, Cambridge, UK
[24] Institute of Industrial Science, The University of Tokyo, Tokyo, 153-8505, Japan
[25] World Wildlife Fund, Washington, DC, 20037, USA

*Correspondence to*: Bernhard Lehner (bernhard.lehner@mcgill.ca)

**Abstract.** In recognition of the importance of inland waters, numerous datasets mapping their extents, types, or changes have been created using sources ranging from historical wetland maps to real-time satellite remote sensing. However, differences

in definitions and methods have led to spatial and typological inconsistencies among individual data sources, confounding their complementary use and integration. The Global Lakes and Wetlands Database (GLWD), published in 2004, with its globally seamless depiction of 12 major vegetated and non-vegetated wetland classes at 1 km grid cell resolution, has emerged over the last decades as a foundational reference map that has advanced research and conservation planning addressing freshwater biodiversity, ecosystem services, greenhouse gas emissions, land surface processes, hydrology, and human health. Here, we present a new iteration of this map, termed GLWD version 2, generated by harmonizing the latest ground- and satellite-based data products into one single database. Following the same design principle as its predecessor, GLWD v2 aims to avoid double-counting of overlapping surface water features while differentiating between natural and non-natural lakes, rivers of multiple sizes, and several other wetland types. The classification of GLWD v2 incorporates information on seasonality (i.e., permanent vs. intermittent vs. ephemeral); inundation vs. saturation (i.e., flooding vs. waterlogged soils); vegetation cover (e.g., forested swamps vs. non-forested marshes); salinity (e.g., salt pans); natural vs. non-natural origins (e.g., rice paddies); and a stratification of landscape position and water source (e.g., riverine, lacustrine, palustrine, coastal/marine). GLWD v2 represents 33 wetland classes and—including all intermittent classes—depicts a maximum of 18.2 million km$^2$ of wetlands (13.4% of the global land area excluding Antarctica). The spatial extent of each class is provided as the fractional coverage within each grid cell at a resolution of 15 arc-seconds (approximately 500 m at the equator), with cell fractions derived from input data at resolutions as small as 10 m. The updated GLWD v2 offers an improved representation of inland surface water extents and their classification for contemporary conditions (~1984-2020). Despite being a static map, it includes classes that denote intrinsic temporal dynamics. GLWD v2 is designed to facilitate large-scale hydrological, ecological, biogeochemical, and conservation applications, aiming to support the study and protection of wetland ecosystems around the world.

## 1 Introduction

Wetland ecosystems ranging from lakes and rivers to marshes, swamps, peatlands, mangroves, and numerous other wetland types are critically important for humans and Earth system processes. As key components of global hydrological and biogeochemical cycles and as habitats for biodiversity, they provide some of the most valuable ecosystem services to human society (Costanza 1998; Millennium Ecosystem Assessment 2005). Wetlands directly and indirectly influence many environmental and socio-economic systems through their carbon storage (e.g., Chmura et al., 2003; Duarte et al., 2005; Raymond et al., 2013; Hugelius et al., 2020); nutrient processing (e.g., Cheng et al., 2020); provision of water, food and other resources (e.g., Mitsch et al., 2015); biological productivity (e.g., Gibbs, 2000; Mitchell, 2013); flood and drought mitigation (e.g., Tallaksen & Van Lanen, 2004; Čížková et al., 2013; Junk et al., 2013); coastal protection (e.g., Gedan et al., 2011; Marois & Mitsch, 2015); and water quality regulation (e.g., Verhoeven & Setter, 2010).

Accurate and comprehensive maps of wetland ecosystems are fundamental to quantifying their role within the water, carbon, and nutrient cycles, to plan conservation and restoration actions, to guide effective resources management, and to assess and

mediate human interactions and pressures (van Asselen et al., 2013; Qiu et al., 2021). Beyond knowledge of their areal extent, characteristics such as vegetation, hydrology, salinity, and connectivity are critical for distinguishing the roles and behaviors of different wetland types. As a critical input to hydrologic and Earth system models, global lake and wetland distributions are of particular interest for current and future water resources assessments, carbon and nutrient budget calculations, climate change projections, and other large-scale land surface studies (e.g., Bullock & Acreman, 2003; Lauerwald et al., 2023). Consistent information across large scales is required to set a global baseline to contextualize long-term degradation of wetland ecosystems (Vörösmarty et al., 2010; Darrah et al., 2019; Murray et al., 2019) and forecasted risks from environmental change (e.g., Xi et al., 2021), as well as to offer interim data to countries currently lacking (or having outdated) national inventories (Davidson et al., 2018). While freshwater biodiversity is among the most threatened in the world (Ramsar Convention on Wetlands, 2021), several regions or countries have nearly eradicated their wetland cover since pre-industrial times (Fluet-Chouinard et al., 2023). Reliable maps are therefore needed for monitoring the progress towards global conservation targets, such as to track changes in the extent of water-related ecosystems over time as mandated by the United Nations Sustainable Development Goal 6.6 (*"Protect and restore water-related ecosystems, including mountains, forests, wetlands, rivers, aquifers and lakes"*).

Global maps of inland (non-marine) wetland ecosystems have improved continuously over the last four decades (Figure 1). Literature estimates of global wetland extents range broadly from 5 to 13 million $km^2$, with lower and upper boundaries of 2 and 17 million $km^2$ (Lieth, 1975; Matthews & Fung, 1987; Aselmann & Crutzen, 1989; Dugan, 1993; Finlayson & Davidson, 1999; Spiers, 1999, 2001; Lehner & Döll, 2004; Prigent et al., 2007; Tiner, 2009; Fluet-Chouinard et al., 2015; Mitsch & Gosselink, 2015; Zhang et al., 2023). The wide range is explained by differences in data sources, methodologies, and definitions. Early wetland estimates inherited gaps and inconsistencies from the compilation of national or regional inventories, limiting the reliability of their global perspective (Nivet & Frazier, 2004; Davidson et al., 2018). Over time, compilations of paper maps were replaced by satellite remote sensing imagery and its interpretation using machine learning and artificial intelligence which allowed for seamless mapping across the world at shorter time intervals (Gallant, 2015). These improvements in methods coincided with an increase of the global area of wetland ecosystems mapped over time (Davidson et al., 2018). Nonetheless, wetlands remain the land cover class with the least agreement when comparing across global data products, in which wetlands are often being misclassified as forest, shrub, cropland, or grassland (Nakaegawa, 2012). Even advanced remote sensing methodologies and sensors face challenges in detecting different wetland types or delineating the hydrologically active extent of wetlands, for example when cloud or vegetation cover obstructs the view or when saturated soils are confused with surface inundation (Gallant, 2015). Besides restrictions in spatial and/or temporal resolution, remote sensing approaches are also constrained by their limited historical extent as the first missions launched only in the 1970s.

Differences in definitions of what constitutes an aquatic ecosystem or wetland are the primary factor impeding comparisons across estimates and data sources. The Ramsar Convention on Wetlands (1971) adopted a broad definition of wetlands, comprising nearly all types of aquatic ecosystems as *"areas of marsh, fen, peatland or water, whether natural or artificial, permanent or temporary, with water that is static or flowing, fresh, brackish, or salt, including areas of marine water the depth*

*of which at low tide does not exceed six meters."* However, this definition is not universally accepted (Gerbeaux et al., 2018) and wetland criteria designed for field use are not practical for broad-scale mapping as shown by the wide range of areal estimates across studies (Mahdavi et al., 2018). Individual global map products typically provide their own, narrower definitions, justified by methodological limitations. For instance, inundation maps from passive microwave sensors may omit non-inundated peatlands and may require post-processing to exclude coastal and/or offshore ecosystems to avoid issues of signal oversaturation (Aires et al., 2017; Prigent et al., 2020). Similarly, a specific wetland definition may be required for different applications. For example, ecosystem conservation planning may exclude artificial wetlands such as rice paddies (Reis et al., 2017); or estimates of methane emissions from wetlands may separate open waterbodies from vegetated wetlands to partition the emission budget (Saunois et al., 2020; Zhang et al., 2021) or to remove coastal regions because salinity inhibits methane production (Melton et al., 2013; Poffenbarger, 2011).

Some wetland ecosystem types and extents are better captured by current remote sensing capabilities than others. The distinction of open waterbodies from other wetland ecosystems has become easier with the advent of global river, lake, and other permanent water coverages derived from optical remote sensing (Pekel et al., 2016; Allen & Pavelsky, 2018; Pickens et al., 2020). However, seasonal fluctuations in inundation caused by changes in vegetation and/or saturated soils are not as reliably mapped and contribute disproportionately to the large uncertainties in global wetland estimates (Gallant, 2015). For instance, decadal-long observations estimate that the annual minimum and maximum global inundated areas vary by a factor of 2.8 (Prigent et al., 2007; Fluet-Chouinard et al., 2015). In contrast, some static wetland maps may represent average or maximum conditions, concealing major seasonal or interannual variation in inundation patterns (Prigent & Papa, 2015). Depending on the observation period and the definitions and methods used, different estimates of wetland ecosystems may prove to be complementary, overlap partially, or disagree entirely, thereby further complicating attempts to achieve a comprehensive view across all wetland ecosystem types (Rajib et al, 2024; Junk 2024).

To address the issue of spatial inconsistency, Lehner & Döll (2004) produced the Global Lakes and Wetlands Database (GLWD, hereon GLWD v1) by compiling and harmonizing existing wetland datasets into a single, coherent global database that distinguishes 12 types of waterbodies and wetlands. As one of the most comprehensive global wetland datasets (Nakaegawa, 2012; Mitsch & Gosselink, 2015), GLWD v1 facilitated the integration of wetlands into a broad range of large-scale land surface studies, and it remains one of the most widely used global wetland map to date (Lindersson et al., 2020). However, GLWD v1 has several limitations and drawbacks, including its coarse spatial resolution, outdated sources, the omission of small lakes and rivers, inaccuracies due to projection or generalization issues, and ambiguous definitions of wetland classes (Lehner & Döll, 2004). Since the publication of GLWD v1, newer maps of specific waterbody and wetland types have surpassed single classes of GLWD v1 in their accuracy and spatial or temporal resolution thanks to improved sensors and algorithms, longer archives, and refined training data (Figure 1). Despite these advances on individual waterbody and wetland types, GLWD v1 has not yet been replaced by a harmonized representation of the full range of inland wetland ecosystems. Consequently, the limitations of GLWD v1 described above still constrain scientific and management applications that require detailed knowledge of the global distribution of waterbodies and wetland types.

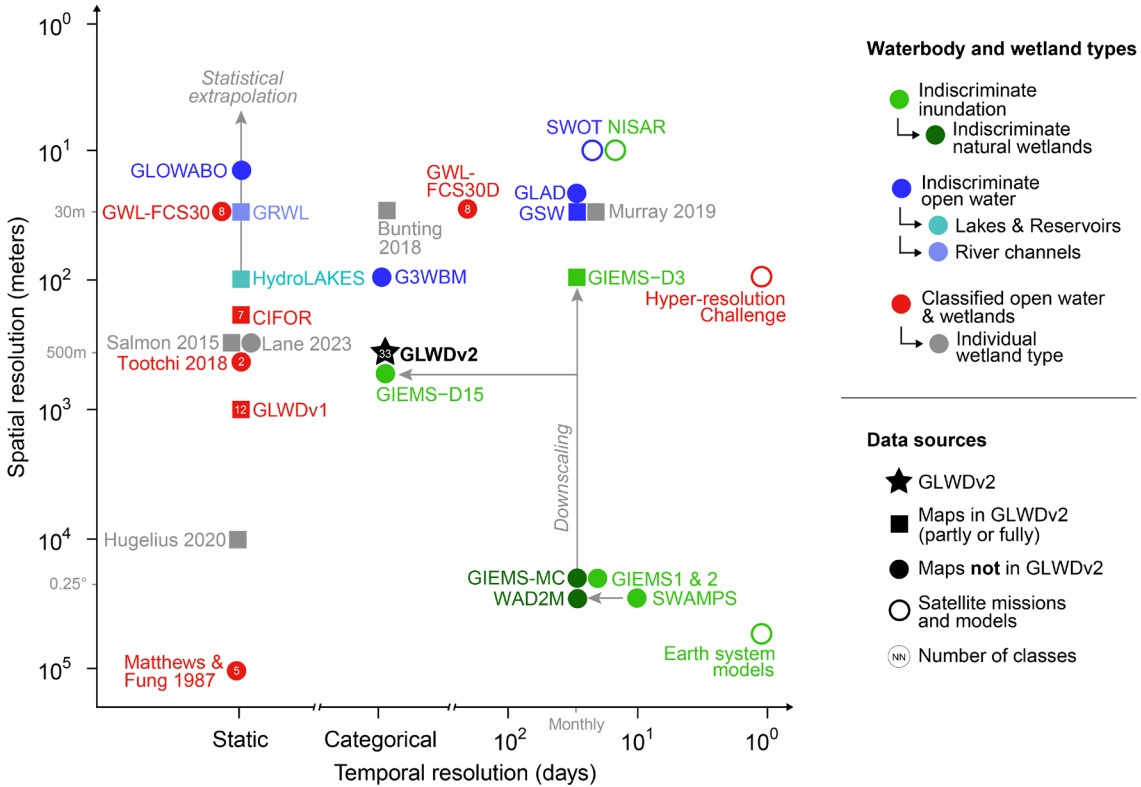

**Figure 1: Common surface water datasets plotted according to their spatial and temporal resolution. Only maps with global or near-global extent, covering >80° of latitudinal swaths, are included. The colors of points represent the typological level of each dataset and together illustrate that classified maps including multiple wetlands and waterbody types have largely remained at a coarser resolution than available data products of indiscriminate wetland types. Arrows in the plot represent which datasets have been used in the production of others. Square points represent data products that were included in the creation of GLWD v2 as presented in this paper (additional detail on these sources can be found in Table 1). The spatial resolution is in meters at the equator. Data products listed as 'Static' do not contain information on inundation frequency, while data products depicting hydrological regimes with qualitative measures are labelled 'Categorical'. Explanations of dataset abbreviations and brief descriptions of each dataset's main characteristics are provided in Table A1 (Appendix A). References for data sources are as follows: G3WBM: Yamazaki et al., 2015; GIEMS-1: Prigent et al., 2007; GIEMS-2: Prigent et al., 2020; GIEMS-D3: Aires et al., 2017; GIEMS-D15: Fluet-Chouinard et al., 2015; GIEMS-MC: Bernard et al., 2024; GLAD: Pickens et al., 2020; GLOWABO: Verpoorter et al., 2014; GLWD v1: Lehner & Döll, 2004; GRWL: Allen & Pavelsky, 2018; GSW: Pekel et al., 2016; GWL_FCS30: Zhang et al., 2023; GWL_FCS30D: Zhang et al., 2024; HydroLAKES: Messager et al., 2016; SWAMPS: Jensen & McDonald, 2019; WAD2M: Zhang et al., 2021.**

Here, we introduce the Global Lakes and Wetlands Database version 2 (GLWD v2) which follows the same design principles as GLWD v1 and is intended to succeed it. GLWD v2 draws upon the best available free data sources to provide a comprehensive and seamless global map of inland surface waters distinguished into 33 non-overlapping waterbody and wetland types. To avoid double-counting across multiple sources and classes, we harmonized input sources at their finest resolution (see Methods) and aggregated the results to a common grid at 15 arc-second resolution (approximately 500 m at the equator). Beyond higher quality inputs and a higher spatial resolution, GLWD v2 features a key structural improvement over its predecessor in that it provides fractional cell coverage of wetland extents per class, rather than a single majority class per cell. This creates two important advantages, namely that 1) multiple classes can share the same grid cell (while the sum of all classes is constrained to not exceed full cell coverage); and 2) individual class layers can preserve wetland extents from original sources at sub-cell resolution without information loss, i.e., the cell's fractional wetland coverage can be calculated from fine-scale maps at resolutions as small as 10 m, where available. The classification of GLWD v2 follows a multi-factor hierarchical system, such that most classes can be grouped with others according to multiple criteria, including landscape position (inland vs. coastal/marine), water source (lacustrine vs. riverine vs. palustrine), vegetation (forested vs. non-forested), and soil type (mineral vs. organic). Furthermore, all 33 individual class maps were combined into one additional majority map to identify the dominant waterbody or wetland type in each grid cell, akin to the original map of GLWD v1. While GLWD v2 represents maximum extents of wetland ecosystems as a static map over the broad contemporary period of 1984-2020, it also provides a simple depiction of intrinsic hydrological dynamics and variability through its classification (permanent, regular, seasonal, and ephemeral). With these numerous improvements, GLWD v2 offers a detailed baseline map of inland surface waters in preparation for time-resolved monitoring of the world's wetland ecosystems in the future.

## 2 Definitions and data sources

### 2.1 Wetland versus waterbody definitions

The working definition of wetlands and waterbodies as applied in GLWD v2 arises from the objective of being all-inclusive, and from practical considerations stemming from the fact that GLWD v2 inherits, at least in part, given definitions from its source datasets by association. As a result, the overarching wetland definition of GLWD v2 does not follow pre-established criteria but is nested within the broader perspective of the Ramsar Convention on Wetlands (1971; see Introduction) in that it includes all inland surfaces that are flooded or saturated longer than a certain period. However, a few Ramsar wetland types are excluded from GLWD v2: subtidal and offshore marine wetlands (e.g., coral reefs, kelp forests) because they lie outside the continental land surface; subterranean, karst and cave environments; as well as subglacial lakes (in part as Antarctica was excluded from the mapping efforts, see Methods).

To simplify the terminology, we here refer to the entire surface water extent covered by GLWD v2 as 'wetland', and in the context of GLWD v2 we consider 'wetlands', 'aquatic environments', and 'inland/terrestrial surface waters' as equivalent. We

use the term 'waterbody' to designate all standing or flowing open water surfaces of any size, typically detectable by optical remote sensing, regardless of whether the water is fresh, brackish, or saline, or whether the waterbody is of natural or human-made origin (e.g., reservoirs). Most but not all waterbodies have permanent open water, while some are intermittent. We then refer to 'other wetlands' as all types of emergent and bare wetlands beyond waterbodies, whether inundated or saturated, permanent or seasonal, fresh, brackish, or saline, vegetated or non-vegetated, natural or human-made (e.g., rice paddies). We acknowledge that the name Global Lakes and Wetlands Database is not entirely consistent with this working definition, but we chose to retain it for historical continuity.

Waterbodies in GLWD v2 are divided into 7 types aligning closely with Ramsar classes, although ignoring lake size as a criterion (see Figure 2). Other wetlands are separated into 26 classes from a combination of biotic, geomorphic and hydrologic factors similar to the Ramsar system, specifically adding elements of temporal inundation dynamics and connectivity as well as soil and vegetation characteristics. Moreover, all 26 other wetland ecosystem types in GLWD v2 can be grouped into five higher-level categories following the Cowardin system (Cowardin et al., 1979) on which Ramsar's is based: lacustrine (lake-associated; lentic); riverine (river-associated; lotic); estuarine (river-associated; tidal); palustrine (depressional; isolated); and coastal (marine; tidal).

## 2.2 Data sources, characteristics, and resolution

GLWD v2 was produced by fusing 25 primarily global datasets (Table 1) ranging from broad representations of wetland ecosystems (e.g., indiscriminate inundated surfaces) to individual types (e.g., mangroves) and ancillary information (e.g., forest cover). The selection of these input datasets was made to a) avoid duplication of information by choosing the single most complete dataset per type based on criteria described below (e.g., only one lake dataset), and b) include only data with unrestricted use permissions so that GLWD v2 can be released with a free and open data license. Dataset characteristics and minimum requirements included globally consistent coverage, spatially uniform quality, sufficiently high spatial resolution (grid cell sizes mostly between 30 m and 500 m, or equivalent for vector layers), and proper documentation. The selection of some datasets was done for coherency with other inputs, for instance a shared shoreline delineation for freshwater lakes, saline lakes, and reservoirs. Data sources representing narrower types of waterbodies or wetlands were preferred over more general sources in order for GLWD v2 to depict wetland types in as much detail as possible.

Our approach of selecting the single best data source when multiple candidates exist for the same feature type suffers from the disadvantage of inheriting all of the source's inaccuracies and uncertainties while precluding the potential benefits of correcting systematic deficiencies by compositing multiple datasets (e.g., filling gaps from cloud, snow or vegetation cover, or improving limited detection of small objects). However, we opted not to combine multiple datasets of the same feature type because of the inherent risks of duplication, distortion, and bias arising from the merger, in particular for inputs capturing different time periods (e.g., shifting river meanders). Some exceptions were made to augment incomplete information in cases where regional datasets were combined (see Methods).

Applying data fusion procedures at high spatial resolution allows to identify coinciding water features which reduces the risk of double-counting in areas of overlap, yet the accuracy of each source dataset also determines the efficacy of the merger. The

initial grid cell resolution of all processing steps for waterbody datasets (and certain wetland types, such as mangroves) was 1 arc-second (~30 m at the equator), reflecting the original resolution of most input datasets. Some preprocessing steps, such as reprojection and resampling, were conducted at even higher resolutions (3 m to 10 m) to minimize loss of information (see Methods). Other wetland types were processed at their respective native resolutions ranging from the highest resolution of ~10 m for saltmarshes to the coarsest dataset of ~1 km for saline/brackish wetlands; the latter requiring disaggregation. All input

datasets were ultimately converted to the GLWD v2 target resolution of 15 arc-seconds (~500 m) and were expressed as fractional cell coverage to retain maximum information. Throughout all processing steps it was ensured that combined waterbody and wetland extents of all classes cannot exceed 100% in a single output grid cell.

**Table 1: Source datasets used for the creation of GLWD v2.**

| Feature | Dataset/Source | Contents/Description/Accuracy/Uncertainty | Time Period | In GLWD v2 |
|---|---|---|---|---|
| Lakes | HydroLAKES *Messager et al., 2016* | Vector polygons of 1.4 million lakes, regulated lakes, and reservoirs. Saline lakes determined using methods by Ding et al. (2024). Only includes lakes ≥10 ha. Dataset is manually verified and corrected. | ~1980-2010 | Freshwater lakes and saline lakes |
| Reservoirs | Global Dam Watch (GDW) database *Lehner et al., 2024b* | Vector polygons of 35,295 reservoir outlines created by merging existing global datasets and applying semi-automated and manual curation protocols. Errors from different inputs were individually verified and corrected. | ~2020 | Reservoirs |
| Rivers | Global River Width from Landsat (GRWL) database *Allen & Pavelsky, 2018* | 30 m resolution raster map derived from Landsat data; supervised detection and classification of large rivers and estuarine rivers of widths >90 m. River and stream surface is 44±15% greater than that found by Raymond et al. (2013), and 15±12% greater than the maximum from Downing et al. (2012). | ~1980-2015 | Large rivers and estuarine rivers |
| Rivers | SWOT River Database (SWORD) *Altenau et al., 2021* | Vector product of center lines of large rivers for use by the Surface Water and Ocean Topography (SWOT) satellite mission. Spatially aligned with the GRWL database. | ~1984-2015 | Augmentation of large rivers |
| Rivers | RiverATLAS as part of HydroATLAS database *Linke et al., 2019* | Vectorized line network of all global rivers that have a catchment area of at least 10 km² or an average river flow of at least 0.1 m³ s⁻¹; extracted from the gridded HydroSHEDS layers at 500 m resolution. | ~1971-2000 | Small streams |
| Open water | Global Surface Water (GSW) dataset by European Commission's Joint Research Center *Pekel et al., 2016* | 30 m (0.9 arc-second) resolution raster product from Landsat providing maps of global surface water from 1984 to present; water presence/absence (including maximum extent, recurrence). Detects visible open water but contains omissions, e.g., due to cloud or vegetation cover. Omission accuracy of 98.8-99.1% for permanent water and 73.8-77.4% for seasonal water. | 1984-2021 | Permanent, seasonal, ephemeral open water |
| Mangroves | Global Mangrove Watch 3.0 *Bunting et al., 2022* | 0.8 arc-second resolution mangrove classification from SAR and Landsat data; baseline classification expanded into a time-series of mangrove change using SAR data from 1996-2020. Estimated overall accuracy of 87.4% (86.2–88.6%). | 1996-2020 | Mangroves |
| Saltmarshes | Global tidal marshes 2020 dataset (version 2.6) *Worthington et al., 2024* | Spatial distribution of tidal marshes between 60°N to 60°S at 10 m grid cell resolution, derived using a random forest classification model applied to earth observation data. Overall accuracy of 85% with a Kappa coefficient of 0.09-0.78, omission errors of 0-29%, and commission errors of 16-94% in different realms for the final tidal marsh map. | 2020 | Saltmarshes |
| Saltmarshes | Global Distribution of Saltmarshes (by UNEP) *Mcowen et al., 2017* | Polygons of saltmarshes across 99 countries, synthesized from a range of national and local datasets. Available points were not included in GLWD v2. Polygon sources lack data in some regions, e.g., Canada and northern Russia. | 1977-2013 | Saltmarshes |
| Intertidal areas | Tidal wetland probability *Murray et al., 2022* | 30 m raster product predicting probability of tidal wetlands based on Landsat data and machine learning. Excludes areas north of 60˚N. Overall accuracy of 86.1% (84.2–86.8%). | 1984-2019 | Other coastal wetlands |

| Feature | Dataset/Source | Contents/Description/Accuracy/Uncertainty | Time Period | In GLWD v2 |
|---|---|---|---|---|
| Peatlands | PEATMAP *Xu et al., 2018* | Global polygon product from meta-analysis which synthesizes national and regional peatland maps. Inputs of varying resolutions/quality; some artefacts at the transitions between regions; differing definitions between inputs. | ~1990-2010 | Composite peatlands |
| Peatlands | SoilGrids250m *Hengl et al., 2017* | 250 m resolution raster product of soil properties at multiple depths based on machine learning. Peatland extents can be approximated by the histel (TAXOUSDA) and histosol (HISTPR) soil layers. Validation of soil organic carbon shows RMSE of 32.8 g/kg and $R^2$ of 0.64. | 2010 | Composite peatlands |
| Peatlands | Northern peatland extents *Hugelius & Olefeldt, unpublished* | 500 m grid showing percent probability of peatlands in northern regions (above 23°N) created by merging of soil grids (versions of 2013), northern & mid-latitude soil databases, and others. Methods described in Olefeldt et al. (2021). | ~1997-2013 | Composite peatlands |
| Wetlands | Global Wetlands Map (CIFOR) *Gumbricht et al., 2017* | 250 m raster product of wetland classes and peatland depth in the tropics and subtropics (south of 40°N) based on an expert system. Wetland area comparable to GLWD v1. Peatlands showed reasonable agreement to ground validation points. | 2011 | Composite peatlands |
| Wetlands | GLWD v1 *Lehner & Döll, 2004* | 1 km global raster map with 12 wetland classes produced from regional data, including class 'Salt pan, saline/brackish wetland'. Source data of varying quality and accuracy, representing both historic and contemporary conditions. | ~1970-2000 with some older data | Salt pan, saline/ brackish wetlands |
| Inundated areas | GIEMS-D3 (Global Inundation Extent from Multi-Satellites, Downscaled 3 arc-seconds) *Aires et al., 2017* | Downscaled 90 m (3 arc-second) raster map of inundation frequency (version 2 as of 2022) derived from multi-sensor satellite data; includes saturated soils and areas with vegetation. Some gaps around large waterbodies. Overall accuracy of 89-93% against SAR Amazon inundation map at high and low water. | 1993-2007 | Indiscriminate inundation surface |
| Floodplains | CaMa-Flood model results *Yamazaki et al., 2011* | CaMa-Flood model simulates floodplain inundation dynamics and flood probability using a river-routing model with floodplain topography at 90 m (3 arc-second) resolution. No information on interfluvial (palustrine) floodplains. | 2001-2014 | Flooding classes and frequencies |
| Rice paddies | GRIPC (Global Rain-fed, Irrigated, and Paddy Croplands) *Salmon et al., 2015* | 500 m global raster product developed from remote sensing imagery, climate data, and national and sub-national agricultural inventory data. Contains classes of rain-fed, irrigated, and paddy cropland. Over USA, overall accuracy of 96% (producer 59%, user 44%). More uncertain over humid areas. | 2005 | Rice paddies |
| Rice paddies | RiceAtlas *Laborte et al., 2017* | RiceAtlas (including RiceCalendar v1) shows the seasonal distribution of the world's rice-producing areas and countries in polygon units. | 2009-2012 | Correction of rice paddies |
| Deltas | Deltas at Risk *Tessler et al. 2015* | Compilation of 48 large river deltas around the world as coarse, generalized polygons. Partly delimited from soil maps, topography, and channel position. | 1974-2003 | Delta classification |
| Forests | Global Forest Change map *Hansen et al., 2013* | 30 m (0.9 arc-second) resolution map of forest extent (percent forest cover) and change from 2000-2022 derived from Landsat imagery. Global accuracy of 99.6% and above 99% for each latitudinal band. | 2000-2022 | Separation of forest vs. non-forest |
| Climate | World Climate Regions *Sayre et al., 2020* | Raster map of 18 climate regions at 250 m resolution, derived by combining global temperature and global moisture datasets. Susceptible to threshold settings and quality of source data. | 1970-2000 | Climate separation of peatland classes |
| Discharge | RiverATLAS *Linke et al., 2019* | RiverATLAS includes a global grid at 500 m resolution of downscaled long-term (1971-2000) average discharge estimates (Müller Schmied et al., 2021). Validation against 3003 gauges showed $R^2$ of 0.99, 0.2% positive bias, and sMAPE of 35%. | 1971-2000 | Source of riverine classes |
| Glaciers | GLIMS (Global Land Ice Measurements from Space) *Raup et al., 2007* | Global polygon map of glacier extents, ranging from 1850 to present. The collection includes data from approximately 70 percent of the world's 200,000 glaciers. | ~1999-2021 with some older data | Masking of glaciers |
| Urban areas | WSF (World Settlement Footprint) 2019 *Marconcini et al., 2020* | 10 m resolution binary map showing the presence of human settlements derived from Sentinel-1 and Sentinel-2 data. Overall accuracy of 83.3-89.0% across different comparison criteria. | 2019 | Masking of urban areas |


# 3 Methods

## 3.1 Overview of methodology

The guiding principle for creating GLWD v2 was to consolidate and harmonize—without duplication—all input data sources to produce a versatile global map of wetland types that is useable in a broad spectrum of applications. Antarctica was excluded

from the mapping efforts due to generally incomplete or unreliable spatial input data. Results are provided as a series of grids with a target cell size of 15 arc-seconds (~500 m) which was chosen as a compromise between the spatial resolution of existing input data sources, computing demands, and ease of use for global applications. It is important to note, however, that the information from finer resolution input data, including permanent water surfaces at ~30 m and saltmarshes at ~10 m resolution, is preserved in the fractional cell coverage of each wetland type. The classification scheme of GLWD v2 (Figure 2) is designed

to be manageable (i.e., limited to a reasonable number of classes), expert guided rather than statistically derived, and representative of the needs of various research fields and disciplines. Each of the 33 wetland classes is provided as an individual global map depicting the extent of the respective class as cell fractions. The 33 maps are then combined to derive the total global wetland extent and to identify the dominant wetland class per grid cell.

The main processing steps of GLWD v2 are outlined in Figure 3 and are described in more detail in sections 3.2 to 3.5. The

central procedure combines four types of data: a) high-resolution data of waterbodies; b) data of various resolutions of other wetland types; c) high-resolution downscaled or modeled data of indiscriminate inundated areas; and d) ancillary data to support the classification of indiscriminate wetland types and the refinement of classes.

For the merger, higher quality data sources were assigned priority over lower quality ones based on reliability, precision, resolution, confidence, completeness (in time and space), coherence, and information content (e.g., classified vs. unclassified

data). When these criteria were ambiguous or conflicting (e.g., higher resolution but lower confidence), the prioritization of input datasets was guided by expert decision. The sequential merger of data layers was performed by a process we hereafter refer to as "inserting" wetland extents, whereby the next lower priority layer is successively allowed to occupy the grid cell space that remains free after all higher priority waterbodies and wetlands have been processed (analogous to 'mosaicking' in GIS terminology). Data sources representing waterbodies were first combined following the order: lakes > reservoirs > rivers

> other subclasses. Next, data sources depicting individual wetland types were inserted around the waterbodies, followed by indiscriminate inundated areas that were subsequently classified using ancillary information. Thus, predominantly permanent waterbodies were spatially allocated first, and mostly non-permanent wetland extents were inserted thereafter to complement and surround these waterbodies. Finally, the map was refined by masking urban (built-up) and glaciated areas.

The sequential merger of multiple layers of different original resolutions to one common layer results in a combined grid where

multiple wetland types can overlap in a 15 arc-second output cell. Importantly, the sequence of layer stacking ensures that higher level (finer) features are systematically subtracted from lower level (coarser) ones during the overlay and insertion process. This eliminates—or at least reduces (depending on spatial precision and resolution of data sources)—double-counting in cases of spatial overlap and asserts that the summed waterbody and wetland coverage is bounded by the total area of each output cell.


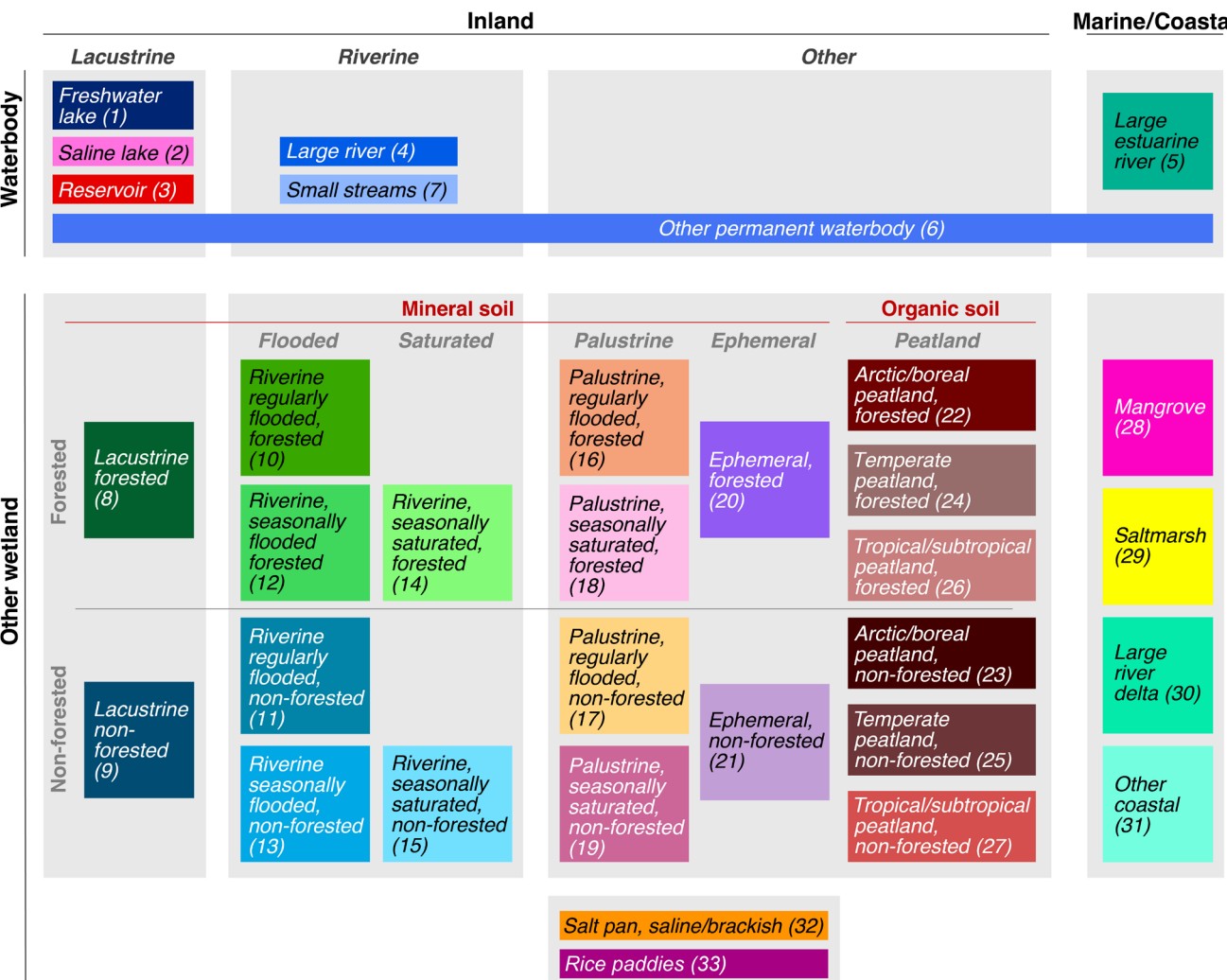

**Figure 2: Schematic of classification hierarchy and distinctions among the 33 classes represented in GLWD v2. At the highest level, classes are grouped into 4 realms resulting from the 2×2 combinations of the overarching wetland division (Waterbody vs. Other wetland [including emergent and bare wetlands]) with landscape position (Inland vs. Marine/Coastal). Inland waterbodies and other wetlands are then further divided according to water source and dynamic (Lacustrine [lentic], Riverine [lotic], and Other [including Palustrine and Peatland]). Other characteristics, such as soil type (Mineral vs. Organic) and vegetation cover (Forested vs. Non-forested) can be used to regroup wetland classes across water sources. Finally, mineral wetlands are further separated by their hydrological conditions (Flooded vs. Saturated) and regimes (Ephemeral).**

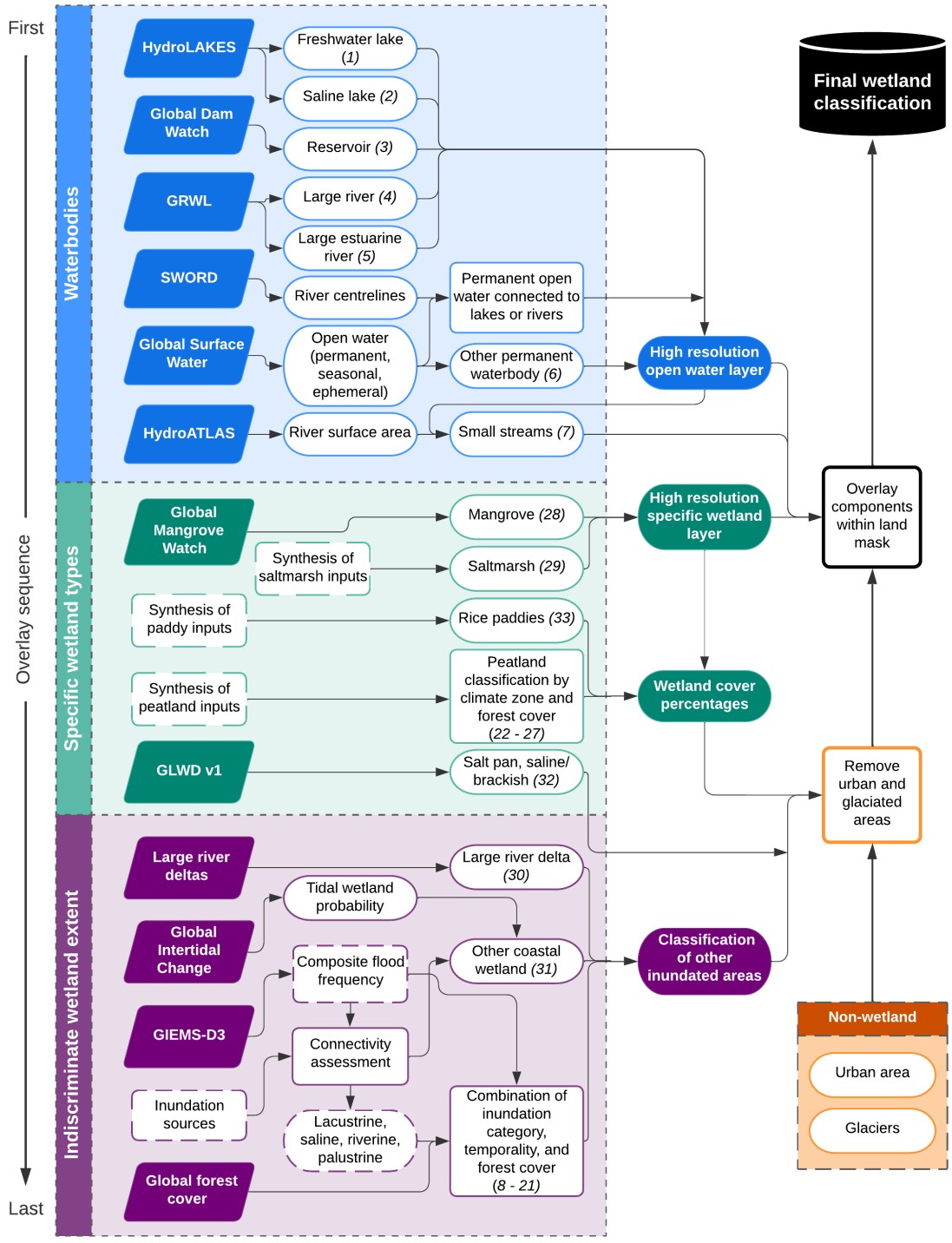

**Figure 3: Schematic of the workflow and main processing steps to create GLWD v2. The processing steps are grouped into three main parts, corresponding to sections 3.2 (blue), 3.3 (green), and 3.4 (purple) of the text. Input datasets are**

**represented by parallelograms, while processes are represented by boxes. Interim and final layers and classes are represented by ovals, with class numbers indicated in parentheses after each class name. Input datasets and results of each section are shown in solid shading. Shapes with dashed outlines represent complex processes with inputs that are not specifically described in this diagram; the details of these steps and their source data are explained in the text. This schematic broadly indicates the sequential order in which the different datasets were combined (from top to bottom =**

**first to last); however, some wetland types were reclassified or grouped together to produce the final set of 33 classes, as described in sections 3.2-3.4. A more detailed version of this schematic with additional sub-steps is provided in Figure B1 (Appendix B).**

## 3.2 Processing of waterbodies

Figure 3 illustrates the main processing steps of this section in blue color, and all data sources are listed in Table 1. The input datasets of waterbodies were processed globally at 1 arc-second (~30 m) resolution, except for small streams which were processed at 15 arc-second (~500 m) resolution. Some preprocessing steps were executed at higher resolutions (see details below).

### 3.2.1 Lakes, saline lakes, and reservoirs (classes 1-3)

Lakes were extracted from the polygons of the HydroLAKES database (Messager et al., 2016), which contains ~1.4 million lakes globally with a size of at least 10 ha. We converted the lakes of HydroLAKES v1.1 (including regulated lakes but excluding reservoirs) to a raster layer at 1 arc-second resolution according to whether at least half of each grid cell area was covered by a lake polygon. Reservoirs were extracted from the Global Dam Watch (GDW) database v1.0 (Lehner et al., 2024b), which contains 35,295 reservoir polygons globally, applying the same polygon to raster conversion as for lakes. It should be

noted that HydroLAKES and the GDW database are spatially complementary and thus do not include any overlapping polygons.

Furthermore, we distinguished saline lakes (assuming a relatively high salinity threshold of 30 ppt, i.e., 30 g L$^{-1}$) using a classification framework based on hydrography datasets, satellite imagery, and literature documentation as described in Ding et al. (2024). The supervised classification identified a total of 24,374 saline lakes, mostly located in endorheic (closed) inland

depressions and arid or semi-arid climate zones. These conditions are conducive to salinity accumulation due to lack of surface outflow, strong potential evaporation, or both. Many of the detected saline lakes exhibit lacustrine evaporites visible from satellite images. To evaluate the overall robustness of the classification method, we conducted an independent literature search for all lakes exceeding 500 km$^2$ in surface area which confirmed that all 66 reported saline lakes (with salinity levels exceeding 3 g L$^{-1}$) in that size class were correctly detected in the supervised classification, and only one saline lake from the supervised

classification required conversion to non-saline.

### 3.2.2 Rivers and estuarine rivers (classes 4-5)

Rivers and estuarine rivers were extracted from the raster layers of the Global River Width from Landsat (GRWL) database (Allen & Pavelsky, 2018). It should be noted that the original GRWL data also offer a 'lake' class, but we used this class only as a component layer to identify and conserve critical connections between lakes and their in- or out-flowing river courses.

After Step 3.2.3 below, the 'lake' class from GRWL was discarded to avoid double-counting of lakes.

We reprojected and resampled all published 10×10-degree GRWL tiles from their original 30 m resolution and UTM projection to match the geographic coordinate system of GLWD v2 at a 0.25 arc-second (~8 m) resolution. The resulting high-resolution tiles were then aggregated and merged to create a seamless global layer that retained all 1 arc-second cells with at least 50% river coverage. The GRWL tiles can exhibit minor gaps at their edges when combined into a seamless global coverage, which

we rectified by inserting data inside a 0.1-degree buffer around all edges from an unpublished version of GRWL (provided by the authors) of slightly inferior quality but with overlapping tiles.

To further ensure connectivity between river surfaces and adjacent lakes in the subsequent combination steps, we also processed and inserted the related vector product of the Surface Water and Ocean Topography (SWOT) River Database (SWORD) (Altenau et al., 2021) which presents center lines for all GRWL rivers including their paths traversing through

lakes. We converted these vector lines to a grid at 1 arc-second resolution, added a one-cell buffer to produce slightly wider river lines, and retained only those SWORD cells that coincided with a GRWL 'lake'. Furthermore, as the SWORD river center lines can cross land, such as over islands within a braided river system, we removed all SWORD river cells farther than one cell from the permanent open water class of the Global Surface Water (GSW) dataset (see next step).

### 3.2.3 Other permanent waterbodies (class 6)

We used the Global Surface Water (GSW) dataset (Pekel et al., 2016) to complement the lakes, reservoirs, and rivers. GSW offers gridded data at 0.9 arc-second resolution (~27 m) compiled from Landsat imagery spanning the years 1984 to present. We used the separation of GSW into permanent, seasonal, and ephemeral classes from its 'transitions' layer for the years 1984 to 2020 and resampled it to our target 1 arc-second resolution. Cells labeled as seasonal or ephemeral and coinciding with a GRWL 'lake' were reclassified as permanent to conserve the lake-river connections in subsequent steps. We then inserted

permanent GSW cells as their own waterbody class in GLWD v2, which nominally includes—but does not distinguish between—small lakes, ponds, rivers, and canals that exceed the 27 m detection threshold of GSW and have not been depicted in any of the other waterbody datasets. The seasonal and ephemeral classes were integrated into the 'other wetland' classification in subsequent steps (section 3.4).

### 3.2.4 Combination of waterbody classes and reclassification of some cells

We combined the open waterbody features at the 1 arc-second resolution described in the previous steps by overlaying them in the following priority order: freshwater lake > saline lake > reservoir > river > estuarine river > other permanent waterbody.

In instances where waterbody boundaries were misaligned in the source datasets (e.g., a lake from HydroLAKES may not cover the entire permanent water from GSW; or a gap exists between the outlines of hydrologically connected lakes and rivers), we reclassified some of the gap cells of GSW from 'other permanent waterbody' to the type of the adjacent waterbody. This reclassification was performed based on proximity along contiguous cells from the waterbodies up to a maximum distance of 0.002 degrees (~200 m).

### 3.2.5 Adding small streams (class 7)

The surface extent of small rivers and streams is not well captured in global remote sensing imagery due to the narrow, linear features in sub-meter dimensions (Allen et al., 2018). To account for this omission, a statistical estimate of the surface area of small streams was produced using river area estimates from the RiverATLAS database (Linke et al., 2019), which in turn were derived from downscaled discharge estimates (Müller Schmied et al., 2021) and simple hydraulic geometry laws (Allen et al., 1994). The total surface area of rivers and streams was calculated by multiplying the estimated channel width and length of every river reach that exceeds 10 km$^2$ in catchment area or 0.1 m$^3$ s$^{-1}$ (100 liters per second) in average flow in each 15 arc-second grid cell (Linke et al., 2019). To represent only small streams and avoid double-counting with larger rivers already mapped by GRWL (Step 3.2.2), the GRWL river extent was subtracted from the total river area provided by RiverATLAS in each 15 arc-second cell. Given the uncertainty of this estimation method, small streams were given the lowest priority among all waterbodies. Finally, the maximum extent of small streams was limited to 10% of each 15 arc-second cell (~2.5 ha), which resembles a river reach of approximately 500 m length (one cell) and 50 m width, as the GRWL and GSW products should cover rivers exceeding this size, even if not coinciding within a given cell due to potential spatial mismatches. It should be noted that while 'small streams' are grouped within the waterbody classes of GLWD v2, 50-60% of small streams globally have been estimated to be intermittent or ephemeral (Messager et al., 2021).

### 3.3 Processing of explicit wetland types

Figure 3 illustrates the main processing steps of this section in green color, and all data sources are listed in Table 1. Datasets representing the distribution of explicit wetland types were processed globally at 0.3, 1, 3, or 15 arc-second resolution (~10, 30, 90, or 500 m, respectively) depending on their native data format.

### 3.3.1 Insertion of high-resolution coastal wetlands (classes 28-29)

We used original high-resolution source data to define the extent of three explicit coastal wetland types at the target processing resolution of 1 arc-second (~30 m): mangroves (Bunting et al., 2022), saltmarshes (Worthington et al., 2024; Mcowen et al., 2017), and intertidal areas (Murray et al., 2022). The mangrove class was produced from the maximum mangrove extent in the source data after resampling from its original 0.8 arc-second resolution. The saltmarsh class was created by first resampling the original ~10 m resolution tiles of the dataset by Worthington et al. (2024) and converting all provided saltmarsh polygons of the dataset by Mcowen et al. (2017) to the target 1 arc-second resolution. Given the lower accuracy and completeness of the

dataset by Mcowen et al. (2017), it was only used for regions north of 60° N where no data from Worthington et al. (2024) existed. The intertidal wetland areas, which were later integrated into the 'other coastal wetland' class (see Step 3.4.3), were resampled from their original ~30 m resolution and all grid cells with a given probability of inundation of at least 50% were retained. The three classes were then inserted into the map of harmonized waterbody classes (result of section 3.2), giving priority to waterbodies followed by mangroves > saltmarshes > intertidal areas.

### 3.3.2 Masking of urban and glaciated areas

Up until this step, all previous data sources were included into GLWD v2 without further corrections because they met high standards of spatial accuracy and detail. Before adding coarser resolution information, however, high-resolution non-wetland masks for urban areas and glaciated areas were inserted at the 1 arc-second resolution to prevent subsequent steps from allocating wetlands to these surfaces. The urban areas were aggregated from the original 10 m resolution of the World Settlement Footprint 2019 binary mask (Marconcini et al., 2021) to produce a percentage cover at 1 arc-second resolution and cells with at least 50% settlement cover were classified as urban. Glaciated areas (Raup et al., 2007) were converted from their original polygon format to the target 1 arc-second resolution. At the end of all processing steps, i.e., before the creation of the final GLWD v2 maps (section 3.5), the urban and glaciated classes were discarded and replaced by the 'dryland' (non-wetland) class.

### 3.3.3 Insertion of rice paddies (class 33)

Rice paddy extents (Salmon et al., 2015) were inserted as percent coverage into all remaining unoccupied areas. The original grid—which delineates global rice paddy extents at 500 m resolution based on a predictive model—included numerous artifacts such as erroneous small patches over regions with no known rice production. Furthermore, under realistic conditions rice paddies typically form only part of a heterogeneous landscape mosaic, where rice fields intersperse with other agriculture, roads, and small settlements, i.e., at a 500 m resolution each grid cell is covered by less than 100% rice paddies. We therefore converted the rice paddy layer from its original binary format to a fractional 0-100% range using several preprocessing steps. After reprojecting and resampling the original data to the target geographic coordinate system and 15 arc-second cell resolution of GLWD v2, we calculated each 15 arc-second cell's rice fraction as the percentage of the original rice paddy extent found within a distance of ~2 km around the cell (i.e., in a 9×9 cell neighborhood). We then used the administrative areas available as part of the RiceAtlas (Rice Calendar v1; Laborte et al., 2017) to discard regions where no paddy rice production is reported (after some minor manual corrections). Finally, the maximum rice paddy extent within a grid cell was capped at 50% and any rice paddy coverage below 20% was considered an inherent data error (mostly occurring along marine coastlines) and was removed. These thresholds, besides delivering visually plausible rice paddy regions, were chosen in an iterative trial-and-error process to approximately match the reported global rice paddy extent of ~1.2 million km$^2$, as well as the reported extents of the two dominant rice producing countries of India (~400,000 km$^2$) and China (~250,000 km$^2$) (see Table 4 in Results for sources).

### 3.3.4 Insertion of peatlands (classes 22-27)

Several global or near-global peatland extent maps have been developed in the past, each with its own specificities, strengths, and weaknesses, which led us to conclude that no single data product is of sufficient quality and/or completeness to represent all peatlands in GLWD v2. Therefore, we created a new composite peatland probability map from four input datasets (Table 1): PEATMAP (Xu et al., 2018; global), SoilGrids250m (Hengl et al., 2017; global), Northern Peatlands (Hugelius & Olefeldt, unpublished; north of 23° N), and CIFOR (Gumbricht et al., 2017; south of 40° N, of which we only used data south of 23.5° N). The four input datasets were first reprojected and/or resampled into the geographic coordinate system of GLWD v2 and converted to a peatland percentage cover in each 15 arc-second grid cell, as follows:

PEATMAP originally offers spatial peatland percentages for regions in Canada and some areas in eastern Asia (0-100%), and otherwise binary presence/absence information which we set to 100% and 0%, respectively. PEATMAP is provided in polygon format which we corrected for some slight locational misalignments across Oceania and some regions of East Asia. Also, individual polygon parts with an area <20 ha (i.e., smaller than one grid cell in our target 15 arc-second resolution) were removed as upon visual inspection many of them represented spurious outliers and artifacts rather than precise peatland boundaries. SoilGrids250m offers cumulative probabilities (0-100%) of histosols occurring in any 250 m grid cell globally, as well as an independent probability of histels. We used the maximum value of histosols or histels per 15 arc-second cell and interpreted the result as the spatial probability of peatland occurrence in percent. The Northern Peatlands grid is based on the same underpinning data and methods as presented in Olefeldt et al. (2021) and was re-produced here as a 15 arc-second grid specifically for the purpose of inclusion in GLWD v2. It offers percent peatland extent per grid cell for histosols and histels, separately, which we summed into one grid (0-100%). Finally, the CIFOR dataset includes a binary peatland classification which we interpreted as 0 or 100% coverage, respectively. Furthermore, to avoid abrupt spatial transitions in the binary information of CIFOR, we inserted the values from SoilGrids250m wherever CIFOR showed zero values.

After standardization, the four layers of peatland probabilities were combined into an equally-weighted average; i.e., by calculating the average of the respective three input grids that existed north of 23.5° N and south of 23° N; and the average of all four input grids in the 0.5° transition zone using an edge smoothing (blending) approach. Calculating averages ensures that final extent probabilities remain within 0 and 100% and that the total global peatland extent falls within the individual estimates of the input datasets. We removed values below 3% from the final composite peatland map as these low percentages occurred throughout the globe including in areas of no known peatland extent, mostly due to artefacts of low probabilities inherent in the SoilGrids250m product of histosols and histels.

To create three climatological peatland types, we combined the composite peatland map with reclassified climate zones from the World Climate Regions (Sayre et al., 2020) which we first resampled from the original ~250 m to 15 arc-second resolution. We separated peatlands into arctic/boreal (original polar and boreal climates), temperate (original cool and warm temperate climates), and tropical/subtropical (original tropical and subtropical climates); and we applied a manual adjustment in that arctic/boreal climates were reclassified to temperate in regions below 43° N (with some additional adjustments of small non-

contiguous areas between 43° N and 55° N) to avoid the occurrence of minor arctic/boreal peatlands within tropical/subtropical mountains.

Finally, each of the three peatland classes was further subdivided into forested vs. non-forested using the same ancillary forest data and approach as described in more detail in Step 3.4.3 below. The six resulting combinations of climatological and forested/non-forested peatland classes were then inserted into GLWD v2.

### 3.3.5 Insertion of salt pans, saline and brackish wetlands (class 32)

In the absence of better global information, the extent of salt pans and saline/brackish wetlands was taken from the gridded
version of GLWD v1 and disaggregated from its original 30 to 15 arc-second resolution. The salt pans and saline/brackish wetlands were assumed to occupy 100% of the original grid cells. Before insertion into GLWD v2, this class was augmented with the saline class derived in Step 3.4.2 below. An exception to our fusion rules was made in that this class could later be replaced by the two wetland types 'large river delta' and 'other coastal wetland' (see Step 3.4.3) as these two classes were considered more reliable than the coarse GLWD v1 product.

## 3.4 Processing and classification of indiscriminate wetland extents

Figure 3 illustrates the main processing steps of this section in purple color, and all data sources are listed in Table 1. Datasets representing the distribution of indiscriminate wetland extents were processed globally at 3 arc-second (~90 m) resolution. First, an all-encompassing global inundation extent map was created, which was then classified using ancillary data and an analysis of connectivity to the nearest waterbody.

### 3.4.1 Determination of maximum inundation extent and flood frequencies

We created an indiscriminate maximum inundation extent map at 3 arc-second resolution and assigned flood frequency values to each cell by combining four input datasets: a) the downscaled GIEMS-D3 inundation data at 3 arc-second resolution over 1993-2007 (Aires et al., 2017) which formed the majority of the maximum extent as it includes both permanent open water and temporary wetlands; b) the waterbody layer of GLWD v2 produced in Steps 3.2.1 to 3.2.4 at 1 arc-second resolution (i.e.,
without small streams); c) the seasonal and ephemeral open water cells of the GSW datasets at 1 arc-second resolution; and d) the flooded extent simulated by the CaMa-Flood model as inundated for more than 7 days per year at 3 arc-second resolution (Yamazaki et al., 2011). The 1 arc-second input datasets were aggregated to 3 arc-second resolution by defining each 3 arc-second grid cell as inundated if it contained at least one wetland cell at 1 arc-second resolution.

All four inundation data sources were combined by extracting the maximum inundation frequency (0-100%) per grid cell
among the sources. With its broad coverage, the GIEMS-D3 database provided most of the inundation frequency estimates (0-100%), but was supplanted by the following (wherever they occurred and showed higher inundation frequencies): GLWD v2 waterbodies (assumed to have 100% inundation frequency as most of these waterbodies are permanent); seasonal GSW cells (80% inundation frequency, broadly based on GSW statistics); CaMa-Flood inundation (10% inundation frequency, slightly

above the applied minimum inundation threshold of 7 days per year); or ephemeral GSW (5% inundation frequency, GSW statistics).

### 3.4.2 Division of indiscriminate inundation into broad categories using hydrological connectivity

In order to classify the wetlands encompassed by the indiscriminate inundation extent, we first stratified the maximum extent map (Step 3.4.1) into one of five broad water source categories: lacustrine, saline, riverine, coastal, and palustrine—which we then further refined in Step 3.4.3 below. These five categories were derived by determining the nearest hydrologically connected flooding source (waterbody, ocean, or local runoff) for each indiscriminate inundation cell, with hydrologic connectivity and distances being measured along flow paths between contiguous wetland cells. The flooding sources for the five categories originated from the previously assigned GLWD v2 classes, such that: cells nearest to freshwater lakes or reservoirs were classified as lacustrine; cells nearest to saline lakes as saline; cells nearest to rivers as riverine; cells nearest to estuarine rivers or the ocean as coastal; and all other cells disconnected from a source as palustrine.

Several additional criteria were applied in the determination of connectivity and proximity, and all parameters and thresholds were set by expert judgment guided by visual comparisons to known wetland complexes. We used the flood frequency map (Step 3.4.1) as input to trace paths of flooding between every inundated cell and its most likely source of flooding. The most plausible connectivity was determined through a custom algorithm which ensured that the shortest flow paths followed preferential flow directions from each cell towards the neighboring cell with highest flood frequency, while remaining within contiguous inundation cells. This approach permits cells to be assigned a more spatially distant source if the flood frequencies are higher along that path. The process development and thresholds of lacustrine and coastal source attribution (see below) were informed by visual comparisons with the elevation range from variations in lake surface water elevations observed by ICESat-2 (Cooley et al., 2021) and along coastlines by a reanalysis of tides and surges (Muis et al., 2022).

Two iterations of the connectivity assessment were performed. First, connectivity along cells with flood frequencies ≥80% was determined to represent more persistent inundation and direct connectivity of wetlands fringing their adjacent waterbodies. This iteration was assumed to fully define the lacustrine and saline categories and they were removed from the following iteration. Second, unassigned inundated cells were categorized into riverine and coastal with an expanded connectivity assessment over cells of >10% inundation frequency and using previously assigned riverine and coastal cells as additional sources. Also, riverine sources were supplemented in the second iteration by cells with a long-term average discharge exceeding 1 m$^3$ s$^{-1}$ from the RiverATLAS database (Linke et al., 2019). During both iterations, grid cells with an elevation above 10 m a.s.l. were excluded from becoming coastal. All grid cells without an assigned category after both iterations, signifying no surface hydrological connectivity to flooding sources, were labeled as palustrine.

### 3.4.3 Final classification of indiscriminate wetlands with ancillary data (classes 8-21, 30, and 31)

The lacustrine, riverine, and palustrine categories were further subdivided into 14 classes based on inundation frequencies and forest cover (Table 2). Due to the thresholds used in the previous step, the only category containing grid cells with inundation

frequencies below 10% was palustrine. These palustrine wetlands with low-frequency flooding were further constrained to a minimum frequency of 3% to remove the highly uncertain representation of rarely inundated extents in GIEMS-D3 data, and then relabeled as 'ephemeral'.

Forest cover (Hansen et al., 2013) was used to separate between wetlands that fit the general definition of forested swamps vs.
non-forested freshwater marshes. For this process, the percent tree cover values were first resampled by averaging from the original 0.9 to 3 arc-second resolution. To also accommodate shrubbed swamps, we set a relatively low threshold of 10% tree coverage for forested wetlands, which was visually calibrated to match known swamp occurrences including parts of the Pantanal in South America; the Tonle Sap freshwater swamp forests in Asia; and the Sudd, Okavango, Bangweulu, and Niger Delta swamps in Africa.

Large river deltas were discerned as an additional class within the indiscriminate inundation areas using ancillary information. We converted the polygons of large river deltas (Tessler et al., 2015) to a grid at 3 arc-second resolution, and because of their low-precision outlines we extended them with a ~1 km buffer (15 grid cells) to avoid spurious gaps at the land-ocean boundary. Delta areas were clipped to the extent of the maximum inundation map (Step 3.4.1). The large river delta class (#30) superseded all other classes of the indiscriminate inundation areas.

Furthermore, we grouped a small number of conceptually similar classes to simplify and eliminate ambiguities: outside of large river deltas, the coastal wetland category was combined with the intertidal wetlands (Step 3.3.1) to form the 'other coastal wetlands' class (#31); and the saline wetland category was added to the 'salt pan, saline/brackish wetland' class (Step 3.3.5). Finally, all lacustrine, riverine, palustrine, ephemeral, coastal, and saline classes derived for the indiscriminate wetland areas were inserted into the remaining open grid cell spaces of GLWD v2.


**Table 2: Thresholds used to define lacustrine, riverine, palustrine, and ephemeral wetland classes.**

| ID | GLWD v2 class | Category | Inundation frequency | Forest cover |
|----|---------------|----------|----------------------|--------------|
| 8 | Lacustrine, forested | Lacustrine | ≥80% recurrence on GIEMS-D3 | ≥10% |
| 9 | Lacustrine, non-forested | | ≥80% recurrence on GIEMS-D3 | <10% |
| 10 | Riverine, regularly flooded, forested | Riverine | ≥50% recurrence on GIEMS-D3 | ≥10% |
| 11 | Riverine, regularly flooded, non-forested | | ≥50% recurrence on GIEMS-D3 | <10% |
| 12 | Riverine, seasonally flooded, forested | | Flooded on CaMa-Flood | ≥10% |
| 13 | Riverine, seasonally flooded, non-forested | | Flooded on CaMa-Flood | <10% |
| 14 | Riverine, seasonally saturated, forested | | 10-49% recurrence on GIEMS-D3, or seasonal on GSW | ≥10% |
| 15 | Riverine, seasonally saturated, non-forested | | 10-49% recurrence on GIEMS-D3, or seasonal on GSW | <10% |
| 16 | Palustrine, regularly flooded, forested | Palustrine | ≥50% recurrence on GIEMS-D3 | ≥10% |
| 17 | Palustrine, regularly flooded, non-forested | | ≥50% recurrence on GIEMS-D3 | <10% |
| 18 | Palustrine, seasonally saturated, forested | | 10-49% recurrence on GIEMS-D3, or seasonal on GSW | ≥10% |
| 19 | Palustrine, seasonally saturated, non-forested | | 10-49% recurrence on GIEMS-D3, or seasonal on GSW | <10% |
| 20 | Ephemeral, forested | Palustrine | 3-9% recurrence on GIEMS-D3, or ephemeral on GSW | ≥10% |
| 21 | Ephemeral, non-forested | | 3-9% recurrence on GIEMS-D3, or ephemeral on GSW | <10% |

### 3.5 Creation of final GLWD v2 maps

For each of the 33 GLWD v2 wetland classes, an individual global grid was produced at the output 15 arc-second resolution showing the percent coverage of the respective wetland class per grid cell. In addition, the resulting spatial extents of all wetland classes were summed for each cell, creating a total global wetland extent map (the maximum total extent was capped at 100% where rounding caused slight exceedances). These 34 fractional maps were also produced to show absolute areas (in ha) per grid cell—using geodesic calculations—for ease of application. Finally, the dominant wetland class per grid cell (i.e.,

the class showing the highest fractional wetland coverage per cell) was determined to create a single global map of wetland types. In cases of ties, the dominant class was assigned to be the lower class number.

### 4 Results

GLWD v2 distinguishes 7 waterbody types and 26 other wetland types for a total of 33 distinct non-overlapping classes (Table 3). It provides a static snapshot of the inland surface water extent and climatology for contemporary conditions, centered

around the period 1984-2020 which represents the varying time periods of most of its input data (see Table 1). Its nominal spatial resolution is 15 arc-seconds (~500 m), yet it provides cell fractions of wetland cover that are derived from water surfaces at resolutions as fine as 0.3 arc-seconds (~10 m) to preserve smaller waterbodies. This database surpasses its predecessor, GLWD v1 (Lehner & Döll, 2004) in detail, consistency, and comprehensiveness to serve a broad range of applications by offering a composite global map of wetland ecosystem types.

### 4.1 Global wetland extent

The total combined extent of all wetland classes in GLWD v2 including all inland and coastal waterbodies and wetlands of all inundation frequencies—that is, the maximum extent—covers 18.2 million km$^2$, equivalent to 13.4% of the total global land area excluding Antarctica (Table 3). Most wetlands are found in Asia (43.8% of global wetland extent) followed by North and Central America (26.7%). These two continents also show the highest wetland-to-land ratios (18.7% and 19.9%, respectively)

while Africa and Oceania exhibit the lowest wetland ratios (5.3% and 6.5%, respectively). Regions with high densities of wetlands include South and Southeast Asia, in part due to large swaths of paddy rice fields, the tropics where large riverine complexes exist, and areas north of ~45° N where lakes and peatlands dominate the landscape (Figures 4 and 5). Overall, the patterns of global wetland distribution correspond closely with regional climatic, physiographic, and hydrologic conditions and generally agree with the results from the compilation of multiple wetland inventories undertaken by Davidson et al. (2018).


**Table 3: Continental and global extents of GLWD v2 wetland classes. Values in parentheses represent the continent's percent of the global extent of each class, except for the two bottom rows which refer to all wetlands globally. Areas are in $10^3$ km$^2$ except for totals in the two bottom rows which are in $10^6$ km$^2$. Asia includes all of Russia; North America includes Greenland; Oceania includes Australia, New Zealand, Melanesia, Micronesia, and Polynesia; total land area excludes Antarctica. A breakdown of all wetland classes by country is available in the Supplementary Information.**


| ID | Class Name | Continental area [$10^3$ km$^2$] \| (% of global class area) | | | | | | Global extents | |
|---|---|---|---|---|---|---|---|---|---|
| | | Africa | Asia | Europe & Middle East | North & Central America | South America | Oceania | Global class area [$10^3$ km$^2$] | % of total wetland area |
| 1 | Freshwater lake | 197.3 (9.6) | 407.3 (19.9) | 119.8 (5.8) | 1226.6 (59.9) | 83.3 (4.1) | 13.8 (0.7) | 2048.1 | (11.3) |
| 2 | Saline lake | 34.2 (5.1) | 531.9 (79.7) | 17.0 (2.6) | 22.2 (3.3) | 21.0 (3.2) | 40.9 (6.1) | 667.2 | (3.7) |
| 3 | Reservoir | 40.2 (12.7) | 108.9 (34.5) | 26.3 (8.3) | 88.2 (28.0) | 47.4 (15.0) | 4.6 (1.5) | 315.7 | (1.7) |
| 4 | Large river | 40.6 (10.6) | 177.0 (46.2) | 13.6 (3.6) | 53.1 (13.9) | 93.2 (24.3) | 5.9 (1.5) | 383.6 | (2.1) |
| 5 | Large estuarine river | 6.1 (7.8) | 35.9 (45.7) | 4.1 (5.2) | 12.7 (16.1) | 15.5 (19.7) | 4.3 (5.5) | 78.6 | (0.4) |
| 6 | Other permanent waterbody | 21.9 (3.6) | 214.3 (35.3) | 57.6 (9.5) | 234.1 (38.5) | 46.9 (7.7) | 33.1 (5.4) | 607.7 | (3.3) |
| 7 | Small streams | 20.9 (16.4) | 45.6 (35.8) | 9.4 (7.4) | 21.9 (17.2) | 24.1 (18.9) | 5.4 (4.3) | 127.2 | (0.7) |
| 8 | Lacustrine, forested | 19.6 (4.6) | 67.8 (15.8) | 28.6 (6.7) | 261.3 (60.9) | 49.7 (11.6) | 1.7 (0.4) | 428.8 | (2.4) |
| 9 | Lacustrine, non-forested | 21.0 (4.2) | 154.5 (30.9) | 27.6 (5.5) | 235.4 (47.1) | 55.5 (11.1) | 5.6 (1.1) | 499.6 | (2.7) |
| 10 | Riverine, regularly flooded, forested | 27.9 (7.4) | 114.0 (30.1) | 14.8 (3.9) | 108.6 (28.7) | 109.7 (29.0) | 3.7 (1.0) | 378.6 | (2.1) |
| 11 | Riverine, regularly flooded, non-forested | 31.2 (5.6) | 299.8 (53.9) | 35.2 (6.3) | 121.3 (21.8) | 64.0 (11.5) | 4.6 (0.8) | 556.3 | (3.1) |
| 12 | Riverine, seasonally flooded, forested | 220.3 (27.4) | 157.9 (19.6) | 15.7 (2.0) | 79.1 (9.8) | 311.3 (38.7) | 20.9 (2.6) | 805.2 | (4.4) |
| 13 | Riverine, seasonally flooded, non-forested | 202.4 (22.7) | 323.2 (36.2) | 93.1 (10.4) | 63.0 (7.1) | 114.9 (12.9) | 96.0 (10.8) | 892.6 | (4.9) |
| 14 | Riverine, seasonally saturated, forested | 68.0 (9.7) | 276.9 (39.5) | 31.4 (4.5) | 168.0 (24.0) | 149.2 (21.3) | 7.6 (1.1) | 701.2 | (3.9) |
| 15 | Riverine, seasonally saturated, non-forested | 202.7 (10.0) | 1109.1 (54.6) | 165.6 (8.2) | 270.5 (13.3) | 233.1 (11.5) | 50.4 (2.5) | 2031.3 | (11.2) |
| 16 | Palustrine, regularly flooded, forested | 1.4 (2.0) | 11.6 (15.9) | 5.9 (8.1) | 48.9 (67.5) | 4.3 (5.9) | 0.4 (0.6) | 72.5 | (0.4) |
| 17 | Palustrine, regularly flooded, non-forested | 3.3 (2.8) | 26.1 (22.3) | 6.3 (5.4) | 74.7 (63.6) | 5.8 (5.0) | 1.3 (1.1) | 117.5 | (0.6) |
| 18 | Palustrine, seasonally saturated, forested | 7.3 (5.2) | 28.1 (20.3) | 11.1 (8.0) | 82.0 (59.2) | 9.2 (6.6) | 0.9 (0.7) | 138.5 | (0.8) |
| 19 | Palustrine, seasonally saturated, non-forested | 32.6 (10.8) | 102.2 (33.9) | 28.4 (9.4) | 109.1 (36.2) | 21.2 (7.0) | 8.1 (2.7) | 301.7 | (1.7) |
| 20 | Ephemeral, forested | 4.9 (13.3) | 16.0 (42.8) | 1.5 (4.0) | 7.8 (21.0) | 6.1 (16.5) | 0.9 (2.4) | 37.3 | (0.2) |
| 21 | Ephemeral, non-forested | 12.0 (5.5) | 96.9 (44.2) | 10.4 (4.7) | 32.4 (14.8) | 32.8 (15) | 34.6 (15.8) | 219.1 | (1.2) |
| 22 | Arctic/boreal peatland, forested | 0.0 (0.0) | 737.2 (52.3) | 21.3 (1.5) | 651.9 (46.2) | 0.0 (0.0) | 0.0 (0.0) | 1410.4 | (7.8) |
| 23 | Arctic/boreal peatland, non-forested | 0.0 (0.0) | 858.7 (66.8) | 18.7 (1.5) | 408.8 (31.8) | 0.0 (0.0) | 0.0 (0.0) | 1286.1 | (7.1) |
| 24 | Temperate peatland, forested | 1.4 (0.3) | 143.3 (33.2) | 97.2 (22.5) | 162.3 (37.6) | 15.7 (3.6) | 11.7 (2.7) | 431.7 | (2.4) |
| 25 | Temperate peatland, non-forested | 0.5 (0.2) | 94.3 (40.5) | 80.3 (34.5) | 39.7 (17.1) | 15.3 (6.6) | 2.6 (1.1) | 232.8 | (1.3) |
| 26 | Tropical/subtropical peatland, forested | 129.0 (16.1) | 294.4 (36.6) | 0.0 (0.0) | 21.6 (2.7) | 313.9 (39.1) | 44.6 (5.6) | 803.5 | (4.4) |
| 27 | Tropical/subtropical peatland, non-forested | 7.3 (7.0) | 47.2 (45.7) | 0.0 (0.0) | 10.9 (10.5) | 33.4 (32.3) | 4.6 (4.5) | 103.3 | (0.6) |
| 28 | Mangrove | 29.3 (19.4) | 59.8 (39.7) | 0.4 (0.3) | 23.8 (15.8) | 20.5 (13.6) | 16.9 (11.2) | 150.8 | (0.8) |
| 29 | Saltmarsh | 2.3 (4.0) | 11.6 (19.6) | 6.0 (10.1) | 32.1 (54.2) | 4.7 (7.9) | 2.5 (4.2) | 59.2 | (0.3) |
| 30 | Large river delta | 19.6 (7.0) | 148.7 (53.3) | 12.8 (4.6) | 36.8 (13.2) | 60.3 (21.6) | 0.6 (0.2) | 278.7 | (1.5) |
| 31 | Other coastal wetland | 29.2 (7.9) | 133.5 (36.3) | 33.1 (9.0) | 98.9 (26.9) | 35.9 (9.8) | 37.1 (10.1) | 367.8 | (2.0) |
| 32 | Salt pan, saline/brackish wetland | 109.9 (24.5) | 98.1 (21.9) | 92.4 (20.6) | 18.5 (4.1) | 57.1 (12.7) | 71.9 (16.0) | 447.9 | (2.5) |
| 33 | Rice paddies | 53.7 (4.4) | 1034.8 (85.7) | 22.0 (1.8) | 33.6 (2.8) | 45.7 (3.8) | 17.4 (1.4) | 1207.1 | (6.6) |
| | Total wetlands [$10^6$ km$^2$] (% among all wetlands) | 1.60 (8.8) | 7.97 (43.8) | 1.11 (6.1) | 4.86 (26.7) | 2.10 (11.6) | 0.55 (3.0) | 18.19 | (100) |
| | Total land [$10^6$ km$^2$] (% wetland-to-land ratio) | 29.9 (5.3) | 42.7 (18.7) | 12.0 (9.3) | 24.4 (19.9) | 17.8 (11.8) | 8.5 (6.5) | 135.3 | (13.4) |

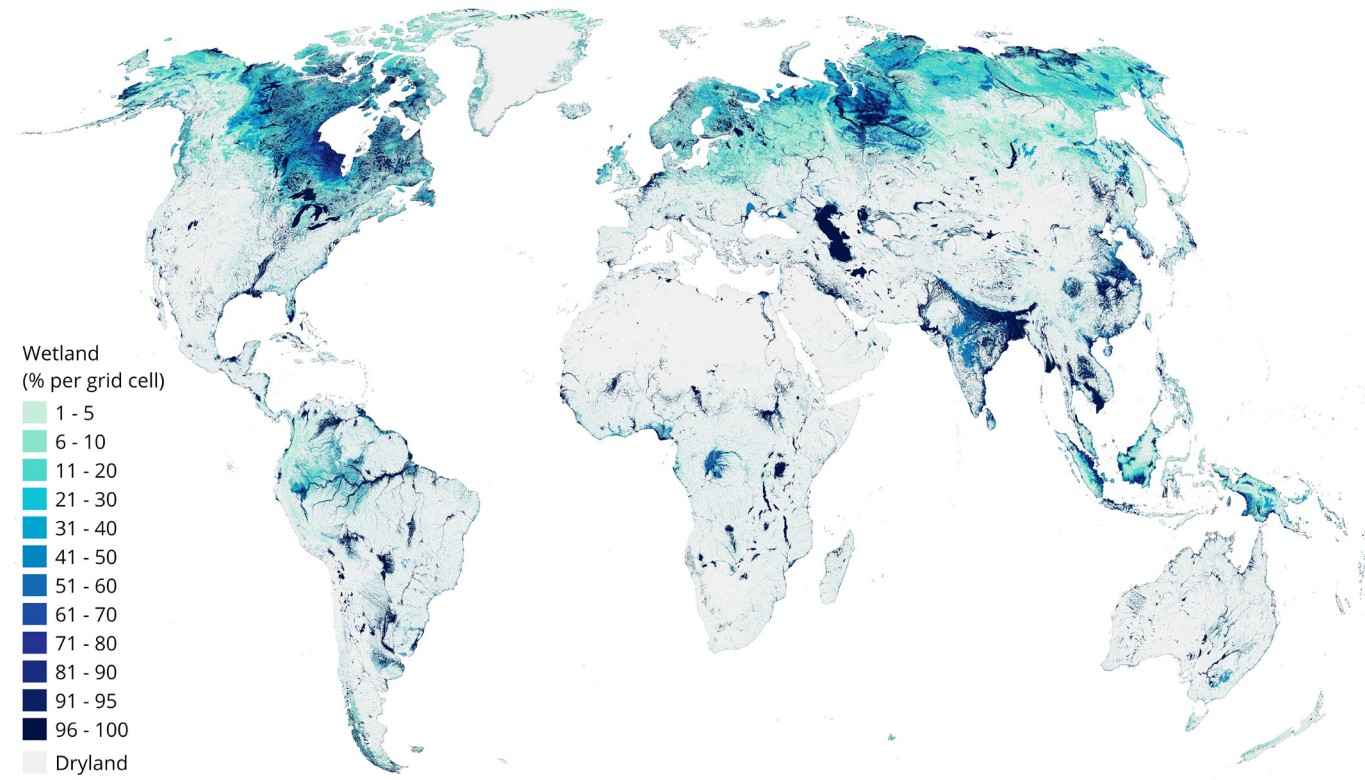

**Figure 4: Total wetland extent as estimated by the Global Lakes and Wetlands Database (GLWD) v2. Values show the combined fractional coverage of all wetland classes per 500 m grid cell. Total wetland extent in each cell is bounded to 1-100%; cells with 0% wetland extent are classified as dryland.**

### 4.2 Wetland class distribution

Grouping specific classes into broad categories reveals global trends of wetland distribution. Unsurprisingly, marine/coastal wetland classes cover only 5% of the total extent, while the majority of 95% of wetlands are inland. Waterbody classes occupy 23% of the total wetland extent while other wetland classes, including emergent and bare wetlands, occupy 77%. Freshwater marshes (i.e., non-forested) and freshwater swamps (i.e., forested) (combined classes 8-21) compose 39% of all wetlands, with two-thirds being marshes (64%) and one-third swamps (36%). Within these marsh and swamp areas, the vast majority (68%) is seasonally flooded or saturated, highlighting the strong intra-annual variability of these wetlands, while 16% are regularly flooded, 4% are ephemeral, and 13% are lacustrine wetlands with no specified periodicity (total not summing to 100% due to rounding). When these marsh and swamp wetlands are grouped by flooding source, riverine wetlands account for the largest share (75%), followed by lacustrine (13%), palustrine (9%), and ephemeral (4%) wetlands.

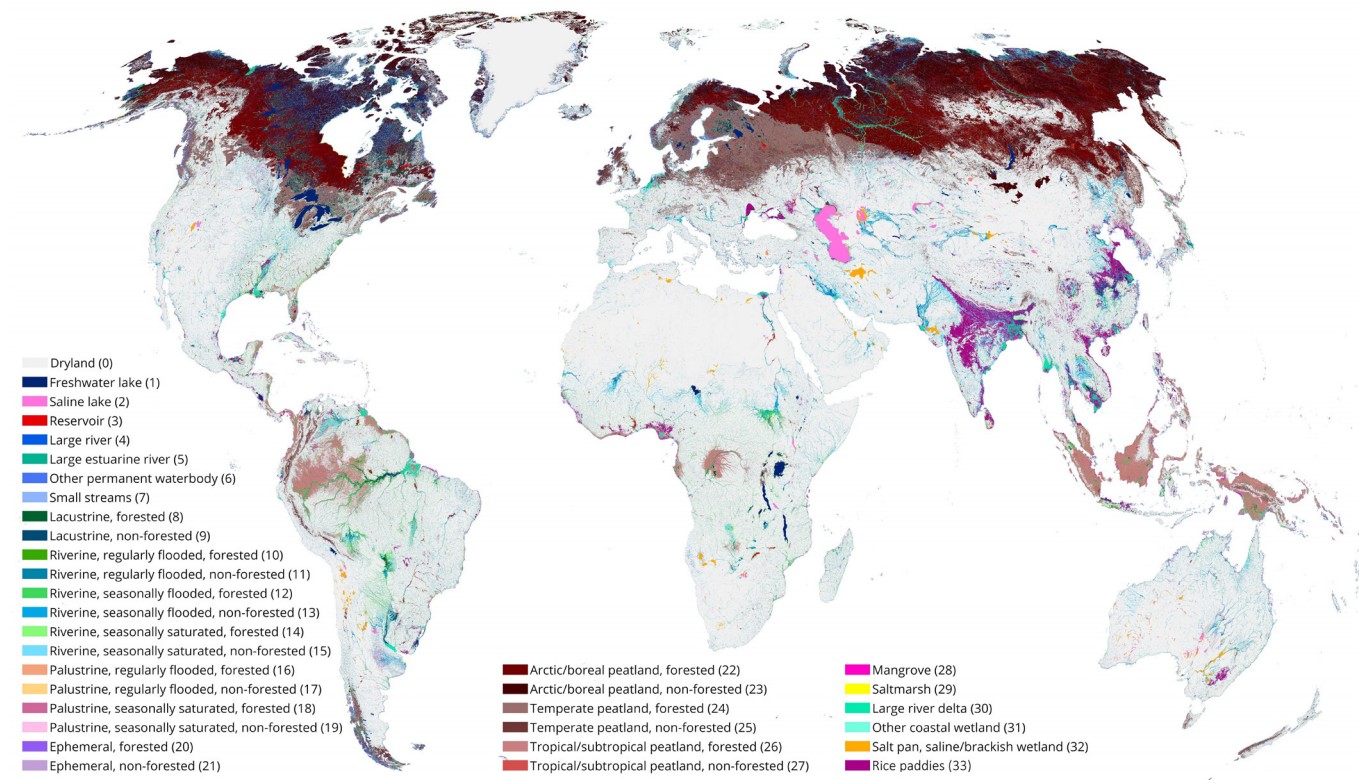

**Legend:**

- Dryland (0)
- Freshwater lake (1)
- Saline lake (2)
- Reservoir (3)
- Large river (4)
- Large estuarine river (5)
- Other permanent waterbody (6)
- Small streams (7)
- Lacustrine, forested (8)
- Lacustrine, non-forested (9)
- Riverine, regularly flooded, forested (10)
- Riverine, regularly flooded, non-forested (11)
- Riverine, seasonally flooded, forested (12)
- Riverine, seasonally flooded, non-forested (13)
- Riverine, seasonally saturated, forested (14)
- Riverine, seasonally saturated, non-forested (15)
- Palustrine, regularly flooded, forested (16)
- Palustrine, regularly flooded, non-forested (17)
- Palustrine, seasonally saturated, forested (18)
- Palustrine, seasonally saturated, non-forested (19)
- Ephemeral, forested (20)
- Ephemeral, non-forested (21)
- Arctic/boreal peatland, forested (22)
- Arctic/boreal peatland, non-forested (23)
- Temperate peatland, forested (24)
- Temperate peatland, non-forested (25)
- Tropical/subtropical peatland, forested (26)
- Tropical/subtropical peatland, non-forested (27)
- Mangrove (28)
- Saltmarsh (29)
- Large river delta (30)
- Other coastal wetland (31)
- Salt pan, saline/brackish wetland (32)
- Rice paddies (33)

**Figure 5: Dominant wetland class for each 500 m grid cell of the Global Lakes and Wetlands Database (GLWD) v2. Total wetland extent in each cell is bounded to 1-100%; cells with 0% wetland extent are classified as dryland. Legend classes include numerical class values in parentheses.**

A more granular inspection of individual classes highlights the predominance of specific wetland types. Among the 33 classes (Table 3 and Figure 5), five classes exceed 1 million km$^2$ globally: freshwater lakes (2.05 million km$^2$, of which 60% are in North and Central America); riverine, seasonally saturated, non-forested wetlands (2.03 million km$^2$); forested arctic/boreal peatlands (1.41 million km$^2$); non-forested arctic/boreal peatlands (1.29 million km$^2$); and rice paddies (1.21 million km$^2$). All peatlands combined (arctic/boreal, temperate, and tropical/subtropical, both forested and non-forested) cover a total of 4.27 million km$^2$, representing nearly a quarter (23%) of the total wetland extent on Earth (Table 3 and Figure 6). They emerge as the dominant wetland type across almost all northern latitudes above 50° N as well as parts of the tropics; however, as the organic soils of peatlands are difficult to map with remote sensing methods, the coarser resolution of the input source data used in GLWD v2 creates local uncertainties. Rice paddies (6.6% of total global wetland extent) occur predominantly throughout southern and eastern Asia including India, northeast China, Vietnam, Thailand, Bangladesh, Sri Lanka, Myanmar, and, to a lesser extent, other regions such as the Nigerian coast and within the Mississippi floodplains (Figure 5). Various other waterbody and wetland classes are regionally dominant, including freshwater lakes in North America and northern Eurasia;

riverine wetlands in South America, Sub-Saharan Africa, and Asia; saline lakes in Central Asia; and ephemeral wetlands in Australia. Small streams occur as small percentages all around the world and dominate in locations where no other wetland type occurs; but they are not easily discernable on the global map (Figure 5) among other more prominent wetland classes. A breakdown of all wetland classes by country is available in the Supplementary Information.

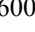

**Figure 6: Total peatland extent as estimated by the Global Lakes and Wetlands Database (GLWD) v2 for all 6 peatland classes combined (arctic/boreal, temperate, and tropical/subtropical; forested and non-forested). Values show fractional coverage per 500 m grid cell. Total peatland extent in each cell is bounded to 1-100%; cells with 0% peatland extent are classified as no peatland.**

### 4.3 Comparison to independent data

To assess the robustness of the resulting GLWD v2 wetland maps, we conducted several comparisons as well as a validation analysis. First, we compared the output of GLWD v2 against independent wetland extents reported in the literature, situating our estimates relative to previous large-scale wetland maps and data compilations. Second, we conducted a validation against

~25,000 global wetland validation samples provided for 8 distinct wetland classes. Finally, we cross-compared GLWD v2 against the predecessor map of GLWD v1, as well as a multi-class satellite-based mapping product at high spatial resolution.

### 4.3.1 Total area comparisons against literature estimates

Table 4 provides comparisons of GLWD v2 against available global, regional, national, or large individual wetland extent estimates for various wetland types, compiled from >70 literature sources including field-based surveys, remote sensing analyses, model simulations, expert assessments, meta-analyses, and national statistics. The global wetland extent of GLWD v2 (18.2 million $km^2$) lies within the wide range of 2.0 to 30.5 million $km^2$ from literature. In a review of global wetland datasets, Hu et al. (2017a) found that estimates from compilation datasets range between 2.8 and 12.7 million $km^2$ and estimates from remote-sensing approaches range between 2.1 and 17.3 million $km^2$. GLWD v2 thus matches the high end of the remote sensing-based estimates. The much larger global wetland extent estimates of 27.5 and 30.5 million $km^2$ produced by Tootchi et al. (2019) and Lane et al. (2023), respectively, are partly explained by the inclusion of model-simulated wetlands that are determined by shallow groundwater occurrences.

The Amazon River Basin is a well-studied wetland hotspot and a frequently used benchmark for new wetland maps. The total wetland extent of GLWD v2 for the entire Amazon Basin is 834,300 $km^2$, of which 444,400 $km^2$ are over lowland floodplains. Fleischmann et al. (2022) compared 29 inundation datasets over lowland regions (elevation <500 m) and estimated the upper bounds of the seasonal minimum and maximum extents as 284,200 $km^2$ and 872,700 $km^2$, respectively. That the area of GLWD v2 falls within the range of these independent estimates demonstrates the ability of GLWD v2 to reasonably capture forested and seasonally inundated wetlands in the tropics, some of the most challenging wetland types to detect. The largest spatial discrepancies in this basin occur for interfluvial (or palustrine) wetlands characterized by shallower and more variable rainfall-driven flooding patterns than the more predictable riparian floodplains. To improve the identification of interfluvial wetland ecosystems, more refined efforts may be needed to include additional, small-scale parameters such as landform (or geomorphic setting) and vegetation (see also section 5.2).

Independent estimates of other large wetland extents across the world, including the Pantanal in South America; the Niger Inland Delta, Sudd Swamps, and Okavango Delta in Africa; and the Mesopotamian Marshes in the Middle East also confirm the overall reliable wetland coverage of GLWD v2, consistently near or within the literature estimates that are often wide-ranging (Table 4). One exception to this is the GLWD v2 estimate of 3.4 million $km^2$ of wetlands in Canada (34% of land area), more than double the national estimate of 1.3 million $km^2$ (13% of land area; Environment and Climate Change Canada, 2016). This discrepancy is explained by the maximalist perspective of GLWD v2 contrasting with the more restricted national definition. Moreover, the lower national estimate is exceeded by independent peatland and lake area estimates alone (Table 4), demonstrating the discrepancies originating from conflicting definitions and goals. This example underlines the value of GLWD v2 in providing a transparent and spatially explicit baseline of composite wetland extents using fractional cell coverages.

**Table 4: Comparisons of global and regional wetland extents. Data sources used in the comparison include field-based surveys, remote sensing products, expert assessments, meta-analyses, and national statistics. Regional estimates are shown in italics. More details on the main characteristics and methodological approach of each referenced literature source are provided in Table A2 (Appendix A).**

| Wetland type [GLWD v2 class(es)] | Region | Extent [$10^3$ km²] GLWD v2 | Extent [$10^3$ km²] Other sources | References (note that many references report on multiple wetland types and regions, thus only selected key references are listed here; multiple references for individual classes are sorted, where possible, from low to high estimates) |
|---|---|---|---|---|
| All types [1-33] | Global | 18,187 | 2,000 – 30,500 | Aselmann & Crutzen, 1989; Finlayson & Davidson, 1999; Fluet-Chouinard et al., 2015; Hu et al., 2017a, 2017b; Lane et al., 2023; Lehner & Döll, 2004; Lieth, 1975; Matthews & Fung, 1987; Melton et al., 2013; Mitsch & Gosselink, 2015; Prigent et al., 2007; Spiers, 1999; Tiner, 2015; Tootchi et al., 2019 |
| | *Canada* | *3,399* | *1,300 – >2,090 [a]* | *Environment and Climate Change Canada, 2016; Tarnocai et al., 2011; Messager et al., 2016* |
| | *Amazon Basin* | *444.4 - 834.3 [b]* | *25.0 – 872.7* | *Fleischmann et al., 2022* |
| | *Pantanal* | *106.4* | *138 – 160* | *Alho, 2005; Mitsch & Gosselink, 2015; Padovani, 2010* |
| | *Congo Cuvette Centrale* | *141.3* | *132 – 360 [c]* | *Dargie et al., 2017; Campbell, 2005; Bwangoy et al., 2010* |
| | *Sudd Swamps* | *64.9* | *30 – 57 – 130 [d]* | *Sutcliffe & Parks, 1999; Ramsar, 2006; Republic of South Sudan, 2015* |
| | *Niger Inland Delta* | *45.7* | *15 – 47* | *Olivry, 1995; Ramsar, 2004; Sutcliffe & Parks, 1989* |
| | *Okavango Delta* | *7.4* | *2.5 – 16* | *McCarthy et al., 2003* |
| | *Mesopotamian Marshes* | *26.7* | *5.4 – 35 [e]* | *Ramsar, 2012, 2015a, 2015b; Al-Handal & Hu, 2015; Buringh, 1960* |
| Freshwater & saline lakes [1, 2] | Global | 2,715 | 2,000 – 4,760 [f] | Mulholland & Elwood, 1982; Downing et al., 2006; Messager et al., 2016; Pi et al., 2022; Verpoorter et al., 2014 |
| Reservoirs [3] | Global | 315.7 | 251.0 – 492.1 | Lehner & Döll, 2004; Downing et al., 2006; Lehner et al., 2011; Wang et al., 2021 |
| Rivers [4, 5, 7] | Global | 589.3 | 404.0 [g] – 662.0 | Allen & Pavelsky, 2018; Raymond et al., 2013; Downing et al., 2012 |
| | *USA* | *41.0* | *30.4* | *Dahl, 2011* |
| Forest swamp [8, 10, 12, 14, 16, 18, 20] | Global | 880.0 [h] – 2,562 | 1,087 – 1,370 | Matthews & Fung, 1987; Gumbricht et al., 2017 |
| | *USA* | *30.9 [h] – 166.5* | *208.9* | *Dahl, 2011* |
| Freshwater marsh [9, 11, 13, 15, 17, 19, 21] | Global | 1,173 [h] – 4,618 | 274.0 – 2,787 | Aselmann & Crutzen, 1989; Gumbricht et al., 2017 |
| | *China* | *97.7 [h] – 590.8* | *217.3 [i]* | *Sun et al., 2015* |
| | *USA* | *40.9 [h] – 281.2* | *185.9 [j]* | *Dahl, 2011* |
| Peatland [22, 23, 24, 25, 26, 27] | Global | 4,268 | 3,700 [k] – 4,232 | Hugelius et al., 2020; Joosten, 2009; Spiers, 1999; Xu et al., 2018 |
| | *Canada* | *1,068* | *1,136* | *Tarnocai et al. 2011* |
| | *Finland* | *48.5* | *90.0* | *Tanneberger et al., 2017* |
| | *Germany* | *10.2* | *12.8* | *Tanneberger et al., 2017* |
| Tropical/ subtropical peatland [26, 27] | Global | 906.9 | 441.0 – 1,700 | Page et al., 2011; Gumbricht et al., 2017 |
| | *Brazil* | *170.2* | *25.0 – 312.3* | *Page et al., 2011; Gumbricht et al., 2017* |
| | *DR Congo* | *67.3* | *2.8 – 115.6* | *Page et al., 2011; Gumbricht et al., 2017* |
| | *Indonesia* | *260.6* | *207.0 – 265.5* | *Page et al., 2011; Gumbricht et al., 2017* |
| Mangrove [28] | Global | 150.8 | 137.6 – 166.0 | Bunting et al., 2018; Giri et al., 2011; Spalding et al. 2010; Sanderman et al., 2018 |
| | *Indonesia* | *30.3* | *29.5* | *Bunting et al., 2022* |
| | *USA* | *2.4* | *2.8* | *Dahl, 2011* |
| Saltmarsh [29] | Global | 59.2 | 22.0 [l] – 400.0 [m] | Chmura et al., 2003; Mcowen et al., 2017; Woodwell et al., 1973 |
| | *USA* | *19.9* | *15.6 – 18.5* | *Dahl, 2011; Worthington et al., 2024* |
| | *Canada* | *8.7* | *3.6* | *Rabinowitz & Andrews, 2022* |
| Large river delta [30] | Global | 278.7 | 305.9 – 710.2 | Syvitski et al., 2009; Ericson et al., 2006; Tessler et al., 2015; Edmonds et al., 2020 |
| | *Amazon* | *32.9 – 72.1 [n]* | *160.0 – 467.0* | *Edmonds et al., 2020* |
| Coastal wetland [28, 29, 30, 31] | Global | 856.4 | 160.0 – 540.0 1,290 [o] | Najjar et al., 2018; Hoozemans et al., 1993; Pendleton et al., 2012 |
| Rice paddies [33] | Global | 1,207 | 1,138 – 1, 663 | Yu et al., 2020; Rosegrant et al., 2002; Portmann et al., 2010; FAOSTAT, 2024 |
| | *China* | *239.6* | *174.7 – 289.5 [p]* | *Yu et al., 2020; National Bureau of Statistics of the People's Republic of China, 2023* |
| | *India* | *524.8* | *304.6 – 478.0 [p]* | *Yu et al., 2020; Government of India, 2023* |
| | *Nigeria* | *26.8* | *18.0 – 45.8 [p]* | *Federal Republic of Nigeria, 2009; FAOSTAT, 2024* |

[a] Sum of total peatland (Tarnocai et al., 2011) and lake extent (Messager et al., 2016)

[b] Low estimate is for lowland floodplains, high estimate is for entire Amazon Basin

[c] Low estimate is for peatland only, high estimate includes all (seasonal) wetlands

[d] Low and middle estimates are for permanent and seasonal swamps, high estimate is for extreme flooding (Republic of South Sudan, 2015)

[e] High estimate is for pre-desiccation marshland extent (i.e., before 1991); low estimate is for post-desiccation (i.e., start of restoration efforts after 2003)

[f] Including extrapolations to lakes ≥ 1 ha

[g] Estimate for rivers wider than 90 m

[h] Counting only riverine classes that are regularly or seasonally flooded (rather than saturated, or undefined in the case of lacustrine and ephemeral classes)

[i] Estimate for marshes and swamps

[j] Estimate for freshwater marshes/wet meadows and shrub wetlands

[k] Estimate of northern peatlands only (>23˚N latitude)

[l] From Chmura et al. (2003), based on inventories from Canada, Europe, Morocco, Tunisia, USA, and South Africa

[m] From Woodwell et al. (1973), extrapolated only from data of USA and not expecting an accuracy better than +-50%

[n] Low estimate is for class 30 (large river delta) only, high estimate is for all wetland classes within delta region

[o] Sum of maximum reported extents of mangrove, saltmarsh, and river delta in previous rows

[p] High estimate for harvested area, meaning that land cropped for rice multiple times in a year is counted multiple times

### 4.3.2 Per-class comparisons against literature estimates

We consider comparisons of individual classes with independent estimates from the literature to be more meaningful in cases where multiple literature estimates converge around a tighter range of values. Therefore, we evaluated GLWD v2 classes by groups tiered by the difference between the maximum and minimum areas found in literature (Table 4): strong agreement (<2-fold discrepancy), moderate agreement (2-3 fold discrepancy) and poor agreement (>3-fold discrepancy).

GLWD v2 classes with strong agreement in literature include reservoirs, rivers, forest swamps, peatlands, mangroves, and rice paddies. Of those classes, all but forest swamps and rice paddies show good agreement between GLWD v2 and global or national independent estimates, with GLWD v2 often falling at the higher end of the reported range. For rice paddy extents, the global area of GLWD v2 agrees well with the global *physical* area but is closer to *harvested* area in some countries (accounting for multiple cropping cycles), suggesting either a regional overestimate by GLWD v2 or a potential interannual

change in physical area or the type of cropping (see Table 4 for country-level examples). In contrast, GLWD v2 estimates for forest swamps are substantially higher than literature because GLWD v2 broadly defines forest swamps as any inundated area (not otherwise claimed by a different wetland class) with >10% tree coverage whereas other definitions of forest swamps also consider more demanding criteria such as soil moisture and hydrophytic vegetation.

Wetland types with moderate literature agreement include lakes and river deltas. In the case of lakes, discrepancies with

literature arise depending on the smallest lake size accounted for, i.e., whether estimates were extrapolated to smaller or even undetectable ponds. GLWD v2 explicitly classifies lakes with a surface area of at least 10 ha, which falls within the range found in literature, and many smaller lakes are expected to be included within the 'other permanent waterbody' class. For large river deltas, disagreements in literature estimates about global extents are largely due to the different approaches in delineating the boundaries of deltas from satellite imagery or topographical information, often leading to only coarse outlines of the delta

region as a whole. The GLWD v2 estimate for large river deltas is lower than independent estimates in part because we prioritized explicit wetland classes, such as rivers, lakes, and rice paddies, over the generic 'large river delta' class in cases of overlap (e.g., Amazon Delta in Table 4).

Finally, wetland types with poor literature agreement include freshwater marshes, tropical/subtropical peatlands, and saltmarshes. Diverging estimates both within literature and to GLWD v2 are due to multiple issues, including differences in

wetland definitions, small wetland occurrences relative to mapping resolutions, difficulties in detection through remote

sensors, and sparse and incomplete reporting. Recent methodological improvements have led to larger estimated extents of

some classes over time. For example, benefitting from improved remote sensing and field data, tropical peatland complexes

in Africa and South America have been mapped to exceed earlier estimates, indicating that previous studies have

underestimated their extent. As GLWD v2 incorporates some of the most recent maps of global peatland and saltmarsh extents,

it captures a similar total area as referenced in these sources. This also confirms that the multi-step merging process did not

cause substantial distortion of original data. Saltmarshes may still be underestimated by GLWD v2 globally, but data quality

and completeness varies regionally as shown by the larger saltmarsh areas for the USA and Canada in GLWD v2 compared to

literature estimates. The area of freshwater marshes estimated by GLWD v2 is substantially higher than the literature range

because GLWD v2 uses freshwater marshes as a catch-all class for all inundated wetlands—not otherwise classified—with

sparse vegetation cover (<10% forest). This goes far beyond the definitions from literature, which tend to rely on narrower

interpretations of vegetation types and soil moisture conditions to identify freshwater marshes (often in ways applicable only

to a specific region).

### 4.3.3 Validation of GLWD v2 against point observations

To validate the resulting maps of GLWD v2, we compared them against a set of global wetland validation samples provided

by Zhang et al. (2023) representing point observations for the year 2020. The validation dataset comprises a total of 24,566

sample points located within the landmask of GLWD v2, equally distributed across the world using a stratified random

approach, and each independently interpreted by five experts with the use of time-series optical observations on the Google

Earth Engine cloud platform. The samples represent 10,324 non-wetland observations and 14,242 wetland observations, the

latter divided into the same 8 wetland classes as represented by the GWL_FCS30 global wetland map of Zhang et al. (2023;

see section 4.3.5): Permanent water, Swamp, Marsh, Flooded flat, Saline, Mangrove, Salt marsh, and Tidal flat.

As the 33 classes of GLWD v2 only partially correspond to the classification system of the validation points, and as GLWD

v2 can report multiple fractional wetland classes within each grid cell, no simple one-to-one match with standard omission and

commission error calculations is possible. Instead, we first created a confusion matrix that tabulates for each validation class

the average fractional wetland extent of each GLWD v2 class, calculated from those grid cells that coincide with a respective

validation point (for results see Table A3 in Appendix A). We then paired groups of GLWD v2 classes that reasonably aligned

with the validation classes (see Table A3 for details). For example, this led to grouping all permanent waterbody types (classes

1-6) as '*Permanent water*', all forested classes to represent '*Swamp*', and all non-forested classes to represent '*Marsh*'. Because

the confusion matrix indicated that validation class '*Saline*' (which refers to saline soils and halophytic plants along saline

lakes) was roughly matched by the '*Saline lake*' and '*Salt pan, saline/brackish wetland*' classes of GLWD v2, we combined

these two classes while recognizing a potential discrepancy to the validation class definition. For the validation class '*Salt

marsh*', the confusion matrix showed the strongest correlations to both the '*Saltmarsh*' and '*Other coastal wetland*' classes in

GLWD v2, which we therefore grouped. This misalignment indicates a shortcoming in GLWD v2 of not accurately

distinguishing saltmarshes from other coastal wetlands. The validation classes '*Flooded flat*' and '*Tidal flat*' have no direct equivalents in GLWD v2 and were thus loosely compared to an amalgamation of all regularly or seasonally flooded classes for flooded flats, and to the '*Other coastal wetland*' and '*Other permanent waterbody*' classes in GLWD v2 for tidal flats.

Using these groupings of GLWD v2 classes, we calculated omission and commission errors as follows: An omission error is assumed to exist for GLWD v2 cells that coincide with a validation point but do not contain any fraction of the paired validation class (including the non-wetland class). A commission error is assumed to exist for GLWD v2 cells that coincide with a validation point but do not contain any fraction of the paired validation class (including the non-wetland class), and the cell is covered by at least 50% of a single validation class grouping that is different from the point's validation class (including the non-wetland class). The latter constraint avoids commission errors for GLWD v2 cells that are occupied only by minority classes, or by a majority class that does not relate to any validation class, such as '*Large river delta*' or '*Rice paddies*'. It should be noted that despite our attempt to replicate traditional omission and commission error calculations, careful interpretation of the results is advised as fractional classes within the GLWD v2 cells obscure a precise colocation against the validation points.

Following these definitions, we calculated an overall accuracy of 90.5% between GLWD v2 classes and validation samples, indicating good overall agreement. Omission errors (Table 5) ranged from 1.1% for '*Permanent water*' to 24.6% for '*Flooded flats*', the latter likely caused by the inherent mismatch of class definitions. Elevated omission errors for '*Marshes*' (17.5%) and '*Salt marshes*' (21.2%) reveal a possible mismatch in class definitions or a limited ability of GLWD v2 to depict these wetland types. Commission errors ranged from 4.3% for '*Mangroves*' to 23.8% for '*Tidal flats*' and 33.6% for '*Flooded flats*', the latter two again likely due to inconsistent definitions. The commission error for '*Permanent water*' (14.7%) can be explained, in part, by historic interpretations of lake extents in GLWD v2, such as the now reduced Aral Sea extent or the fluctuating water area of Lake Chad (see Figure 7), as well as grid cells dominated by small lakes (mostly in northern latitudes) which may coincide with validation points that mark surrounding patches of marsh, swamp, or upland areas within the cell.

Despite the limited alignment between classification systems and the fractional wetland classes in GLWD v2 which introduce ambiguity in the interpretation, we believe that the validation assessment provides strong support regarding the overall robustness of GLWD v2 results.

**Table 5: Accuracy assessment of 8 wetland classes from 24,566 wetland validation point samples provided by Zhang et al. (2023) against GLWD v2 class combinations. For the non-standard definition of omission and commission errors as applied here see main text.**

| Validation class *(and brief description)* | GLWD v2 class grouping *and class number(s)* | Number of validation points | GLWD v2 | |
|---|---|---|---|---|
| | | | Omission error (%) | Commission error (%) |
| Non-wetland | Dryland (non-wetland) *Class 0* | 10,324 | 5.9 | 5.9 |
| Permanent water *(lakes and rivers)* | Permanent waterbody *Classes 1-6* | 2261 | 1.1 | 14.5 |
| Swamp *(forest or shrubs)* | All forested wetlands *Classes 8, 10, 12, 14, 16, 18, 20, 22, 24, 26* | 2952 | 8.2 | 14.4 |
| Marsh *(herbaceous vegetation)* | All non-forested wetlands *Classes 9, 11, 13, 15, 17, 19, 21, 23, 25, 27* | 4112 | 17.5 | 9.6 |
| Flooded flat *(non-vegetated areas along rivers and lakes)* | Lacustrine and all riverine regularly or seasonally flooded *Classes 8-13 and 16-17* | 871 | 24.6 | 33.6 |
| Saline *(saline soils and halophytic plants along saline lakes)* | Saline lake and salt pan, saline/brackish wetland *Classes 2 and 32* | 921 | 8.1 | 12.1 |
| Mangrove *(forests in coastal brackish or saline water)* | Mangrove *Class 28* | 1208 | 8.2 | 4.3 |
| Salt marsh *(herbaceous vegetation in upper coastal intertidal zone)* | Saltmarsh and other coastal wetland *Classes 29 and 31* | 1248 | 21.2 | 13.2 |
| Tidal flat *(coastal zone between high and low tide level)* | Other coastal wetland and other permanent waterbody *Classes 31 and 6* | 669 | 13.5 | 23.8 |

### 4.3.4 Statistical comparison of GLWD v2 against GLWD v1

To demonstrate the progress in upgrading from GLWD v1 to v2, we compared the respective wetland distributions and geographic extents as depicted in the two products. Table A4 (Appendix A) shows the confusion matrix of spatial overlap between classes, and Figure 7 illustrates select visual examples. At the highest level, i.e., when evaluating the agreement with regards to separating wetlands from non-wetlands, GLWD v1 and v2 reach an overall accuracy of 88.2%, mostly reflecting the dominant agreement in non-wetland area. When focusing solely on wetlands, however, only 31.3% of the combined wetland extents are coinciding, indicating a rather strong discrepancy in the distribution of wetland areas between the two datasets. This is also confirmed by a modest F1 score (harmonic mean of precision and recall) of 0.43.

More specifically, GLWD v2 shows an overall omission error across all 12 wetland classes of GLWD v1 of 38.7%. The highest omission error exists for the '*Intermittent wetland/lake*' class (74.4%; Table A4), indicating either high spatial uncertainty in the mostly coarse and generalized delineations of this class in GLWD v1, or a limited ability of GLWD v2 to capture intermittent or ephemeral wetlands at higher resolution. Spatial generalizations in GLWD v1 can also explain why about half of the areas classified as '*Freshwater marsh & floodplain*' or '*Swamp forest, flooded forest*' are mapped as dryland in GLWD v2. The high omission error for the '*Bog, fen, mire (peatlands)*' class in GLWD v1 (58.2%) may be due to the novel

representation of peatlands in GWLD v2 as landscape fractions (rather than binary presence/absence), lowering their overall spatial extent in each overlapping grid cell.

In contrast, the lowest omission error exists for '*Salt pan, saline/brackish wetlands*' (0.1%) which is not surprising given that this GLWD v1 class was used as an input in the creation of GLWD v2. The '*Lake*' class in GLWD v1 is well captured by the '*Freshwater lake*' and '*Saline lake*' classes from GLWD v2, though 9.6% of lake surfaces in GLWD v1 are not covered in GLWD v2 by any wetland class. These omissions of lake areas in GLWD v2—similarly observed for reservoirs and rivers— are mostly caused by known spatial uncertainties in GLWD v1 due to projection issues that introduced substantial misalignments as well as the representation of some reservoirs as circular shapes rather than true shoreline polygons (Lehner and Döll, 2004).

On the other hand, GLWD v2 shows an addition of wetland areas not represented by GLWD v1 that outweighs the omissions, as confirmed by an overall commission error of 61.0%. Additional wetland extents are due to a combination of a) new classes that were not mapped in GLWD v1, and b) a more comprehensive depiction of wetlands in v2. For example, the substantial addition of lake, reservoir and river surfaces as compared to GLWD v1 (see Table A4) can be attributed to the ability of GLWD v2 to represent many smaller waterbodies across various distinct classes. The wetland types in GLWD v2 making the largest spatial additions to GLWD v1 are '*Riverine, seasonally saturated, non-forested*' and '*Rice paddies*', followed by '*Arctic/boreal peatland, non-forested*' and '*Arctic/boreal peatland, forested*'. These and several other classes have not been explicitly mapped in GLWD v1.

Some GLWD v2 classes (e.g., palustrine wetlands and peatlands) do not have direct equivalents in v1 and can only be evaluated in terms of mixed or partial matches with multiple classes, including overlaps with the indiscriminate fractional wetland classes in GLWD v1. For example, the '*0-25% wetlands*' class of GLWD v1, covering large areas of northern Canada including the Hudson Bay Lowlands, overlaps with several of the riverine wetland classes in GLWD v2 as well as the two '*Arctic/boreal peatland*' classes (forested and non-forested), reflecting the refined ability of GLWD v2 to discriminate large wetland complexes into separate ecosystem types. Finally, some mismatches can be explained by regional or class definition issues. For instance, the '*Swamp forest, flooded forest*' class of GLWD v1 aligns best with the '*Tropical/subtropical peatland, forested*' class of GLWD v2. This nominal misalignment is caused by the GLWD v1 class only being present over tropical areas, particularly over the Congo Basin.

Our comparison corroborates that GLWD v2 provides an expanded coverage of wetlands compared to GLWD v1 and offers an improved level of separation into distinct classes, especially for riverine and peatland ecosystems as well as rice paddies. Overall, GLWD v1 represents a much more generalized interpretation of global wetland classes (see Figure 7), underlining the advanced quality and detail of the upgraded GLWD v2.

### 4.3.5 Visual comparisons of GLWD v2 against a multi-class satellite-based product

Given recent remote sensing advances aiming at detecting and mapping distinct wetland types, we compared GLWD v2 against GWL_FCS30, a global 30 m wetland map with a fine classification system designed for dynamic wetland monitoring (Zhang

et al., 2023; 2024). GWL_FCS30 maps 8 wetland classes (the same as those of the validation points presented in section 4.3.3) derived from Landsat and other satellite imagery for the time period 2000-2022 and produced by training regional random forest models on sample points generated from select wetland maps. The lack of alignment in class numbers and class definitions between GLWD v2 and GWL_FCS30 (33 classes vs. 8 classes), combined with mismatching resolutions (500 m vs. 30 m) and cell values (fractional classes vs. binary classes) precludes a simple statistical overlay analysis as the colocation

of paired GWL_FCS30 wetland classes in the larger GLWD v2 cells remains ambiguous. Besides a basic assessment of overall accuracy, we therefore performed a visual comparison in select wetland regions (Figure 7). We aggregated the 30 m grid cells of GWL_FCS30 to 500 m resolution and computed the dominant wetland class in each cell in analogy to the dominant wetland class map of GLWD v2. We also masked the 500 m cells where wetlands (from all classes) occupy less than 50% to compare the fractional cover of wetlands in each product. This approach is intended to focus on the agreement in the distribution of

dominant classes in wetland-dense regions. For reference, Figure 7 also shows depictions of GLWD v1 and optical imagery from Google Maps.

We first assessed the agreement between GLWD v2 and GWL_FCS30 (using GWL_FCS30 data for the year 2020) in their ability to separate wetland from non-wetland cover. The two maps reach an overall agreement of 91.0%, largely driven by the dominant coverage of upland areas. However, the two sources only agree on 27.7% of the shared wetland cover. This

discrepancy reflects major differences in the respective mapping techniques, definitions, goals, and time periods applied, leading to inherently dissimilar global wetland extents with a total wetland area of 18.2 million $km^2$ for GLWD v2 and 6.4 million $km^2$ for GWL_FCS30 (for year 2020; Zhang et al., 2023). Despite drawing from the same data sources in some cases, GLWD v2 and GWL_FCS30 can present substantial disparities due to the process of data fusion in GLWD v2 versus subsampling to training points and random forest classification in GWL_FCS30. In particular, seasonal wetlands can be missed

by remote sensing studies if observations only occur over short time periods.

The visual comparisons shown in Figure 7 (representing the extended period 2000-2020 for GWL_FCS30) highlight instances of both agreement and disagreement between the maps. High agreement is observed between the '*Permanent water*' class of GWL_FCS30 and the open water classes of GLWD v2 (classes 1-6). However, differences in the represented time periods can cause significant deviations even for water surfaces that are deemed permanent. For instance, the Aral Sea and Lake Chad

show larger open water extents in their long-term depiction of GLWD v2 (including the decade of the 1980s) compared to the more recent observation period of GWL_FCS30 (2000-2020) which shows the two lakes in contracted extents. Mangroves show high agreement between the two maps (e.g., Everglades in Figure 7), which is expected as both maps used common mangrove datasets as inputs. In contrast, despite similar inputs, lower agreement is observed for the respective saltmarsh classes (not shown in Figure 7), indicating some confusion with the '*Other coastal wetland*' class in GLWD v2.

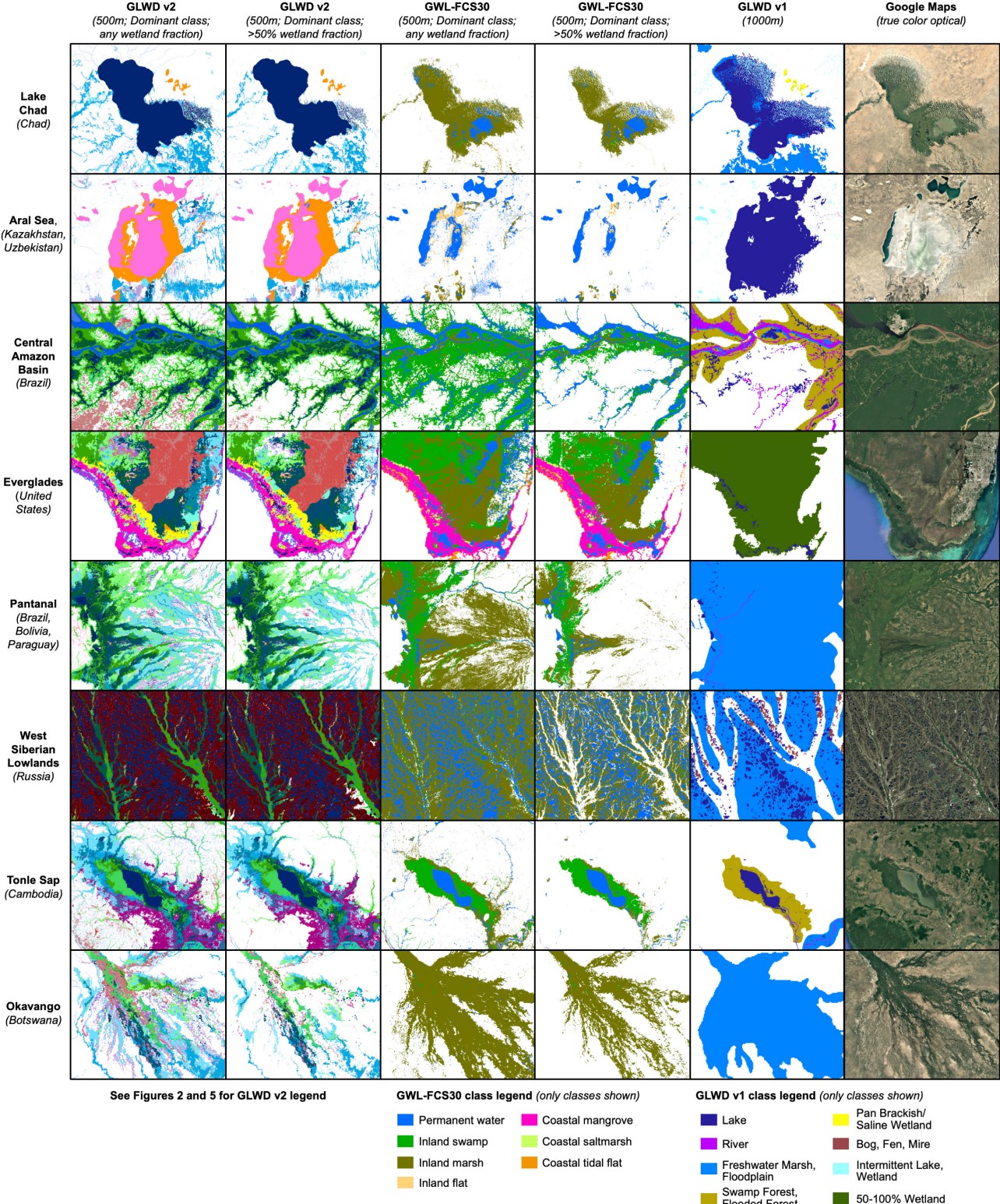

| | GLWD v2<br>*(500m; Dominant class; any wetland fraction)* | GLWD v2<br>*(500m; Dominant class; >50% wetland fraction)* | GWL-FCS30<br>*(500m; Dominant class; any wetland fraction)* | GWL-FCS30<br>*(500m; Dominant class; >50% wetland fraction)* | GLWD v1<br>*(1000m)* | Google Maps<br>*(true color optical)* |
|---|---|---|---|---|---|---|
| **Lake Chad** *(Chad)* | | | | | | |
| **Aral Sea,** *(Kazakhstan, Uzbekistan)* | | | | | | |
| **Central Amazon Basin** *(Brazil)* | | | | | | |
| **Everglades** *(United States)* | | | | | | |
| **Pantanal** *(Brazil, Bolivia, Paraguay)* | | | | | | |
| **West Siberian Lowlands** *(Russia)* | | | | | | |
| **Tonle Sap** *(Cambodia)* | | | | | | |
| **Okavango** *(Botswana)* | | | | | | |

**See Figures 2 and 5 for GLWD v2 legend**

**GWL-FCS30 class legend** *(only classes shown)*

- Permanent water
- Inland swamp
- Inland marsh
- Inland flat
- Coastal mangrove
- Coastal saltmarsh
- Coastal tidal flat

**GLWD v1 class legend** *(only classes shown)*

- Lake
- River
- Freshwater Marsh, Floodplain
- Swamp Forest, Flooded Forest
- Pan Brackish/Saline Wetland
- Bog, Fen, Mire
- Intermittent Lake, Wetland
- 50-100% Wetland

**Figure 7: Comparison of GLWD v2, GWL_FCS30, GLWD v1, and optical imagery from Google Maps over eight globally significant lakes or wetlands. To provide visual comparisons of both class distributions and sub-pixel coverage across data sets, we present the dominant wetland classes per grid cell with two cut-offs of wetland fraction (0% and 50%) at a common resolution of 500 m to eliminate misleading optical effects from visualizations at differing scales. However, the spatial aggregation of GWL_FCS30 from its native 30 m resolution understates its precision in wetland delineation, which is unmatched by the other maps in the comparison. GWL_FCS30 data represent the time period of 2000-2020. For a color legend of the GLWD v2 panels, please refer to Figures 2 and 5.**

Other examples of Figure 7 show overall broad agreement between GLWD v2 and GWL_FCS30, yet with notable differences in the delineation of individual classes. In many cases, overall wetland coverage is comparable, but GLWD v2 offers more distinct classes, in particular varying subclasses of riverine, lacustrine, and palustrine wetland types in large river and lake systems (e.g., Amazon, Pantanal, Tonle Sap, and Okavango in Figure 7). The '*Swamp*' class of GWL_FCS30 (i.e., forested wetlands) aligns primarily with forested types in GLWD v2, while the '*Marsh*' class overlaps with both forested and non-forested classes, including forested and non-forested arctic/boreal peatlands in GLWD v2 (e.g., Siberian Lowlands in Figure 7). This misalignment could be caused by disagreements regarding the tree density needed for classification as forested wetlands between the two sources. Overall, caution is recommended when using forested wetland maps, as they remain among the most uncertain wetland extents globally.

Despite GLWD v2 presenting a larger overall wetland area and additional classes such as rice paddies, it omits some areas that are classified as '*Swamp*' and '*Marsh*' by GWL_FCS30, indicating specific gaps in GLWD v2. Visual inspections suggest that '*Marsh*' areas can extend beyond the edges of peatlands in GLWD v2. These marshes are dispersed geographically and are particularly clustered in northern latitudes. Similarly, some missing '*Swamp*' areas appear to capture minor wetland complexes in tropical regions omitted by GLWD v2, often near the edge of river deltas and coastlines.

In conclusion, there are both areas of agreement and disagreement between the two maps. Nonetheless, given their different classification systems, they display overall reasonable levels of concurrence. Considering the higher spatial resolution of GWL_FCS30 and its ability for short-term updates, and the more refined classification and longer-term vision of GLWD v2, these two products demonstrate the substantial opportunities that exist for combining different mapping approaches as a step towards dynamic remote sensing products of wetland classes in the future.

**5 Discussion**

GLWD v2 provides a comprehensive representation of the world's wetland ecosystems by harmonizing state-of-the-art data sources at grid cell resolutions ranging from ~10 m to 1 km into a target resolution of 500 m. By drawing on global or near-global inputs, 33 individual wetland classes were mapped consistently across the world, avoiding regional discrepancies that can emerge from a patchwork of regional or national data sources. The 33 resulting classes of GLWD v2 improve upon

previously available maps, including GLWD v1. By design, GLWD v2 aims to address the call for consistency and integration in global surface water mapping (Rajib et al., 2024) and to help in closing the gap between field inventory typologies and globally applicable classifications (Davidson et al., 2018).

## 5.1 All-inclusive wetland definition and applied criteria

The compilation of GLWD v2 was carried out with the objective of including all wetlands at their maximum extent rather than enforcing strict wetland type definitions. As a result, each wetland type is determined by a distinct set of criteria, resulting from the approaches and constraints of the original data sources, rather than a single harmonized definition. Different wetland classes were specified by a combination of spatial, temporal, and ancillary characteristics, and can be grouped into three broad categories: waterbodies, other inland wetlands, and other coastal wetlands (Figure 2). Aside from the exceptions described in the Methods, the following general class characterizations can be distilled from our data fusion procedures and merger rules:

- Waterbodies comprise open water surfaces wider than 30 m for rivers and larger than 10 ha for lakes. Areas of small streams were predicted statistically for those exceeding 0.1 m$^3$ s$^{-1}$ (100 liters per second) in average flow or 10 km$^2$ in catchment size. Waterbodies generally have persistent presence of open water surfaces; however, specific waterbody types may not be fully inundated at all times; for example, reservoir polygons may delineate surfaces at high water levels, rivers may encompass multiple shifting channels, and a substantial number of small streams will experience intermittent flow. Other permanent waterbodies include, but are not differentiated for, additional parts of rivers, small lakes, ponds, and artificial water surfaces such as canals, as long as they exceed 30 m in width which reflects the detection limit of the used optical imagery.

- Other inland wetlands represent either periodically inundated or surface-saturated areas of various frequencies, with or without forest cover, or represent organic peatland soils, rice paddies, or salt pans.

- Other coastal wetlands are defined by either a particular vegetation cover or are collectively defined as wetlands located less than 10 m above sea level and connected to the coastline, as these criteria were used across several of the data sources or were introduced during the data fusion process.

The broad wetland definition of GLWD v2 allows it to encompass most national definitions, except for specific wetland types that cannot be reliably mapped globally (e.g., explicit identification of aquaculture ponds, or presence of subterranean or geothermal wetlands). Setting aside missing classes, GLWD v2 may still underrepresent wetland extents due to the detection size and revisit period of observation systems, but not due to restrictions derived from definitions. With its broad wetland definition, GLWD v2 aims to address the widely shared concern that published global wetland extent estimates from either national inventories or remote sensing technology may still underestimate the true global wetland extent (Davidson et al., 2018).

**5.2 Classification design and relationship to existing classification systems**

The multiple factors used to categorize wetland types—including hydrology, inundation, soils, vegetation, landscape position, and connectivity—allow GLWD v2 to represent a wide variety of wetland conditions while filling the need for a generalized and manageable classification system. In particular, the inclusion of criteria beyond inundation, such as vegetation and soil conditions, more closely aligns GLWD v2 with field-based and national classifications and inventories (Ramsar Convention on Wetlands, 2002; Gerbeaux et al., 2018; Junk, 2024). In certain regions, GLWD v2 reaches a level of detail comparable to

national and regional classification schemes.

GLWD v2 does not follow a strictly hierarchical classification approach with one specific criterion for subdivisions at each level. Such an approach would yield a much larger number of subclasses. Instead, our grouping of classes into a simplified, systematic but versatile classification system (Figure 2) allows users to combine classes in various ways for applications where fewer and/or broader classes are more useful. For instance, regrouping of classes can occur along various axes, including open

water vs. vegetated, inundated vs. saturated, forested vs. non-forested, connected to waterbodies vs. isolated, or mineral vs. organic soils.

The design of GLWD v2 classes stemmed from two primary methodological procedures: First, we harmonized existing maps of explicit wetland types. In this process, we selected one representative dataset per class (e.g., one river dataset) wherever possible to reduce the issue of double-counting or temporal mismatches for overlapping water features. Second, once all pre-

defined classes were harmonized, we classified the extent of indiscriminate inundation by attaching hierarchical labels. This classification of indiscriminate inundation incorporates ideas from several classification schemes. We elected to combine components of the *"landscape position, landform, water flow path and waterbody type (LLWW) descriptors"* (Tiner, 2014) along with simple biotic discriminants (forested vs. non-forested). The classification in GLWD v2 diverges from proposed hydro-geomorphic wetland classification schemes (e.g., Brinson, 1993; Semeniuk & Semeniuk, 1995) by not including

landform (slope, channel, depression, etc.) among its criteria, not least due to the lack of high-precision input data to determine small-scale geomorphic features. Instead, GLWD v2 uses inundation and saturation frequency as well as spatial connectivity between wetland ecosystems via contiguous surface water extents as a proxy for hydrological, biogeochemical, and ecological connectivity. This is also why inundation extents and frequencies from GIEMS-D3 and the CaMa-Flood hydrological model were chosen as inputs over purely topographical definitions of floodplains such as those produced by Tootchi et al. (2019) and

Lane et al. (2023).

Although GLWD v2 presents a novel classification scheme, its typology shares basic similarities with some of the most common classification systems, including those of the Ecosystem Functional Groups (EFGs) of the International Union for Conservation of Nature (IUCN) (Keith et al., 2022), the US National Wetland Inventory (Cowardin et al., 1979), and the Ramsar Convention on Wetlands (Table 6). Although classes rarely have one-to-one equivalencies, several comparable

groupings emerge. GLWD v2 is less detailed than IUCN's EFGs for waterbodies but offers more levels of separation for other freshwater wetlands (28 EFGs qualify as wetlands). To increase concordance with the IUCN system and facilitate potential

crosswalks of classifications, the simplified representation of rivers and lakes in GLWD v2 could be further expanded by employing ancillary datasets for river types (e.g., the Global River Classification (GloRiC); Ouellet Dallaire et al., 2019) and lake characteristics (e.g., LakeATLAS; Lehner et al., 2022). An analogous division into bioclimatic regions as proposed in the

IUCN typology (e.g., tropical, temperate, alpine, etc.) could also be added to GLWD v2.

GLWD v2 generally aligns well and shares nomenclature with the system and subsystem levels of the US National Wetland Inventory (hereafter NWI; Cowardin et al., 1979; Cowardin & Golet, 1995), a classification of wetlands and deep-water habitats used by the US Fish & Wildlife Service. Comparing higher levels of classification hierarchy, GLWD v2 applies its landscape connectivity labels (lacustrine, riverine, or coastal) far more broadly than NWI does, as GLWD v2 is inspired by

the LLWW approach. At the lower levels of classification, GLWD v2 follows a vegetation dichotomy similar to the more numerous 'modifiers' used by NWI (e.g., vegetation, soil, sediment). Finally, the Ramsar Convention's classification, which was gradually expanded over time to accommodate the diversity of the world's wetlands and later simplified in the Global Wetland Outlook (Davidson & Finlayson, 2018), covers a similarly large breadth of wetlands as GLWD v2. However, the ambiguity and overlap between some of the Ramsar class definitions (Semeniuk & Semeniuk, 1997; Finlayson, 2016) present

relatively few direct equivalencies with GLWD v2 wetland classes.

**Table 6: Class equivalency between GLWD v2 and common global wetland typologies: the wetland and deep-water classification of the US National Wetland Inventory (NWI) (Cowardin et al., 1979), the classification of the Ramsar Convention on Wetlands, the simplified Ramsar types of the Global Wetland Outlook (GWO) (Davidson & Finlayson,**

**2018), and the IUCN global Ecosystem Functional Groups (EFGs) (Keith et al., 2022). Classes listed on the same row signify partial equivalence, ranging from incomplete overlap to complete nestedness. Additional class overlaps are possible depending on application and we recommend case-by-case re-evaluation of this crosswalk. Some classes from Ramsar, GWO and NWI are not listed on the table because of the absence of an equivalent class in GLWD v2. Class names were modified for brevity.**

| GLWD v2 Class ID and Name | NWI Classification (system, subsystem, water regime modifier) | Ramsar Convention on Wetlands classification system | Global Wetland Outlook (classes/subclasses) | IUCN Ecosystem Functional Groups (EFGs) |
|---|---|---|---|---|
| 1. Freshwater lake | Lacustrine, Limnetic | K- Coastal freshwater lagoons<br>O- Permanent freshwater lakes<br> P- Seasonal/intermittent freshwater lakes | Natural lakes ≥10ha | F2.1 – Large permanent freshwater lakes<br>F2.2 – Small permanent freshwater lakes<br>F2.3 – Seasonal freshwater lakes<br>F2.4 – Freeze-thaw freshwater lakes |
| 2. Saline lake | Lacustrine, Limnetic | Q- Permanent saline/brackish lakes | Natural lakes ≥10ha | F2.6 – Permanent salt and soda lakes<br>F2.7 – Ephemeral salt lakes |
| 3. Reservoir | Lacustrine, Limnetic | 6- Water storage areas | Reservoirs | F3.1 – Large reservoirs |
| 4. Large river | Riverine, Lower Perennial | M- Permanent rivers/streams/creeks | Rivers & streams | F1.2 – Permanent lowland rivers<br>F1.3 – Freeze-thaw rivers and streams<br>F1.5 – Seasonal lowland rivers<br>F1.7 – Large lowland rivers |
| 5. Large estuarine river | Riverine, Tidal | F- Estuarine waters | Rivers & streams | FM1.2 – Permanent open riverine estuaries and bays |
| 6. Other permanent waterbody | | 8- Wastewater treatment areas<br> 9- Canals and ditches | Lakes & pools <10 ha<br>Small/farm ponds | F2.5 – Ephemeral freshwater lakes |
| 7. Small streams | Riverine, Upper Perennial and Intermittent | N- Seasonal/intermittent rivers/streams | Rivers & streams | F1.1 – Permanent upland streams<br>F1.4 – Seasonal upland streams<br>F1.6 – Episodic arid rivers |
| 8. Lacustrine, forested | Palustrine, Forested | W- Shrub-dominated wetlands<br>Xf- Freshwater, tree-dominated wetlands | Forested wetlands | TF1.1 – Tropical flooded forests and peat forests<br>TF1.2 – Subtropical/temperate forested wetlands |

| GLWD v2 Class ID and Name | NWI Classification (system, subsystem, water regime modifier) | Ramsar Convention on Wetlands classification system | Global Wetland Outlook (classes/subclasses) | IUCN Ecosystem Functional Groups (EFGs) |
|---|---|---|---|---|
| 9. Lacustrine, non-forested | Lacustrine, Littoral Palustrine, Emergent | Tp- Permanent freshwater marshes/pools Ts- Seasonal/intermittent freshwater marshes/pools | Marshes & swamps | TF1.3 – Permanent marshes |
| 10. Riverine, regularly flooded, forested | Palustrine, Forested | L- Permanent inland deltas W- Shrub-dominated wetlands Xf- Freshwater, tree-dominated wetlands | Forested wetlands | TF1.1 – Tropical flooded forests and peat forests TF1.2 – Subtropical/temperate forested wetlands |
| 11. Riverine, regularly flooded, non-forested | Palustrine, Emergent | L- Permanent inland deltas Tp- Permanent freshwater marshes/pools | Marshes & swamps | TF1.3 – Permanent marshes |
| 12. Riverine, seasonally flooded, forested | Palustrine, Forested | L- Permanent inland deltas W- Shrub-dominated wetlands Xf- Freshwater, tree-dominated wetlands | Forested wetlands | TF1.1 – Tropical flooded forests and peat forests TF1.2 – Subtropical/temperate forested wetlands |
| 13. Riverine, seasonally flooded, non-forested | Palustrine, Emergent | Ts- Seasonal/intermittent freshwater marshes/pools | Marshes & swamps | TF1.4 – Seasonal floodplain marshes |
| 14. Riverine, seasonally saturated, forested | Palustrine, Forested | W- Shrub-dominated wetlands Xf- Seasonal freshwater, tree-dominated wetlands | Forested wetlands | TF1.1 – Tropical flooded forests and peat forests TF1.2 – Subtropical/temperate forested wetlands |
| 15. Riverine, seasonally saturated, non-forested | Palustrine, Emergent | Tp- Permanent freshwater marshes/pools Ts- Seasonal/intermittent freshwater marshes/pools | Marshes & swamps | TF1.4 – Seasonal floodplain marshes |
| 16. Palustrine, regularly flooded, forested | Palustrine, Forested | W- Shrub-dominated wetlands | Forested wetlands | TF1.1 – Tropical flooded forests and peat forests TF1.2 – Subtropical/temperate forested wetlands |
| 17. Palustrine, regularly flooded, non-forested | Palustrine, Emergent | Tp- Permanent freshwater marshes/pools | Marshes & swamps | TF1.3 – Permanent marshes |
| 18. Palustrine, seasonally saturated, forested | Palustrine, Forested | W- Shrub-dominated wetlands Xf- Seasonal freshwater, tree-dominated wetlands | Forested wetlands | TF1.1 – Tropical flooded forests and peat forests TF1.2 – Subtropical/temperate forested wetlands |
| 19. Palustrine, seasonally saturated, non-forested | Palustrine, Emergent | Ts- Seasonal/intermittent freshwater marshes/pools | Marshes & swamps | TF1.4 – Seasonal floodplain marshes |
| 20. Ephemeral, forested | Palustrine, Forested | W- Shrub-dominated wetlands Xf- Seasonal freshwater, tree-dominated wetlands | Forested wetlands | |
| 21. Ephemeral, non-forested | Palustrine, Emergent | W- Shrub-dominated wetlands Y- Freshwater springs/oases | Marshes & swamps | TF1.5 – Episodic arid floodplains |
| 22. Arctic/boreal peatland, forested | Palustrine, Organic soil | Xp- Forested peatlands and peatswamp | Forested peatlands | TF1.6 – Boreal, temperate, and montane peat bogs |
| 23. Arctic/boreal peatland, non-forested | | U- Non-forested peatlands | Non-forested peatlands | TF1.6 – Boreal, temperate, montane peat bogs |
| 24. Temperate peatland, forested | Palustrine, Organic soil | Xp- Forested peatlands and peatswamp | Forested peatlands | TF1.6 – Boreal, temperate, montane peat bogs |
| 25. Temperate peatland, non-forested | | U- Non-forested peatlands | Non-forested peatlands | TF1.6 – Boreal, temperate, montane peat bogs |
| 26. Tropical/subtropical peatland, forested | Palustrine, Organic soil | Xp- Forested peatlands and peatswamp | Forested peatlands | TF1.1 – Tropical flooded forests and peat forests |
| 27. Tropical/subtropical peatland, non-forested | | U- Non-forested peatlands | Non-forested peatlands | TF1.1 – Tropical flooded forests and peat forests |
| 28. Mangrove | Marine, Subtidal and Intertidal | H- Intertidal forested wetlands | Mangroves | MFT1.2 – Intertidal forests and shrublands |
| 29. Saltmarsh | Estuarine, Intertidal | G- Intertidal marshes | Saltmarshes | MFT1.3 – Coastal saltmarshes and reedbeds |
| 30. Large river delta | Estuarine, Intertidal | F- Estuarine waters H- Intertidal forested wetlands | Coastal deltas | MFT1.1 – Coastal river deltas |
| 31. Other coastal | Estuarine, Intertidal | D- Rocky marine shores E- Sand, shingle, or pebble shores J- Coastal brackish/saline lagoons H- Intertidal forested wetlands | Unvegetated tidal flats Coastal lagoons Shallow subtidal system | FM1.2 – Permanent open riverine estuaries and bays |
| 32. Salt pan, saline/brackish wetland | Lacustrine, Limnetic, Intermittently Flooded | R- Seasonal saline/brackish lakes and flats Sp- Permanent saline/brackish marshes/pools Ss- Seasonal saline/brackish marshes/pools | Saltpans, salinas | F2.7 – Ephemeral salt lakes |
| 33. Rice paddies | Palustrine, Emergent Wetland, Artificially Flooded | 3- Irrigated land, including rice fields | Rice paddy | F3.3 – Rice paddies |


## 5.3 Limitations and uncertainties

Our validation and comparison assessments (section 4.3) confirm an overall reasonable wetland representation of GLWD v2, both at global and regional scales. However, differences in wetland definitions and the intrinsic temporal (both inter- and intra-annual) variability of wetland extents can lead to vast discrepancies and preclude any reliable one-to-one comparison. For example, in our validation against global wetland samples the condensed 8 wetland classes of the observation dataset do not directly align with our approach or outputs, and our applied class aggregations are only partially matching the validation classes. The reported omission and commission errors and overall accuracy therefore encompass an ambiguous mix of detection limitations and incompatible wetland definitions, rendering a detailed interpretation of the achieved quality of GLWD v2 difficult.

The general reliability of the different GLWD v2 classes depends on both their respective data sources and subsequent data manipulation and interpretation steps. As a composite mapping product, GLWD v2 inherits the uncertainties and shortcomings of its data sources. Given the large diversity of input datasets, we offer only brief summaries of key limitations in Table 1 but refrain from discussing the quality of each source in more detail; instead, we refer the reader to the original publications of the source datasets (see Table 1). Overall, the recent growth of optical imagery archives has resulted in high-quality maps of inland surface water, including lakes and rivers, that were directly incorporated into GLWD v2 without substantial alteration. Certain explicit wetland types, such as mangroves and saltmarshes, were available as detailed maps requiring only limited modifications, whereas other classes, such as peatlands or rice paddies, were derived from coarser historical maps or synthesized from multiple input maps. The most challenging wetland types to delineate were those based broadly on connectivity and flood frequency assessments, including lacustrine, riverine, palustrine, and coastal wetlands with varying recurrence intervals of inundation or saturation. These wetland types were mapped from indiscriminate, coarse-resolution multi-sensor estimates spanning over two decades which were downscaled using topography information and then combined with ancillary data to differentiate individual wetland classes. Given the increased complexity in this process and the reliance on expert decisions, these ecosystem types are expected to exhibit higher levels of uncertainty.

Despite our efforts to avoid or reduce double-counting of wetland surfaces and minimize uncertainties stemming from the fusion of diverse inputs, we acknowledge that such uncertainties, distortions, and overestimations likely exist in GLWD v2, especially at local scales and in areas where multiple source datasets overlap. For example, temporal misalignment between water and wetland features are expected across multiple data sources, such as due to migrating channels or shifting littoral edges, which can lead to erroneous overestimations of total water surfaces when merged (by summation) into a static product. Spatial or temporal overestimations may also be caused by projection issues leading to false offsets of waterbody features in multiple overlapping sources, by the insufficient resolution of some source datasets (e.g., by interpreting small wetland patches to cover an entire cell), or by the preferential use of wet periods when mapping wetland extents. Substantial local mismatches are expected between the data sources used for arctic/boreal, temperate, and tropical/subtropical peatlands due to differences in their definition of soil organic content and horizon depth, as well as the accuracy of their distribution. Nonetheless, our

comparisons of GLWD v2 to other datasets (section 4.3) suggest that these uncertainties are neither systematic nor sufficiently large to deviate from most literature estimates. The process of synthesizing multiple data sources through masking, merging, and compositing is essential to produce a comprehensive and coherent map with spatially explicit distinctions between classes, especially given the many different types of wetlands on Earth. However, our harmonization of input sources does not improve upon their individual qualities, hence the original datasets at their native resolutions remain the best sources for specific wetland types.

To be applicable globally, the typology of GLWD v2 simplifies distinctions of certain wetland types while also emphasizing globally observable characteristics. For example, we excluded or used proxy measures for field-level indicators that are not directly observable from space, such as water table depth, hydrophytic vegetation, soil condition, microtopography, bathymetry, and salinity (Tiner, 2016; Gallant, 2015). Similarly, plant productivity and nutrient status of wetland ecosystems are used in some national classifications (e.g., ombrotrophic bog vs. minerotrophic fen in the Canadian classification) but are not applicable globally due to missing information at the necessary level of detail to achieve a reliable discrimination. Moreover, these local key characteristics are of secondary importance to the dominant drivers of wetland condition at the global scale that we used for GLWD v2 (hydrology, vegetation, soil type, and landscape position).

The implementation of landscape position, inundation frequency, and surface connectivity between waterbodies and other wetlands required some notable simplifications that have caused GLWD v2 to deviate from more detailed inventories. First, the separation of coastal wetlands is based on connectivity to the marine coast, which is only a proxy for water salinity or tidal hydrodynamics. This may explain some of the observed confusion of the saltmarsh and other coastal wetland classes (see section 4.3). Second, lacustrine or riverine wetland types were labeled based on surface water connectivity to nearby lakes or rivers, but we did not seek to further separate wetlands fed by groundwater (Tootchi et al., 2019) or local runoff and rainfall (Fan & Miguez-Macho, 2011). Third, palustrine wetland types were intended to represent geographically isolated wetlands, but some can remain connected with waterbodies through subsurface flow (Cohen et al., 2016). Fourth, surface inundation and saturation as depicted in our source datasets ignores non-saturated wet soil conditions, which, if added, may have provided further paths of connectivity. Finally, the applied distance thresholds and methodological assumptions used to determine and further stratify the connectivity-based classes using inundation frequencies were determined by expert judgement. This approach, though guided by visual calibration against known wetland complexes, is prone to subjectivity and ambiguity. Overall, we expect that the subclass distinctions derived from the landscape position, connectivity, and flood frequency analyses are the most uncertain within GLWD v2, and caution should be exercised in applications that rely on their individual characteristics.

Rather than being a time-resolved product, GLWD v2 depicts contemporary conditions and limited aspects of inundation periodicity (seasonal, ephemeral, etc.) as a static map. We argue that a static wetland representation is appropriate to determine the overall extent of wetland ecosystems given that seasonal, annual, or even decadal wetting and drying cycles are part of the ecological condition of a wetland (i.e., the wetland still exists if it is in a naturally drier phase or a dry state). However, the lack of dynamic, time-resolved information in GLWD v2 precludes a more refined classification and definition of wetland

boundaries, such as by delimiting wetlands at their average maximum extent over a specified time period (e.g., as proposed by Junk, 2024). Therefore, GLWD v2 offers only limited utility for applications that require the analysis of dynamic wetland states. Furthermore, the static wetland depiction of GLWD v2 represents a long-term baseline centered around the contemporary period of 1984 to 2020 and should not be used to directly infer or monitor trends over time in global wetland distribution. Some input sources are limited to data without explicit temporality, and despite our best efforts to align associated time periods, there remain unresolved mismatches between sources due to different temporal snapshots (see Table 1) or time-integrated summaries (e.g., flood frequencies). While desirable, a time-resolved version of GLWD v2 at regular intervals would require both a narrower selection of data sources and more lenient assumptions about wetland classes to conform with data limitations. Moreover, temporal representation presents new challenges, such as the high uncertainties of transitional wetland systems that fluctuate between saline, brackish and freshwater types as salinity levels change in response to flooding or drying cycles. Overall, interannual variation in seasonally flooded areas is likely the norm rather than the exception, as exemplified by the analyses of Amazon floodplains (Fleischmann et al., 2022).

## 5.4 Future of mapping wetland ecosystems globally

For continuous monitoring of different wetland types to be achieved, high-resolution remote sensing paired with novel modeling approaches and/or machine learning techniques are needed (e.g., Gallant, 2015; Murray et al., 2022; Bunting et al., 2023). Ideally, such efforts should be supported by wide networks of water level loggers adequately capturing the variety of wetlands across the world. With new satellite missions such as SWOT and NISAR, GLWD v2 may act as a baseline layer and offer a globally applicable classification system onto which new data streams can be added to evaluate decadal-scale changes (Biancamaria et al., 2016). In the future, harmonization of GLWD v2 with time-series information derived from Landsat and/or Sentinel (rivers, lakes, and other permanent surface water) and additional sources such as multi-satellite inundation products could form the backbone of a temporally dynamic representation of wetland ecosystems.

Improvements in spaceborne hydrology observations represent a key component to develop an enhanced approach to detect, classify, and monitor wetlands globally. Dependable estimates of soil surface moisture, sub-canopy inundation, refined topographic data and detection of hydrophytic vegetation would allow for more detailed and reliable class distinctions. Furthermore, the GLWD v2 classification could be expanded by adding labels to waterbodies based on their surroundings, leading to new classes such as 'peatland lakes' or 'floodplain lakes'. Finally, the exploration of a hydro-geomorphic classification could be ideal for functional assessments of wetland ecosystem types and the services they provide (Semeniuk & Semeniuk, 1995; 2011; Davis et al., 2013).

**6 Data availability**

The GLWD v2.0 database, as presented in this manuscript, is available under a CC-BY 4.0 license at https://www.hydrosheds.org/products/glwd and a copy has been deposited at the figshare data repository at https://figshare.com/s/e40017f69f41f80d50df (Lehner et al., 2024a). The data layers are provided in different formats and are

accompanied by a Technical Documentation explaining file names and specifications. *[Note: the temporary figshare URL is for review purposes and will be replaced with a permanent figshare DOI link upon acceptance of the manuscript.]*

**7 Conclusions**

GLWD v2 synthesizes the best available maps and Earth observation data from the last ~30 years into a coherent typology of the world's wetland ecosystems. The resulting 33 wetland types substantially narrow the gap between field-level classification

systems designed for local monitoring or management and globally applicable classifications informing large-scale conservation strategies, Earth system modeling, and international policy making. GLWD v2 provides gridded global maps of dominant wetland classes and fractional cell coverage of each class at 500 m resolution to enable a new generation of research and applications. In particular, the design of GLWD v2 as a set of 33 individual but complementary wetland layers is expected to facilitate the study of specific wetlands of interest while remaining consistent with the total global wetland extent and

distribution. As a comprehensive static wetland map of contemporary (~1984-2020) wetlands generated from satellite and ancillary data, GLWD v2 is an important step in the transition of wetland monitoring from a compilation task to a continuous observation process.

If coupled with time-series information from novel remote sensing technologies, GLWD v2 can provide a foundation to transition towards a monitoring system capable of evaluating trends and variations of individual wetland types. Until that time,

GLWD v2 provides an important baseline of wetland extent and classification that can facilitate the derivation of indicators for tracking progress toward the UN Sustainable Development Goal 6.6 of protecting water-related ecosystems. Given the importance of wetlands at the nexus of water, climate, and biodiversity, the dataset can also inform international policy frameworks such as the Convention on Biological Diversity (CBD), the Convention on Migratory Species (CMS), the Intergovernmental Science-Policy Platform on Biodiversity and Ecosystem Services (IPBES), the Ramsar Convention on

Wetlands, and the United Nations Framework Convention on Climate Change (UNFCCC), among others.

**Additional tables**

**Table A1: Description of data sources shown in Figure 1. Temporal resolutions without recurrence interval are separated into '*Categorical*' and '*Static*' according to whether information on inundation frequency is represented in the classification.**

| ID | Name | Full name | Reference | Spatial resolution | Temporal resolution | Waterbody/Wetland type | Data source description |
|----|------|-----------|-----------|--------------------|--------------------|------------------------|-------------------------|
| 1 | Bunting et al. 2018 | Global Mangrove Watch | Bunting et al. 2018 | 25 m | Categorical | Individual wetland type (mangrove) | Synthetic Aperture Radar (SAR) remote sensing |
| 2 | CIFOR | Center for International Forestry Research | Gumbricht et al. 2017 | 500 m | Static | Classified natural wetlands | MODIS remote sensing, combined with model output |
| 3 | G3WBM | Global 3-second Water Body Map | Yamazaki et al. 2015 | 90 m | Categorical | Indiscriminate open water | Landsat remote sensing |
| 4 | GIEMS-1 | Global Inundation Extent from Multi-Satellites - version 1 | Prigent et al. 2007 | 25 km | 30 days | Indiscriminate inundation | Microwave remote sensing, enhanced with other sensors |
| 5 | GIEMS-2 | Global Inundation Extent from Multi-Satellites - version 2 | Prigent et al. 2020 | 25 km | 30 days | Indiscriminate inundation | Microwave remote sensing, enhanced with other sensors |
| 6 | GIEMS-D15 | Global Inundation Extent from Multi-Satellites - Downscaled 15 arc-seconds | Fluet-Chouinard et al. 2015 | 500 m | Categorical | Indiscriminate inundation | Downscaled microwave remote sensing |
| 7 | GIEMS-D3 | Global Inundation Extent from Multi-Satellites - Downscaled 3 arc-seconds | Aires et al. 2017 | 90 m | 30 days | Indiscriminate inundation | Downscaled microwave remote sensing |
| 8 | GIEMS-MC | Global Inundation Extent from Multi-Satellites - Methane Centric | Bernard et al. 2024 | 25 km | 30 days | Classified natural wetlands | Microwave remote sensing enhanced with ancillary sources |
| 9 | GLAD | Global Land Analysis & Discovery | Pickens et al. 2020 | 30 m | 30 days | Indiscriminate open water | Landsat remote sensing |
| 10 | GLOWABO | GLObal WAter BOdies database | Verpoorter et al. 2014 | ~15 m | Static | Indiscriminate open water | Landsat remote sensing |
| 11 | GLWD v1 | Global Lakes and Wetlands Database - version 1 | Lehner & Döll 2004 | 1 km | Static | Classified natural wetlands | Compilation and synthesis of multiple maps |
| 12 | GLWD v2 | Global Lakes and Wetlands Database - version 2 | Lehner et al., this study | 500 m | Categorical | Classified natural wetlands | Compilation and synthesis of multiple maps |
| 13 | GRWL | Global River Widths from Landsat | Allen & Pavelsky 2018 | 30 m | Static | River channels | Landsat remote sensing |
| 14 | GSW | Global Surface Water | Pekel et al. 2016 | 30 m | 30 days | Indiscriminate open water | Landsat remote sensing |
| 15 | GWL_FCS30 | Global 30 m Wetland Map with Fine Classification System | Zhang et al. 2023 | 30 m | Static | Classified natural wetlands | Landsat and other remote sensing |
| 16 | GWL_FCS30D | Global 30 m Wetland Map with Fine Classification System - Dynamic | Zhang et al. 2024 | 30 m | 1 year | Classified natural wetlands | Landsat and other remote sensing |
| 17 | Hugelius et al. 2020 | - | Hugelius et al. 2020 | 25 km | Static | Individual wetland type (peatland) | Compilation and synthesis of multiple maps |
| 18 | HydroLAKES | - | Messager et al. 2016 | 100 m | Static | Lakes & Reservoirs | Compilation and synthesis of multiple maps |
| 19 | Lane et al. 2023 | - | Lane et al. 2023 | 30 m | Static | Individual wetland type (floodplain) | Landsat remote sensing |
| 20 | Matthews & Fung 1987 | - | Matthews & Fung 1987 | 1 degree (~100 km) | Static | Classified natural wetlands | Compilation and synthesis of multiple maps |
| 21 | Murray et al. 2019 | - | Murray et al. 2019 | 30 m | 30 days | Individual wetland type (salt marsh) | Landsat remote sensing |
| 22 | Salmon et al. 2015 | - | Salmon et al. 2015 | 500 m | Static | Individual wetland type (rice paddies) | MODIS remote sensing |
| 23 | SWAMPS | Surface Water Microwave Product Series | Jensen & McDonald 2019 | 25 km | 10 days | Indiscriminate inundation | Microwave remote sensing, enhanced with other sensors |
| 24 | Tootchi et al. 2018 | Composite Wetland Map | Tootchi et al. 2018 | 90 m | Static | Classified natural wetlands | Compilation and synthesis of multiple maps |
| 25 | WAD2M | Wetland Area and Dynamics for Methane Modeling | Zhang et al. 2021 | 25 km | 30 days | Indiscriminate natural wetlands | Microwave remote sensing enhanced with ancillary sources |

**Table A2: Further information on each literature reference and its related data source(s) used in the comparisons shown in Table 4. Data sources include field-based surveys, remote sensing products, expert assessments, meta-analyses, and national statistics.**

| Order of appearance in Table 4 | Reference | Product name | Spatially explicit | Method | Wetland classes specified | Spatial coverage | Resolution | Time period |
|---|---|---|---|---|---|---|---|---|
| 1 | Aselmann & Crutzen, 1989 | | Yes | Compilation of regional wetland surveys and monographs | Bog, fen, swamp, marsh, floodplain, shallow lake, rice paddies | Global | 2.5° latitude x 5° longitude | |
| 2 | Finlayson & Davidson, 1999 | | No | Summary of regional and international wetland inventories; expert estimates | Natural freshwater wetlands, rice paddies, mangroves, coral reefs | Global | | |
| 3 | Fluet-Chouinard et al., 2015 | GIEMS-D15 | Yes | Remote sensing (downscaled from multi-satellite product) | Inundated areas (three extents: mean annual minimum, mean annual maximum, and long-term maximum) | Global | 15 arc-seconds | 1993-2004 |
| 4 | Hu et al., 2017a | | No | Meta-analysis of existing wetland datasets | | Global | | |
| 5 | Hu et al., 2017b | | Yes | Remote sensing (topographic, land cover, precipitation data) and model simulation for potential wetland distribution; compilation of existing global wetland datasets for actual wetland distribution | All wetlands as defined by Ramsar | Global | 1 km | pre 2000 |
| 6 | Lane et al., 2023 | | Yes | Combination of multiple data sources (remote sensing, model simulation, classification) | Floodplain wetlands and non-floodplain wetlands | Global | 30 m | |
| 7 | Lehner & Döll, 2004 | GLWD version 1 | Yes | Compilation of maps, inventories, remote sensing data | Different types of wetland ecosystems (12 classes) | Global | 30 arc-seconds | |
| 8 | Lieth, 1975 | | No | Expert estimates (results from three consecutive groups of geobotany students at the University of North Carolina, Chapel Hill; adjustments and compromises were made in some cases) | Swamps and marshes, lakes and streams | Global | | ca. 1950 |
| 9 | Matthews & Fung, 1987 | | Yes | Compilation of vegetation, soil properties, and inundation maps | Forested bog, non-forested bog, forested swamp, non-forested swamp, alluvial formations | Global | 1° | |
| 10 | Melton et al., 2013 | WETCHIMP | No | Comparison of model simulations (prognostic-based, remote sensing-based) | Land surface with inundated or saturated conditions (mean annual maximum extent) | Global | | 1993-2004 |
| 11 | Mitsch & Gosselink, 2015 | | No | Meta-analysis of existing wetland datasets | | Global | | |
| 12 | Prigent et al., 2007 | | Yes | Multi-satellite-derived product | Inundated areas (monthly mean estimates; including but not discriminating among rivers, small lakes, irrigated agriculture, ocean-contaminated coastal pixels) | Global | 0.25° | 1993-2000 |
| 13 | Spiers, 1999 | GRoWI | No | Review of (regional) site-based and non-site-based inventories (no continental or global scale maps or remotely sensed imagery) | Freshwater wetlands, peatlands, swamps, lakes and lagoons, coral reefs, seagrasses, mangroves, salt marshes, coastal lagoons, artificial wetlands | Global | | |
| 14 | Tiner, 2015 | | No | Review of published regional wetland inventories | Bogs, fens, swamps, floodplains, marshes, lakes, rice paddies, mangroves, coral reefs | Global | | |
| 15 | Tarnocai et al., 2011 | Peatlands of Canada Database (Version 3) | Yes | Combination of geology maps, soil databases, archived field data, air photo interpretations, survey data, interpolation | Peatlands (bogs, fens, swamps and marshes) | Canada | | |
| 16 | Messager et al., 2016 | HydroLAKES | Yes | Compilation of near-global and regional hydrographic datasets (including remotely sensed); statistical extrapolation | Natural lakes (freshwater and saline), regulated lakes and human-made reservoirs with surface area of at least 10 ha | Global | | |
| 17 | Fleischmann et al., 2022 | | Yes | Intercomparison of inundation datasets (based on remote sensing, hydrological modeling, multi-source) | Inundated areas | Amazon River basin | 12.5 - 25 km | 1950-2020 |
| 18 | Alho, 2005 | | No | Expert estimates (from Global Environment Facility's (GEF) Pantanal/Upper Paraguay Project, detailed watershed management program) | | Pantanal | | |
| 19 | Padovani, 2010 | | Yes | Remote sensing (flood regime and geomorphology) | Flooded areas | Pantanal | 250 m | 2000-2009 |
| 20 | Dargie et al., 2017 | | Yes | Combination of in situ and remotely sensed data (multi-satellite product) | Peatlands | Cuvette Centrale | 50 m | 2000-2010 |
| 21 | Campbell, 2005 | | No | Expert estimates | Swamps and other wetlands | Cuvette Centrale | | |
| 22 | Bwangoy et al., 2010 | CARPE wetland map | Yes | Combination of regional expert knowledge and remote sensing (multi-satellite product) | | Cuvette Centrale | 57 m | |
| 23 | Sutcliffe & Parks, 1999 | | No | Expert estimates | Swamps | Sudd | | |
| 24 | Ramsar, 2006 | | Yes | Expert estimates | Permanent swamps | Sudd | | |
| 25 | Republic of South Sudan, 2015 | | No | National report | Swamps | Sudd | | |
| 26 | Olivry, 1995 | | No | Model simulation (hydrological balance) | Flooded areas | Niger inland delta | | 1991 |
| 27 | Ramsar, 2004 | | Yes | Field surveys | | Niger inland delta | | |

| Order of appearance in Table 4 | Reference | Product name | Spatially explicit | Method | Wetland classes specified | Spatial coverage | Resolution | Time period |
|---|---|---|---|---|---|---|---|---|
| 28 | Sutcliffe & Parks, 1989 | | No | Model simulation (water balance, simple relation between flooded volume and flooded area) | | Niger inland delta | | 1951-1983 |
| 29 | McCarthy et al., 2003 | | Yes | Remote sensing (multi-satellite product), unsupervised classification | Inundated areas | Okavango delta | 1 km | 1972-2000 |
| 30 | Ramsar, 2012 | | Yes | Expert estimates | | Hawizeh Marsh | | |
| 31 | Ramsar, 2015a | | Yes | Delineation from satellite imagery (concordant with administrative area under the authority of the Ministry of Water Resources, Center for the Restoration of the Iraqi Marshlands and Wetlands, CRIMW) | | Central Marshes | | 2000-2014 |
| 32 | Ramsar, 2015b | | Yes | Delineation from satellite imagery (concordant with administrative area under the authority of the Ministry of Water Resources, Center for the Restoration of the Iraqi Marshlands and Wetlands, CRIMW) | | Hammar Marsh | | |
| 33 | Al-Handal & Hu, 2015 | | Yes | Remote sensing (long-term satellite observations) | | Mesopotamian marshes | 250 m | 2000-2012 |
| 34 | Buringh, 1960 | | No | Expert estimates | | Marsh region of Iraq | | pre 2000 |
| 35 | Mulholland & Elwood, 1982 | | No | Review of regional and global studies | Lakes, man-made reservoirs (reservoir size >= 100 km2) | Global | | |
| 36 | Downing et al., 2006 | | No | Combination of global and regional datasets, statistical extrapolation (Pareto distribution) | Natural lakes and ponds, impoundments, farm ponds (size >= 0.001 km2) | Global | | |
| 37 | Pi et al., 2022 | GLAKES | Yes | Remote sensing, deep learning classification | Lakes (size >= 0.03 km2) | Global | 30 m | 1984-2019 |
| 38 | Verpoorter et al., 2014 | GLOWABO | Yes | Remote sensing, extraction algorithm | Lakes (size >= 0.002 km2) | Global | 14.25 m | ca. 2000 |
| 39 | Lehner et al., 2011 | GRanD | Yes | Compilation of existing dam and reservoir datasets, statistical extrapolation (Pareto distribution) | Reservoirs (size >= 0.01 ha) | Global | | |
| 40 | Wang et al., 2021 | GeoDAR | Yes | Combination of global and regional inventories (for dams), combination of multi-source water body datasets, including satellite-based (for reservoirs) | | Global | | |
| 41 | Allen & Pavelsky, 2018 | GRWL | Yes | Remote sensing, statistical extrapolation (Pareto distribution) | Rivers (excluded from database: reservoirs, lakes, canals, Antarctica, Greenland, and water bodies at mean sea level) | Global | | |
| 42 | Raymond et al., 2013 | | Yes | Remote sensing (downscaling of coarser global datasets), statistical extrapolation | Streams and rivers, lakes and reservoirs | Global | | |
| 43 | Downing et al., 2012 | | No | Combination of satellite image interpretation, expert estimates, statistical modeling | Streams and rivers | Global | | |
| 44 | Dahl, 2011 | | No | Sample-based surveys combining remotely sensed imagery and field reconnaissance work | Salt water habitats, freshwater habitats, and upland categories (22 classes) (minimum size of 0.40 ha) | Conterminous United States | | 2004-2009 |
| 45 | Gumbricht et al., 2017 | | Yes | Model simulation (using multiple remote sensing products) | Open water, mangrove, swamps (incl. bogs), fens, riverine and lacustrine, floodplains, marshes (size limitation used by Ramsar for lakes (8 ha) is disregarded) | Tropics and subtropics | 232 m | |
| 46 | Sun et al., 2015 | | No | National survey | Marshes and swamps, lakes, rivers, coastal wetlands | China | | 2009-2013 |
| 47 | Hugelius et al., 2020 | | Yes | Compilation of soil classification maps (including maps derived from machine learning algorithms) | Peatlands (defined as >40 cm surface organic soil material) | Northern Hemisphere (>23˚ latitude) | 10 km | |
| 48 | Joosten, 2009 | IMCG-GPD | No | Compilation of national data, expert estimates | Freshwater peatlands (mangroves, salt marshes, paddies, paludified forests, cloud forests and elfin woodlands, paramos, dambos, cryosols) (minimum peat depth of 30 cm, thus excluding many sub(arctic) and (sub)alpine areas with a shallow peat layer) | Global | | 1990, 2008 |
| 49 | Xu et al., 2018 | PEATMAP | Yes | Meta-analysis of geospatial information collated from a variety of sources at global, regional and national levels | Peatlands | Global | | |
| 50 | Tanneberger et al., 2017 | | Yes | Composite inventory of national peatland information | Peatlands (no minimum peat thickness criterion) | Europe | | |
| 51 | Page et al., 2011 | | No | Compilation of detailed inventories and primary reports of global and tropical peat | Tropical peatlands (including high-altitude) (minimum peat thickness of 30 cm) | Global | | |
| 52 | Bunting et al., 2018 | Global Mangrove Watch | Yes | Classification of remote sensing data | Mangroves | Global | 25 m | 2010 |
| 53 | Giri et al., 2011 | | Yes | Classification of remote sensing data | Mangroves | Global | 30 m | 2000 |
| 54 | Spalding et al., 2010 | World Atlas of Mangroves | Yes | Processing of remote sensing data | Mangroves | Global | | 1999-2003 |
| 55 | Sanderman et al., 2018 | | Yes | Adjustments to spatial domain of Giri et al. (2011) | Mangroves | Global | 30 m | |

| Order of appearance in Table 4 | Reference | Product name | Spatially explicit | Method | Wetland classes specified | Spatial coverage | Resolution | Time period |
|---|---|---|---|---|---|---|---|---|
| 56 | Bunting et al., 2022 | Global Mangrove Watch Version 3.0 | Yes | Classification of remote sensing data | Mangroves | Global | 25 m | 1996-2020 |
| 57 | Chmura et al., 2003 | | No | Compilation of published inventories | Salt marshes | US, Europe, Canada, Tunisia, Morocco, and South Africa | | |
| 58 | Mcowen et al., 2017 | Global saltmarsh distribution | Yes | Compilation of existing saltmarsh distribution data (derived from remote sensing and field surveys) | Salt marshes | Global | | 1973-2015 |
| 59 | Woodwell et al., 1973 | | No | Based on approximate ratios of marsh extent to coastline length (derived from US estuaries); authors do not expect estimates derived this way to be accurate within +- 50% | Salt marshes | Global | | |
| 60 | Worthington et al., 2024 | | Yes | Classification of remote sensing data | Tidal marshes | Global | 10 m | 2020 |
| 61 | Rabinowitz & Andrews, 2022 | | Yes | Compilation of global, provincial and federal datasets (mainly derived from remote sensing) | Salt marshes | Canada | | 1995-2021 |
| 62 | Syvitski et al., 2009 | | Yes | Analysis of remotely sensed data and historical maps | Deltas (dataset includes 33 deltas) | Global | | |
| 63 | Ericson et al., 2006 | | Yes | Combination of aerial photographs, satellite imagery, maps, illustrations, soil properties | Deltas (dataset includes 40 deltas) | Global | 30 arc-seconds | |
| 64 | Tessler et al., 2015 | | Yes | Combination of existing delta dataset (Syvitski et al., 2009), analysis and interpretation of remote sensing data (topography and land cover), soil properties, river network | Deltas (dataset includes 48 deltas) | Global | | |
| 65 | Edmonds et al., 2020 | | Yes | Visual interpretation of Google Earth imagery, simplified five-point delineation approach | Deltas (datasets includes 2,174 delta polygons) | Global | | |
| 66 | Najjar et al., 2018 | | Yes | Compilation of published inventories | Tidal wetlands, estuaries, shelf waters | Eastern North America | | |
| 67 | Hoozemans et al., 1993 | | Yes | Compilation of regional and national inventories | Coastal wetlands (salt marshes, intertidal flats, mangroves); coastal wetlands defined as the areas between approximately MLWS (Mean Low Water Spring) and HHWS (High High Water Spring) | Global | | 1990 |
| 68 | Pendleton et al., 2012 | | Yes | Compilation of international monitoring databases and recently published literature | Tidal marshes, mangroves, seagrass | Global | | |
| 69 | Yu et al., 2020 | SPAM2010 | Yes | Crop disaggregation, optimization and allocation modeling | Rice paddies; database includes 42 major crops; distinction between physical area (area footprint of the crop irrespective of the number of times per year the same area was planted and harvested) and harvested area (accounts for multiple harvests of a crop on the same plot) | Global | 5 arc-minutes | 2010 |
| 70 | Rosegrant et al., 2002 | IMPACT-WATER | No | Compilation of FAO statistics and other sources | Rice (rainfed and irrigated) (harvested area); database includes other crops (wheat, maize, other grains, soybeans, potatoes, etc.) | Global | | 1995 |
| 71 | Portmann et al., 2010 | MIRCA2000 | Yes | Compilation of census-based inventories, global cropland extent grids | Rice (rainfed and irrigated) (harvested area); dataset includes 26 crop classes | Global | 5 arc-minutes | 2000 |
| 72 | FAOSTAT, 2024 | Crops and Livestock Products | No | Compilation of national publications and FAO questionnaires | Rice (harvested area); latest official figures: 2021 for world, 2022 for Nigeria | Global | | 2021, 2022 |
| 73 | National Bureau of Statistics of the People's Republic of China, 2023 | | No | National agricultural census | Rice (planting area) | China | | 2023 |
| 74 | Government of India, 2023 | | No | National survey | Rice (gross area) | India | | 2022-2023 |
| 75 | Federal Republic of Nigeria, 2009 | | No | National report | Rice (production area) | Nigeria | | 2008 |

**Table A3: Confusion matrix showing average fractional grid cell area (percent) of GLWD v2 classes at the location of 24,566 wetland validation samples provided by Zhang et al. (2023). Given the lack of direct equivalencies between the two classification systems, correlations between pairs of individual classes are not as informative as comparisons between groups of classes (e.g., combined waterbody classes 1-6 of GLWD v2 best represent the 'Permanent water' class of the validation points). The GLWD v2 classes highlighted in each column represent the group combinations that**

**were used to match the validation classes and to compute omission/commission rates and accuracy indices in section 4.3.3 and Table 5.**

| | | Class names of wetland validation samples | | | | | | | | |
|---|---|---|---|---|---|---|---|---|---|---|
| ID | GLWD v2 class name | Non-Wetland | Permanent water | Swamp | Marsh | Flooded flat | Saline | Mangrove | Salt marsh | Tidal flat |
| | *Number of validation points* | *10,324* | *2261* | *2952* | *4112* | *871* | *921* | *1208* | *1248* | *669* |
| 0 | Dryland (non-wetland) | 83.5 | 8.7 | 38.5 | 34.4 | 24.1 | 4.1 | 5.7 | 11.6 | 8.4 |
| 1 | Freshwater lake | 0.6 | 34.5 | 1.7 | 3.8 | 14.3 | 0.6 | 0.6 | 4.8 | 2.3 |
| 2 | Saline lake | 0.1 | 7.0 | 0 | 0.3 | 0.7 | 38.7 | 0.3 | 1.2 | 0.5 |
| 3 | Reservoir | 0.1 | 11.3 | 0.3 | 0.5 | 2.6 | 0 | 0 | 0.1 | 0 |
| 4 | Large river | 0.2 | 7.4 | 1.9 | 0.9 | 6.4 | 0 | 0.1 | 0.1 | 0.5 |
| 5 | Large estuarine river | 0.1 | 3.1 | 0.2 | 0.2 | 0.6 | 0 | 2.8 | 2.7 | 9.0 |
| 6 | Other permanent waterbody | 1.1 | 9.7 | 0.4 | 1.5 | 3.9 | 0.3 | 5.7 | 12.9 | 42.9 |
| 7 | Small streams | 0.1 | 0.3 | 0.3 | 0.2 | 0.4 | 0.1 | 0.1 | 0.1 | 0 |
| 8 | Lacustrine, forested | 0.3 | 1.3 | 3.5 | 2.5 | 5.3 | 0.1 | 0 | 0.2 | 0.1 |
| 9 | Lacustrine, non-forested | 0.4 | 2.4 | 0.7 | 4.0 | 4.4 | 0.3 | 0.1 | 1.9 | 0.9 |
| 10 | Riverine, regularly flooded, forested | 0.4 | 0.7 | 5.5 | 2.0 | 2.6 | 0 | 0.1 | 0.1 | 0.1 |
| 11 | Riverine, regularly flooded, non-forested | 0.5 | 1.2 | 0.6 | 2.3 | 2.6 | 0 | 0 | 0.2 | 0.4 |
| 12 | Riverine, seasonally flooded, forested | 0.7 | 0.6 | 9.3 | 3.1 | 2.6 | 0.1 | 0.4 | 0.2 | 0 |
| 13 | Riverine, seasonally flooded, non-forested | 0.7 | 0.7 | 0.7 | 2.8 | 1.9 | 0.3 | 0.1 | 0.8 | 0.4 |
| 14 | Riverine, seasonally saturated, forested | 0.7 | 0.4 | 4.1 | 2.3 | 2.7 | 0.1 | 0.2 | 0.3 | 0.1 |
| 15 | Riverine, seasonally saturated, non-forested | 2.1 | 1.1 | 0.7 | 3.9 | 3.6 | 0.8 | 0.3 | 1.7 | 1.5 |
| 16 | Palustrine, regularly flooded, forested | 0.1 | 0.3 | 0.2 | 0.2 | 1.1 | 0 | 0 | 0.1 | 0 |
| 17 | Palustrine, regularly flooded, non-forested | 0.1 | 0.6 | 0 | 0.4 | 0.7 | 0 | 0 | 0.2 | 0.5 |
| 18 | Palustrine, seasonally saturated, forested | 0.1 | 0.4 | 0.3 | 0.4 | 1.3 | 0 | 0.1 | 0.1 | 0.1 |
| 19 | Palustrine, seasonally saturated, non-forested | 0.2 | 0.6 | 0.1 | 0.5 | 0.8 | 0.1 | 0 | 0.3 | 0.3 |
| 20 | Ephemeral, forested | 0 | 0.1 | 0.2 | 0.3 | 0.3 | 0 | 0.1 | 0.1 | 0 |
| 21 | Ephemeral, non-forested | 0.2 | 0.1 | 0.1 | 0.8 | 0.8 | 0.6 | 0.2 | 0.6 | 0.4 |
| 22 | Arctic/boreal peatland, forested | 1.2 | 0.6 | 8.2 | 11.1 | 5.0 | 0.7 | 0 | 0.2 | 0 |
| 23 | Arctic/boreal peatland, non-forested | 1.0 | 1.1 | 1.1 | 9.6 | 3.1 | 0.4 | 0 | 1.6 | 1.2 |
| 24 | Temperate peatland, forested | 0.6 | 0.1 | 2.0 | 1.2 | 1.0 | 0 | 0 | 0.3 | 0.1 |
| 25 | Temperate peatland, non-forested | 0.4 | 0.2 | 0.4 | 2.7 | 1.2 | 0.2 | 0 | 0.4 | 0.3 |
| 26 | Tropical/subtropical peatland, forested | 1.1 | 0.5 | 12.8 | 1.2 | 0.9 | 0 | 5.2 | 1.2 | 0.6 |
| 27 | Tropical/subtropical peatland, non-forested | 0.2 | 0.3 | 0.3 | 1.3 | 0.6 | 0 | 0.8 | 1.3 | 0.7 |
| 28 | Mangrove | 0.2 | 1.2 | 0.4 | 0.2 | 0.3 | 0 | 65.2 | 9.5 | 3.8 |
| 29 | Saltmarsh | 0.1 | 0.4 | 0.6 | 1.4 | 0.9 | 0 | 0.1 | 18.1 | 3.2 |
| 30 | Large river delta | 0.4 | 0.4 | 2.5 | 1.4 | 1.3 | 0.2 | 0.9 | 3.1 | 2.2 |
| 31 | Other coastal wetland | 0.9 | 2.4 | 1.5 | 1.7 | 1.2 | 0.1 | 10.8 | 21.6 | 18.2 |
| 32 | Salt pan, saline/brackish wetland | 0.3 | 0.3 | 0.5 | 0.7 | 0.3 | 51.8 | 0 | 2.0 | 1.1 |
| 33 | Rice paddies | 1.3 | 0.1 | 0.1 | 0.3 | 0.6 | 0.1 | 0.1 | 0.3 | 0.1 |

**Table A4: Confusion matrix showing fractional grid cell area (in $10^3$ km$^2$) of GLWD v2 classes located in each GLWD v1 class (at 500 m cell resolution). The overlap was performed by first disaggregating GLWD v1 from 1 km (30 arc-second) to 500 m (15 arc-second) resolution and then intersecting it with the fractional wetland area from GLWD v2, using the landmask definition of GLWD v2.**

| GLWD v2 classes | 0. Upland/ Ocean | 1. Lake | 2. Reservoir | 3. River | 4. Fresh-water marsh, floodplain | 5. Swamp forest, flooded forest | 6. Coastal wetland | 7. Salt pan, brackish/ saline wetland | 8. Bog, fen, mire (peatland) | 9. Inter-mittent wetland/ lake | 10. 50-100% wetland | 11. 25-50% wetland | 12. Wetland complex (0-25% wetland) |
|---|---|---|---|---|---|---|---|---|---|---|---|---|---|
| 0. Dryland (non-wetland) | 109458.2 | 231.4 | 40.0 | 88.9 | 1210.3 | 686.4 | 152.8 | 0.3 | 414.5 | 512.7 | 1047.3 | 2511.0 | 788.7 |
| 1. Freshwater lake | 534.2 | 1266.6 | 24.6 | 4.2 | 36.9 | 10.1 | 11.3 | 1.0 | 28.3 | 10.6 | 47.1 | 71.6 | 1.6 |
| 2. Saline lake | 35.8 | 524.5 | 0.0 | 0.1 | 4.6 | 0.3 | 4.6 | 44.9 | 0.0 | 47.3 | 0.4 | 1.6 | 3.2 |
| 3. Reservoir | 116.5 | 36.6 | 139.8 | 6.3 | 4.6 | 1.5 | 0.2 | 0.1 | 0.8 | 0.8 | 0.7 | 6.9 | 1.0 |
| 4. Large river | 202.9 | 7.3 | 2.4 | 102.8 | 30.7 | 16.2 | 1.7 | 0.5 | 3.5 | 1.5 | 7.4 | 5.4 | 1.3 |
| 5. Large estuarine river | 30.1 | 8.6 | 0.0 | 15.6 | 6.5 | 3.3 | 10.4 | 0.7 | 0.2 | 0.1 | 1.7 | 0.9 | 0.5 |
| 6. Other permanent waterbody | 466.7 | 30.2 | 1.7 | 6.0 | 18.1 | 2.8 | 31.3 | 2.0 | 8.9 | 3.5 | 14.9 | 15.5 | 6.3 |
| 7. Small streams | 108.7 | 2.2 | 0.5 | 2.4 | 4.4 | 2.4 | 0.5 | 0.4 | 0.8 | 0.8 | 1.6 | 2.2 | 0.4 |
| 8. Lacustrine, forested | 226.5 | 56.2 | 9.8 | 2.2 | 24.9 | 19.3 | 2.0 | 0.0 | 15.9 | 0.4 | 17.3 | 54.3 | 0.0 |
| 9. Lacustrine, non-forested | 322.3 | 51.4 | 8.6 | 1.7 | 50.7 | 9.0 | 5.6 | 0.0 | 9.1 | 5.8 | 16.7 | 17.0 | 1.7 |
| 10. Riverine, regularly flooded, forested | 227.4 | 10.5 | 2.5 | 18.0 | 31.8 | 35.4 | 1.5 | 0.0 | 10.1 | 0.1 | 15.0 | 26.3 | 0.1 |
| 11. Riverine, regularly flooded, non-forested | 428.0 | 9.8 | 2.6 | 17.5 | 56.2 | 8.1 | 2.0 | 0.0 | 7.3 | 4.0 | 7.3 | 11.9 | 1.6 |
| 12. Riverine, seasonally flooded, forested | 555.3 | 6.2 | 1.2 | 20.2 | 98.2 | 82.8 | 6.7 | 0.0 | 5.7 | 0.4 | 16.1 | 12.2 | 0.5 |
| 13. Riverine, seasonally flooded, non-forested | 684.5 | 12.3 | 1.5 | 12.5 | 100.4 | 15.6 | 7.2 | 0.0 | 3.8 | 35.0 | 8.0 | 8.1 | 3.9 |
| 14. Riverine, seasonally saturated, forested | 499.0 | 11.2 | 3.1 | 9.1 | 66.6 | 31.1 | 3.1 | 0.0 | 10.8 | 0.3 | 23.0 | 43.2 | 0.8 |
| 15. Riverine, seasonally saturated, non-forested | 1687.5 | 16.2 | 3.9 | 10.3 | 143.5 | 27.2 | 9.6 | 0.0 | 8.1 | 21.4 | 16.9 | 47.4 | 39.3 |
| 16. Palustrine, regularly flooded, forested | 47.0 | 5.9 | 0.9 | 0.1 | 1.4 | 0.3 | 0.3 | 0.0 | 2.5 | 0.0 | 3.8 | 10.6 | 0.0 |
| 17. Palustrine, regularly flooded, non-forested | 98.8 | 5.4 | 0.4 | 0.1 | 1.6 | 0.1 | 0.4 | 0.0 | 2.1 | 0.8 | 2.9 | 4.0 | 0.6 |
| 18. Palustrine, seasonally saturated, forested | 93.3 | 7.9 | 1.2 | 0.1 | 3.8 | 1.0 | 1.2 | 0.0 | 4.0 | 0.1 | 7.0 | 18.9 | 0.2 |
| 19. Palustrine, seasonally saturated, non-forested | 258.6 | 7.6 | 1.1 | 0.1 | 5.4 | 0.5 | 2.0 | 0.0 | 3.1 | 3.8 | 4.9 | 8.8 | 5.9 |
| 20. Ephemeral, forested | 26.9 | 1.1 | 0.2 | 0.6 | 3.4 | 0.6 | 0.4 | 0.0 | 0.8 | 0.1 | 1.0 | 2.2 | 0.1 |
| 21. Ephemeral, non-forested | 167.0 | 4.2 | 0.6 | 1.3 | 8.5 | 1.5 | 2.5 | 0.0 | 0.8 | 18.4 | 2.0 | 10.3 | 2.3 |
| 22. Arctic/boreal peatland, forested | 709.9 | 20.5 | 0.6 | 4.2 | 126.5 | 0.0 | 3.0 | 0.0 | 88.2 | 0.1 | 320.0 | 137.4 | 0.0 |
| 23. Arctic/boreal peatland, non-forested | 954.8 | 19.8 | 0.2 | 3.7 | 112.9 | 0.0 | 5.6 | 0.0 | 74.6 | 0.2 | 92.5 | 21.7 | 0.1 |
| 24. Temperate peatland, forested | 313.5 | 5.0 | 0.4 | 0.5 | 18.4 | 0.0 | 0.2 | 0.1 | 0.5 | 0.1 | 30.3 | 62.2 | 0.5 |
| 25. Temperate peatland, non-forested | 179.8 | 2.2 | 0.2 | 0.6 | 17.2 | 0.0 | 0.6 | 1.0 | 1.9 | 2.7 | 6.7 | 15.3 | 4.5 |
| 26. Tropical/subtropical peatland, forested | 562.2 | 3.7 | 0.5 | 9.5 | 46.1 | 151.8 | 15.9 | 0.0 | 0.0 | 0.2 | 4.7 | 5.3 | 3.7 |
| 27. Tropical/subtropical peatland, non-forested | 58.5 | 1.8 | 0.4 | 1.5 | 21.1 | 7.5 | 6.6 | 0.2 | 0.0 | 0.5 | 3.3 | 1.7 | 0.2 |
| 28. Mangrove | 64.2 | 3.8 | 0.0 | 2.3 | 10.6 | 4.7 | 60.4 | 0.4 | 0.0 | 0.1 | 1.6 | 0.2 | 2.5 |
| 29. Saltmarsh | 27.1 | 2.2 | 0.0 | 0.4 | 2.2 | 0.3 | 5.2 | 0.0 | 3.6 | 0.1 | 14.3 | 3.7 | 0.0 |
| 30. Large river delta | 113.0 | 4.3 | 0.2 | 7.3 | 83.7 | 19.2 | 27.9 | 1.4 | 0.0 | 0.2 | 13.9 | 7.6 | 0.0 |
| 31. Other coastal wetland | 246.5 | 6.8 | 0.0 | 3.1 | 19.3 | 9.0 | 51.6 | 2.6 | 1.7 | 2.1 | 13.5 | 8.9 | 2.7 |
| 32. Salt pan, saline/brackish wetland | 23.8 | 26.9 | 0.0 | 0.0 | 2.1 | 0.1 | 1.7 | 377.8 | 0.0 | 11.9 | 0.3 | 1.0 | 2.3 |
| 33. Rice paddies | 1019.3 | 4.7 | 0.6 | 4.7 | 123.5 | 5.3 | 14.9 | 0.6 | 0.0 | 2.7 | 0.7 | 5.9 | 24.2 |
| **Total GLWD v1 wetland area\*** | | 2415.0 | 249.9 | 357.9 | 2496.1 | 1153.4 | 450.9 | 434.0 | 711.6 | 688.8 | 1323.6 | 1185.3 | 112.6 |
| *Omission error of GLWD v2 (all classes) against GLWD v1* | | *9.6* | *16.0* | *24.8* | *48.5* | *59.5* | *33.9* | *0.1* | *58.2* | *74.4* | *45.8* | *45.2* | *0.5* |

\* calculated at middle of range for fractional classes #10, 11, and 12, i.e.: 75% for class 50-100%; 37.5% for class 25-50%; 12.5% for class 0-25%

# Appendix B

 **Additional figures**

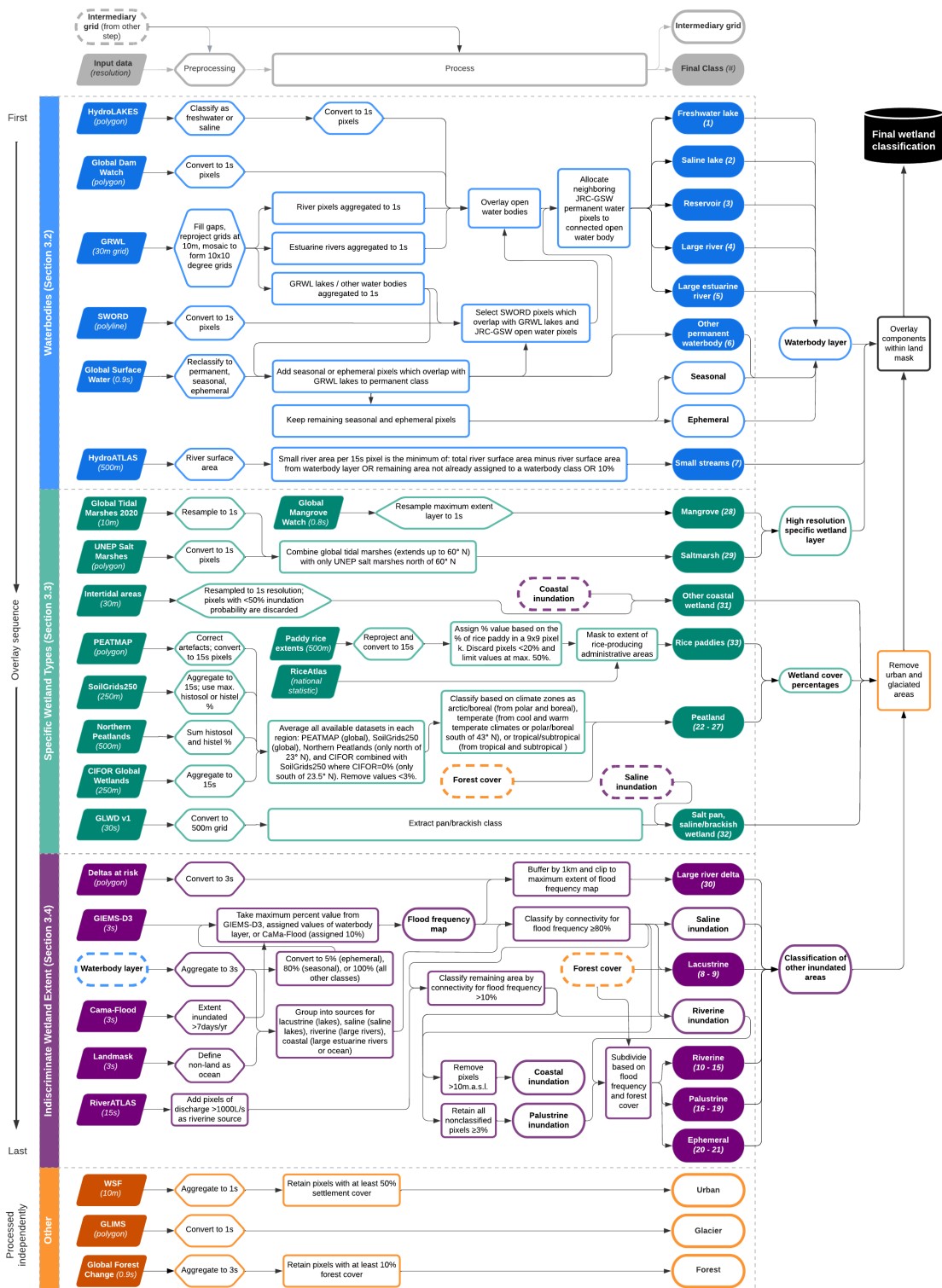

**Figure B1: Detailed schematic of the workflow and processing steps to create GLWD v2, expanding on Figure 3 in the main text. The processing steps are grouped into three main sections, corresponding to sections 3.2 (blue), 3.3 (green), and 3.4 (purple) of the main text. Additional (non-wetland) layers are shown in orange, and a diagram legend is shown in light grey at the top of the figure. This schematic aims to indicate, from top to bottom, the sequential order of insertion of different datasets. In some cases, steps from earlier or later sections of the diagram are used as inputs, as indicated with dashed outlines. From left to right, the schematic describes the input datasets, data preprocessing, main processing steps, and output raster maps of each class (or group of classes). This schematic aims to describe the processing steps of GLWD v2 in high detail while maintaining legibility in a visual format. More specific descriptions of individual steps can be found in the main text in sections 3.2 - 3.4. (Note: an enlarged version of this figure will be available online.)**

### Author contribution

Conceptualization: BL, EF, FA, PB, JGC, ND, CMF, TG, LH, GH, RBJ, PBM, SN, DO, JFP, BP, CP, DY, with support from all authors; Methodology: BL, MA, EF, BP, DY, with support from all authors; Data Contribution: BL, EF, FA, GHA, TG, GH, MCK, LL, DO, TMP, JFP, CP, JW, TAW, DY, XZ; Data Curation: MA, BL; Software, Validation, Visualization: MA, EF, BL; Formal Analysis: BL, MA; Writing—Original Draft: MA, EF, FT, BL; Writing—Review and Editing: BL, with contributions and approval from all authors; Project Administration and Supervision: BL; Funding Acquisition: BL, MT.

### Competing interests

The authors declare that they have no conflict of interest.

### Acknowledgements

The development of the GLWD v2 database was initiated during a workshop on "Mapping the Global Extent and Dynamics of Freshwater Wetlands for CH4 Emissions Modeling" in Washington, DC, in 2017, organized by Stanford University on behalf of the Global Carbon Project – Methane Budget. Financial support for the creation of the database was provided by World Wildlife Fund, Washington, DC, and by The Nature Conservancy, Arlington, VA. The authors wish to express their gratitude to the many contributors beyond the list of co-authors, for their feedback and advice on the design of the database and/or in providing input data along the path of creating GLWD v2. Special thanks are due to Elaine Matthews for her inspiring efforts in advancing global wetland mapping, including her support on the design of this product, throughout her career. And to Nigel Roulet and Tim Moore for their advice on everything related to peat, wet soils, and dirt which encouraged early work on this project.

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
