# Peer review of "Mapping the world's inland surface waters: an upgrade to the Global Lakes and Wetlands Database (GLWD v2)"

_Earth System Science Data, 2024_

## Author Comment (AC1)

Author Comment #1 for ESSD-2024-204 – Lehner et al.: Global Lakes and Wetlands Database GLWD v2

**Referee #1**

Summary:

This study presents the latest version of The Global Lakes and Wetlands Database (GLWD) V2, a foundational reference map depicting major vegetated and non-vegetated wetland classes globally. GLWD V2 was created by harmonizing ground- and satellite-based data products into a single database. Like its predecessor, GLWD V2 avoids double-counting overlapping surface water features and differentiates between natural and non-natural lakes, rivers, and other wetland types. It represents 33 wetland classes, covering approximately 18.2 million km² (13.4% of global land area). GLWD V2 provides an improved representation of inland surface water extents and supports large-scale hydrological, ecological, and conservation applications.

RESPONSE: We thank the Referee for the overall positive assessment of our manuscript, the acknowledgement that GLWD v2 constitutes an improvement of a foundational database, and the constructive comments and suggestions. Following all Referee comments, we prepared a major revision of our manuscript that hopefully addresses all concerns raised. As we noticed several overarching themes shared by both Referees #1 and #2, we would like to start by responding to three general issues before a more detailed point-to-point discussion is provided.

**General Response (for both Referees #1 and #2)**

1. While acknowledging that our new global wetland database provides a substantial and timely contribution to research, there are several comments suggesting the manuscript should contain more explanations on how this database could be used, what kind of applications might be possible, how to integrate it with other data or models, and/or how to increase its usefulness for interdisciplinary research. We appreciate this concern. In the original manuscript, we state in the Abstract that "*GLWD v2 is designed to facilitate large-scale hydrological, ecological, biogeochemical, and conservation applications*", which we kept short due to the desired brevity of the abstract. The original Introduction provides some examples of possible applications of global wetland maps, including "*to quantify [the role of wetlands] within the water, carbon, and nutrient cycles, to plan conservation and restoration actions, to assess and manage human interactions and pressures, … to set a global baseline to contextualize long-term degradation of wetland ecosystems and forecasted risks from climate change, … [and to] monitor the progress towards global targets, such as to track changes in the extent of water-related ecosystems over time as mandated by the UN's Sustainable Development Goal 6.6*" (original lines 70-80, with multiple citations). The Introduction also points out that the predecessor version of our map (i.e., GLWD v1) has been applied for "*advanced research and conservation planning addressing freshwater biodiversity, ecosystem services, greenhouse gas emissions, land surface processes, hydrology, and human health*" (original lines 44-45). Similarly, in the Conclusion section, we suggest that our product can be used to "*inform large-scale conservation strategies, Earth system modeling, and international policy making … such as the Convention on Biological Diversity (CBD), the Convention on Migratory Species (CMS), the Intergovernmental Science-Policy Platform on Biodiversity and Ecosystem Services (IPBES), the Ramsar Convention on Wetlands, and the United Nations Framework Convention on Climate Change (UNFCCC), among others*" (original lines 855-866).

   To avoid repetition with these statements, we carefully expanded the Introduction and Conclusions sections to accommodate the Referees' requests for more application examples. In the Introduction,

we added that global wetland maps are needed to "*guide effective resources management, … as well as to offer interim data to countries currently lacking (or having outdated) national inventories (Davidson et al., 2018).*" We also added that "*as a critical input to hydrologic and Earth system models, global lake and wetland distributions are of particular interest for current and future water resources assessments, carbon and nutrient budget calculations, climate change projections, and other large-scale land surface studies (e.g., Bullock & Acreman, 2003; Lauerwald et al., 2023).*" In the Conclusions, we added that "*the design of GLWD v2 as a set of 33 individual but complementary wetland layers is expected to facilitate the study of specific wetlands of interest while remaining consistent with the total global wetland extent and distribution.*"

Finally, we would like to respectfully note that the predecessor database of GLWD v1 is a similar product to GLWD v2, yet at lower resolution, with less wetland classes, and of substantially lower quality. Despite these shortcomings, the GLWD v1 database has been utilized in a broad range of often interdisciplinary studies which led to more than 2,500 publications (according to Google Scholar). As GLWD v1 is still widely used today, we believe this provides strong evidence of the general applicability of the GLWD databases.

2. Both Referees #1 and #2 raised concern over the fact that GLWD v2 only provides a static product of wetland extents (despite some temporal aspects being represented in the form of seasonal or ephemeral classes). This diminishes the value of GLWD v2 compared to datasets that show dynamic changes for applications requiring a timeseries of wetland extents. We agree with this observation in principle. However, we would also like to note that while saturation or inundation levels may fluctuate, the definition of a wetland as an ecosystem type is more holistic. I.e., a wetland still exists if it is in a drier phase, or a dry state, whether following an annual or decadal cycle. The dry state is part of the ecological condition of a wetland, and as such the extent of the wetland is not dependent on seasonal or interannual fluctuations. The goal of GLWD v2 is to represent the contemporary extent of wetlands from an ecosystem perspective, not a time-resolved inundation perspective. And for that specific goal, we believe that a static map is the appropriate approach. We added an abbreviated version of this explanation to the Discussion (section 5.3) in the manuscript to clarify the value of static wetland maps.

In the original manuscript, we state the 'static' nature of GLWD v2 in the Abstract, and we discuss this 'shortcoming' prominently in the Discussion section (original lines 821-830). Given the concerns raised, we expanded on this discussion in the revision and reframed some parts of the text to describe GLWD v2 more clearly as complementary to (rather than rivaling) temporal datasets. Our goal is to present GLWD v2 as a unique and useful placeholder (or baseline product) until temporally dynamic products with full classifications become available. We are not aware of any existing product that can provide dynamic wetland extents and a comprehensive classification together. The closest may be the GWL_FCS30D wetland map by Zhang et al. (2024) which provides a timeseries from 2000-2022 for 8 wetland classes (compared to the 33 classes of GLWD v2). We took the opportunity of the revision to reword and emphasize (in section 5.3) the importance of static wetland classes in conjunction with maps that provide dynamic wetland extents. And in the existing section 5.4 on the "*Future of mapping wetland ecosystems globally*", we highlight existing endeavors towards creating classified and dynamic wetland maps. Nonetheless, the main intention and focus of our manuscript is to describe GLWD v2, i.e., a static product, rather than to substantially expand on the discussion of advantages or disadvantages when compared to dynamic products.

3. Both Referees #1 and #2 requested a more thorough validation and additional comparisons with other datasets, including with the predecessor database GLWD v1, remote sensing products, and/or field studies. We fully agree with the desire to provide as much validation and as many comparisons as possible. In the original manuscript, Figure 1 shows a comparison of GLWD v2 against 27 other global wetland mapping products or data sources that we are aware of, including GLWD v1 (comparing both spatial and temporal resolutions, as well as discrepancies in their classification methods). Furthermore, and most centrally, Table 4 provides comparisons against >70 individual study results, remote sensing products, and field assessments at global and regional scales, broken down by wetland types, and including some individual wetland areas, across all continents. This table provides upper and lower bounds for each of the comparisons. Table 4 is based on a major literature review and designed as a concise validation table, and we discuss this table in sections 4.3.1 and 4.3.2 in the text, including observed agreements or outliers. Inherently to the design of GLWD v2, we also face the challenge that many of the most reliable input datasets for each wetland class are already included in GLWD v2, leaving remaining independent comparisons (against potentially inferior products) to be less informative.

That said, we addressed the Referees' comments to include more comparisons and validation by revising the manuscript in several major ways:

- We added a new Table A1 in Appendix A which contains the basic characteristics of each of the comparison datasets shown in Figure 1.

- We made it clearer in the manuscript that Table 4 already includes a wide variety of comparison data, from regional to global, including both remote sensing and field-based products. Furthermore, we added a new Table A2 in Appendix A in which we list all data sources shown in Table 4 individually, including a brief description of the main characteristics of each data source (e.g., field-based vs. remote sensing product).

- We added a validation of GLWD v2 against ~25,000 verified point observations of wetland presence/absence globally (new section 4.3.3, including two new tables). As the validation dataset (compiled by Zhang et al., 2023) is not publicly available, we invited 2 co-authors of that team to join our manuscript.

- Furthermore, we conducted a statistical comparison (including a confusion matrix) of GLWD v2 against GLWD v1 to show the substantial changes in the upgrade (new section 4.3.4) and a mostly visual comparison of GLWD v2 against GWL_FCS30 by Zhang et al. (2023; 2024), i.e., a multi-class remote-sensing product (new section 4.3.5, including a new figure).

- Finally, we created an additional table showing the breakdown of all GLWD v2 wetland extents by class and country. Given the size of this table, it cannot be presented in the Appendix of the manuscript, so we opted to add it as Supplementary Information, and we point to it from within the text. We hope this can facilitate that users with national knowledge about wetland distributions can judge the overall quality of GLWD v2 for their own regional or local assessments.

In total, our major revision related to validation and comparison analyses added 6 new tables (1 in manuscript, 4 in Appendix, 1 in Supplementary Information), 1 new figure, and >3 pages of new explanations and discussions, including statistical performance indicators. We hope these additions will sufficiently improve the presentation of the GLWD v2 database and allow readers to judge its validity.

Major comments:

1. The GLWD V2 is a static map, which is insufficient for accurate and comprehensive studies of wetland ecosystems that are fundamental to quantifying their role within the water, carbon, and nutrient cycles, despite its high spatial resolution. I suggest the authors clearly state the significant contributions of this dataset to the field (or its advantages over other similar databases) in the abstract and introduction.

   RESPONSE: Please refer to our **General Responses #1 and #2** above.

2. The authors only provide some comparisons with other databases. Can the authors add comparisons with different satellite-based data for comparison?

   RESPONSE: Please refer to our **General Response #3** above. Besides the new comparisons that we conducted, Figure 1 and Table 4 already include satellite-based comparison products as well, which we clarified in the revised title of Table 4 and in the text.

3. There are too many short paragraphs in section 3. Please consider merging some of them. The overall paragraph structure is loose and needs to be reorganized to be logical and concise.

   RESPONSE: Thank you for this comment. We acknowledge that the methodology, with its nested GIS procedures and sequential dependencies, presents inherent complexity. While the structure of descriptions follows the processing sequence used in creating GLWD v2, we recognize the need to optimize the balance between technical precision, replicability, and clarity.

   To enhance the methodological presentation, we implemented the following improvements in the revised manuscript:

   - We expanded Figure 3 (which provides a schematic overview of the methodology) to include more of the core processing steps. We also strengthened the integration between the written methodology and the figure by incorporating resulting class numbers into the figure and into the titles of the corresponding paragraphs in the text; i.e., the text now relates to the figure more directly.
   - We created a new Figure B1 in Appendix B that is more detailed than Figure 3 and illustrates several of the more complex technical sub-steps; we refer to this figure from the main text.
   - We conducted a thorough review of the text and added refinements and clarifications to the methodological descriptions throughout the manuscript.

   We would also like to note that in the absence of specific feedback from the Referee regarding whether the challenges lie in insufficient methodological detail or overwhelming technical information, we have prioritized enhancing the descriptions while maintaining the comprehensive technical content necessary to fully and transparently present the approach. Our objective remains to provide a thorough methodology that serves both as a clear explanation and a reliable guide for the reproduction of our work.

   As the complexities of each step vary, the step description length is naturally different and there are indeed several short paragraphs. As each paragraph describes an unrelated step in the process, we hesitate to simply merge them as it might increase confusion rather than resolve it. That said, we merged the shortest original section (3.4.4) into the preceding one.

4. The font size of Figure 2 is too small. Please consider making it bigger.

   RESPONSE: Thank you for this suggestion. We redesigned the figure to be able to increase the font size in the figure itself. We will also work with the journal to provide this figure in maximum size in the final publication (and it will be available as a larger image online).

5. The explanation of uncertainties and comparisons with other datasets is insufficient. The potential sources of error for the dataset are also not adequately explained.

   RESPONSE: We appreciate this comment. As for the insufficient amount of comparisons, we would like to refer to our **General Response #3** above, which describes our additions in the revised manuscript. In terms of sources of error, this comment seems also repeated below as Minor Comment #11 (see our response there). As stated in our section 5.3 on limitations and uncertainties, we do not fully explain all the individual underpinning uncertainties of the 25 input data sources as we feel this would be excessive. It would also be speculative because in the merging process, some of the original uncertainties may become obsolete and others may remain. We thus refer to their individual publications instead (listed in Table 1).

   In our major revision, we expanded the descriptions of data sources in Table 1 and explicitly added more characteristics and key limitations of the included datasets, as appropriate. Also, throughout the methodological explanations we describe class-specific problems (such as specific uncertainties and detection problems of open water vs. peatland maps).

6. The advantages of this dataset over others are not clearly highlighted, and the advanced nature of the classification methods is not demonstrated. It appears to be merely an update of the previous version.

   RESPONSE: GLWD v2 has characteristics that make it unique, particularly its 33 individual wetland classes, which to our knowledge is the largest number of ecosystem types represented in any global wetland map. We consider this to be a major advancement over the predecessor GLWD v1.

   Otherwise, our approach has been pragmatic: we utilized existing datasets of different wetland classes and structured them into a meaningful classification based on pre-existing suggestions from other authors (see sections 2.1, 5.1, and 5.2). The resulting 33 classes represent the class distinctions as defined by currently available data sources, rather than an 'ideal' wetland classification. For example, while further classification of lakes by mixing regimes, or distinction between bogs and fens would be valuable, we were unable to identify consistent global datasets to create these classes.

   With respect to providing this pragmatic classification, GLWD v2 is indeed an update of GLWD v1, as stated in the manuscript title. However, given the substantial nature of the improvements, we changed the expression from 'update' to 'upgrade' to better communicate this.

   Please also refer to our **General Response #3** above, which describes additional data validation and comparison steps to demonstrate the advanced quality of the new GLWD v2 database.

7. The discussion in section 5.2 about uncertainties, distortions, and overestimations in regions with overlapping data sources is puzzling. The fusion of multiple data sources should reduce uncertainties in the final results, not increase them.

RESPONSE: We see the Referee's point (assuming it is about section 5.3 'Limitations and uncertainties'), but we wish to clarify that fusing multiple wetland data sources can indeed REDUCE uncertainties, but also AMPLIFY them depending on the methods applied. For example, if an indiscriminate wetland map shows 50% flooding in a grid cell, yet without specification where that flooding occurs or what exact wetland types it includes (some of our input sources provide such indiscriminate information), and a second map shows for the same grid cell an explicit ephemeral lake that covers 25% of the cell area, then this ephemeral lake could be part of the 'flooded' area, or exist in addition to it, depending on whether the first map was able to depict it. If the fusion rules combine the two maps by summation, then the resulting 75% wetland coverage in the cell would be erroneous if the first map already included the ephemeral lake. I.e., both inputs were correct in what they showed, but the particular fusion method increased the uncertainty. This is only one example of many that apply when merging 25 different datasets which all exhibit varying definitions of classes, varying resolutions, and varying spatial accuracies. Of particular concern are multiple maps that show the same water surfaces yet with a slight spatial offset (due to limits in accuracy). In this case, a simple overlay, often intended to correct for missing features, would increase the total surface area; which would constitute an erroneous, systematic overestimation in surface area.

In our revision, we carefully reviewed section 5.3 and made some adjustments to explain this issue more clearly.

8. The advantages of this dataset compared to other remote sensing-based land cover classifications are not apparent. Remote sensing data can maintain high spatial resolution and be dynamically updated.

RESPONSE: We fully agree with the Referee's statement about remote sensing data being able to facilitate dynamic wetland mapping. While GLWD v2 does not claim to be inherently 'superior' to remote sensing-based products, it has specific characteristics that current remote sensing products do not provide. GLWD v2 represents 33 wetland classes originating from many independent sources that each focused on specific classes. Such a detailed classification has not yet been met by any single- or multi-satellite-based product. While we acknowledge the lack of dynamic updates as a limitation of GLWD v2, which we also discuss in the manuscript, we strongly believe that this limitation does not diminish the applicability or value of GLWD v2 as a baseline depiction of current wetland extents with a refined classification. Please also see our **General Responses #1 (applicability) and #2 (static nature of GLWD v2)** where we explain how the revised manuscript has been improved regarding these aspects.

Minor comments:

1. Abstract: The author needs to explain the advantages of GLWD V2 rather than just stating it as an update of GLWD V1.

RESPONSE: Thank you for this suggestion. We added to the Abstract that GLWD v1 has a nominal grid cell resolution of 1 km and depicts 12 wetland classes, which clarifies the improvements of GLWD v2 in terms of resolution and number of classes (now 500 m grid cell resolution with sub-cell information from 10 m pixels; and 33 wetland classes). The other main improvement, which we believe is already clearly stated, is that GLWD v2 was "generated by harmonizing the latest ground- and satellite-based data products" as opposed to the 20-year old GLWD v1.

2. Introduction: Add a table to describe all similar databases from different data sources, including the former GLWD V1, their time period, time step, theory or method, resolution, advantages, and disadvantages.

RESPONSE: We are not aware of any other database that is truly similar to GLWD (v1 and v2) in terms of providing a global map with a broad range of distinct wetland classes. There are, however, other global datasets of unclassified remote sensing imagery showing indiscriminate inundation or water extents, or of individual wetland classes or small groups of classes, such as lakes and/or river floodplains only. These alternative products are presented in Figure 1 (24 existing datasets, including GLWD v1, and 3 planned data sources), stratified by their spatial and temporal resolutions. This figure is intended to not only provide a comprehensive overview of existing (similar) databases, but to also indicate the historic developments, complexity, and broad range of wetland mapping products. Furthermore, another selection of 25 global wetland products is listed in Table 1; these products are those included in the generation of GLWD v2 but also represent many of the best current global wetland maps of individual classes or small groups of classes.

To address the request of the Referee for more information, we expanded the manuscript in two ways in the revision: a) we added a new Table A1 in Appendix A which contains the basic characteristics of each of the 24 existing datasets shown in Figure 1. And b), in response to a related comment of Referee #2, we added information regarding the time periods for each of the datasets listed in Table 1.

3. Importance of the Database: Clearly explain to the readers and the community why this database is important.

RESPONSE: Please refer to our **General Response #1** regarding the applicability and importance of our global wetland map.

4. Figure 1: Please explain 'G3WBM', 'GIEMS1', and all other abbreviations in full the first time they appear in the text.

RESPONSE: Please refer to our response to Minor Comment #2 above: we added a new Table A1 in Appendix A which lists each dataset presented in Figure 1. This new table also spells out all dataset abbreviations and we now point to this table from the caption of Figure 1.

5. Definitions and Data Sources: Please add a table to describe the comparison data used in section 4.

RESPONSE: All comparison datasets or data sources used in section 4 are referenced in Table 4; or are referenced directly in the text if not contained in Table 4. There are over 70 references included in Table 4.

To address the suggestion of the Referee, while not overwhelming the manuscript itself with an elaborate table of more than 70 rows, we added a new Table A2 in Appendix A which lists all data sources shown in Table 4, including a brief description of the main characteristics of each data source.

6. Methods: Please consider rewriting this part, especially merging some short paragraphs in sections 3.2-3.4, as they are not very clear now.

RESPONSE: We are sorry that the Referee considers the Methods sections 3.2-3.4, spanning 230 lines in total, to be unclear. There are indeed several short paragraphs, but they are separated in order to add clarity as they describe unrelated steps in the processing, thus we hesitate to simply merge them. Instead, we carefully revisited the methodological explanations throughout the manuscript and added clarity wherever possible.

See also our response to Major Comment #3 above where we explain how we related the methodological explanations more directly to the improved Figure 3.

7. Figure 3: Please add explanations for the boxes with different shapes and colors. Also, center Figure 3.

RESPONSE: Thank you for this suggestion. We added explanations about the different shapes and colors of the boxes show in Figure 3 (in the figure caption). The centering of the figure will depend on the journal's final formatting.

8. Uncertainties: How do you deal with uncertainties arising from different data sources and their spatial and temporal resolutions?

RESPONSE: We believe that much of the answer to this comment is provided across the Methods section (in particular the overview provided in section 3.1) and parts of the Discussion section. As we are dealing with the combination of 25 different input data sources, each with its own uncertainties and temporal and spatial resolution, we applied a sequence of data selection and ranking procedures accompanied by dedicated data fusion rules that aim to reduce uncertainties and double-counting. We were not able to resolve the problem of differing time periods of the input datasets, other than by selecting only input sources that are complementary and represent contemporary states of wetlands.

See also our response to Major Comment #5 above and Minor Comment #11 below.

9. Section 4.2.3: Please provide more details or static numbers to describe the differences between the GLWD V2 data and other datasets.

RESPONSE: There is no section 4.2.3 in the manuscript, so we assume that this comment refers to section 4.3 which presents our comparisons of wetland extents, including Table 4 which summarizes >70 comparisons. We also assume the Referee suggests adding statistical metrics for these comparisons, such as biases or standard deviations, either by individual class or across all classes. As for providing generalized metrics to describe differences, we are concerned about the appropriateness of doing so because we do not have any comparison datasets at a global scale that are known to be correct, or that use comparable definitions. E.g., the global wetland extent in literature ranges from 2 to 30.5 million square kilometers. Our estimate of 18.2 million square kilometers would thus fall centrally into this range, indicating only a small numerical bias, yet the range boundaries are from datasets that have rather different definitions; hence we consider neither the lower, middle, nor upper boundary to be right or wrong. We have intentionally chosen to compare GLWD v2 against the ranges of other datasets as presented in Table 4 as we believe that this is a more appropriate way to showcase the (known) discrepancies of existing wetland estimates, both at global and regional scales, given their differing definitions; and to demonstrate that the results from GLWD v2 are within these ranges and are therefore plausible. Where our results are outside the range, we discuss the discrepancy in the text.

Nonetheless, to address the Referee's concern, we added several additional comparisons to the manuscript in our major revision, as described in our **General Response #3** above. In particular, we added a new Table A2 in Appendix A which lists each of the comparison datasets contained in Table 4, including their basic characteristics. And we added comparisons between GLWD v2 and GLWD v1 and between GLWD v2 and a multi-class remote sensing product (GWL_FCS30), including numerical indices from a confusion matrix and omission/commission assessments.

We hope that the additional information on comparisons will help the reader to judge the validity of the GLWD v2 database.

10. Classification Methods: Please explain the necessity of using different classification methods here.

RESPONSE: We assume that this comment refers to section 5.2 which is discussing the GLWD v2 classification against other existing classifications. This comparison is a pragmatic effort as other classifications already exist, and we are neutral to the argument whether there is a 'necessity' to create different classifications. Most classifications are purpose-driven, i.e., they are optimized for certain objectives that they are intended to serve (e.g., a wetland classification for conservation purposes may differ from one that is used in methane emission assessments). In our case, we adapted parts of classification methods that can be reasonably applied to global maps. Therefore, we chose to keep section 5.2 only as a comparison rather than adding a justification for different classification methods. We believe that discussing the need for different classification schemes is beyond the scope of our manuscript.

11. Uncertainties and Shortcomings: line 788 Please list all the uncertainties and shortcomings in section 7.8.

RESPONSE: There is no section 7.8 in the manuscript, but uncertainties and shortcomings are indeed discussed in the original manuscript as of line 788 in section 5.3 ("Limitations and uncertainties"). However, we are not sure what additional shortcomings the Referee would suggest for us to include here. We tried to be transparent and comprehensive in this section. As stated, we do not explain all the individual underpinning uncertainties of the 25 input data sources as we feel this would be excessive and speculative because in the merging process, some of the original uncertainties may become obsolete and others may remain. We thus refer to their individual publications instead. However, in our major revision we expanded Table 1 and explicitly added more characteristics and key limitations of the included datasets, which we hope addresses this request.

12. Bias and Uncertainties: lines 789: The authors need to describe the bias and uncertainties with some static numbers.

RESPONSE: There is no section on Bias and Uncertainties, line 789 is about the underpinning source datasets, and in the original manuscript, the term "bias" is only used once, so we are not sure what exactly this comment refers to. We assume it is similar to Minor Comment #9 in that it requests adding more statistical metrics to describe the validity of our results. We therefore refer to our respective response there.

With kind regards,
Bernhard Lehner, on behalf of all co-authors

---

## Author Comment (AC2)

**Referee #2**

The manuscript presents an updated version of the Global Lakes and Wetlands Database (GLWD), which integrates modern ground and satellite-based data to create a harmonized global map of inland surface waters and wetlands. This update provides enhanced resolution and additional classification layers compared to its predecessor (GLWD v1), offering a more detailed and consistent representation of inland surface waters. The contribution is substantial and timely, addressing critical gaps in the representation of wetlands and their dynamic properties, which are crucial for studies in hydrology, ecology, and environmental management.

RESPONSE: We are very thankful for the overall positive review of the Referee and the constructive comments and suggestions. Following all Referee comments, we prepared a major revision of our manuscript that hopefully addresses all concerns raised. As we noticed several overarching themes shared by both Referees #1 and #2, we would like to start by responding to three general issues before a more detailed point-to-point discussion is provided.

**General Response (for both Referees #1 and #2)**

1. While acknowledging that our new global wetland database provides a substantial and timely contribution to research, there are several comments suggesting the manuscript should contain more explanations on how this database could be used, what kind of applications might be possible, how to integrate it with other data or models, and/or how to increase its usefulness for interdisciplinary research. We appreciate this concern. In the original manuscript, we state in the Abstract that "*GLWD v2 is designed to facilitate large-scale hydrological, ecological, biogeochemical, and conservation applications*", which we kept short due to the desired brevity of the abstract. The original Introduction provides some examples of possible applications of global wetland maps, including "*to quantify [the role of wetlands] within the water, carbon, and nutrient cycles, to plan conservation and restoration actions, to assess and manage human interactions and pressures, … to set a global baseline to contextualize long-term degradation of wetland ecosystems and forecasted risks from climate change, … [and to] monitor the progress towards global targets, such as to track changes in the extent of water-related ecosystems over time as mandated by the UN's Sustainable Development Goal 6.6*" (original lines 70-80, with multiple citations). The Introduction also points out that the predecessor version of our map (i.e., GLWD v1) has been applied for "*advanced research and conservation planning addressing freshwater biodiversity, ecosystem services, greenhouse gas emissions, land surface processes, hydrology, and human health*" (original lines 44-45). Similarly, in the Conclusion section, we suggest that our product can be used to "*inform large-scale conservation strategies, Earth system modeling, and international policy making … such as the Convention on Biological Diversity (CBD), the Convention on Migratory Species (CMS), the Intergovernmental Science-Policy Platform on Biodiversity and Ecosystem Services (IPBES), the Ramsar Convention on Wetlands, and the United Nations Framework Convention on Climate Change (UNFCCC), among others*" (original lines 855-866).

   To avoid repetition with these statements, we carefully expanded the Introduction and Conclusions sections to accommodate the Referees' requests for more application examples. In the Introduction, we added that global wetland maps are needed to "*guide effective resources management, … as well as to offer interim data to countries currently lacking (or having outdated) national inventories*

*(Davidson et al., 2018)."* We also added that *"as a critical input to hydrologic and Earth system models, global lake and wetland distributions are of particular interest for current and future water resources assessments, carbon and nutrient budget calculations, climate change projections, and other large-scale land surface studies (e.g., Bullock & Acreman, 2003; Lauerwald et al., 2023)."* In the Conclusions, we added that *"the design of GLWD v2 as a set of 33 individual but complementary wetland layers is expected to facilitate the study of specific wetlands of interest while remaining consistent with the total global wetland extent and distribution."*

Finally, we would like to respectfully note that the predecessor database of GLWD v1 is a similar product to GLWD v2, yet at lower resolution, with less wetland classes, and of substantially lower quality. Despite these shortcomings, the GLWD v1 database has been utilized in a broad range of often interdisciplinary studies which led to more than 2,500 publications (according to Google Scholar). As GLWD v1 is still widely used today, we believe this provides strong evidence of the general applicability of the GLWD databases.

2.  Both Referees #1 and #2 raised concern over the fact that GLWD v2 only provides a static product of wetland extents (despite some temporal aspects being represented in the form of seasonal or ephemeral classes). This diminishes the value of GLWD v2 compared to datasets that show dynamic changes for applications requiring a timeseries of wetland extents. We agree with this observation in principle. However, we would also like to note that while saturation or inundation levels may fluctuate, the definition of a wetland as an ecosystem type is more holistic. I.e., a wetland still exists if it is in a drier phase, or a dry state, whether following an annual or decadal cycle. The dry state is part of the ecological condition of a wetland, and as such the extent of the wetland is not dependent on seasonal or interannual fluctuations. The goal of GLWD v2 is to represent the contemporary extent of wetlands from an ecosystem perspective, not a time-resolved inundation perspective. And for that specific goal, we believe that a static map is the appropriate approach. We added an abbreviated version of this explanation to the Discussion (section 5.3) in the manuscript to clarify the value of static wetland maps.

    In the original manuscript, we state the 'static' nature of GLWD v2 in the Abstract, and we discuss this 'shortcoming' prominently in the Discussion section (original lines 821-830). Given the concerns raised, we expanded on this discussion in the revision and reframed some parts of the text to describe GLWD v2 more clearly as complementary to (rather than rivaling) temporal datasets. Our goal is to present GLWD v2 as a unique and useful placeholder (or baseline product) until temporally dynamic products with full classifications become available. We are not aware of any existing product that can provide dynamic wetland extents and a comprehensive classification together. The closest may be the GWL_FCS30D wetland map by Zhang et al. (2024) which provides a timeseries from 2000-2022 for 8 wetland classes (compared to the 33 classes of GLWD v2). We took the opportunity of the revision to reword and emphasize (in section 5.3) the importance of static wetland classes in conjunction with maps that provide dynamic wetland extents. And in the existing section 5.4 on the *"Future of mapping wetland ecosystems globally"*, we highlight existing endeavors towards creating classified and dynamic wetland maps. Nonetheless, the main intention and focus of our manuscript is to describe GLWD v2, i.e., a static product, rather than to substantially expand on the discussion of advantages or disadvantages when compared to dynamic products.

3. Both Referees #1 and #2 requested a more thorough validation and additional comparisons with other datasets, including with the predecessor database GLWD v1, remote sensing products, and/or field studies. We fully agree with the desire to provide as much validation and as many comparisons as possible. In the original manuscript, Figure 1 shows a comparison of GLWD v2 against 27 other global wetland mapping products or data sources that we are aware of, including GLWD v1 (comparing both spatial and temporal resolutions, as well as discrepancies in their classification methods). Furthermore, and most centrally, Table 4 provides comparisons against >70 individual study results, remote sensing products, and field assessments at global and regional scales, broken down by wetland types, and including some individual wetland areas, across all continents. This table provides upper and lower bounds for each of the comparisons. Table 4 is based on a major literature review and designed as a concise validation table, and we discuss this table in sections 4.3.1 and 4.3.2 in the text, including observed agreements or outliers. Inherently to the design of GLWD v2, we also face the challenge that many of the most reliable input datasets for each wetland class are already included in GLWD v2, leaving remaining independent comparisons (against potentially inferior products) to be less informative.

That said, we addressed the Referees' comments to include more comparisons and validation by revising the manuscript in several major ways:

- We added a new Table A1 in Appendix A which contains the basic characteristics of each of the comparison datasets shown in Figure 1.

- We made it clearer in the manuscript that Table 4 already includes a wide variety of comparison data, from regional to global, including both remote sensing and field-based products. Furthermore, we added a new Table A2 in Appendix A in which we list all data sources shown in Table 4 individually, including a brief description of the main characteristics of each data source (e.g., field-based vs. remote sensing product).

- We added a validation of GLWD v2 against ~25,000 verified point observations of wetland presence/absence globally (new section 4.3.3, including two new tables). As the validation dataset (compiled by Zhang et al., 2023) is not publicly available, we invited 2 co-authors of that team to join our manuscript.

- Furthermore, we conducted a statistical comparison (including a confusion matrix) of GLWD v2 against GLWD v1 to show the substantial changes in the upgrade (new section 4.3.4) and a mostly visual comparison of GLWD v2 against GWL_FCS30 by Zhang et al. (2023; 2024), i.e., a multi-class remote-sensing product (new section 4.3.5, including a new figure).

- Finally, we created an additional table showing the breakdown of all GLWD v2 wetland extents by class and country. Given the size of this table, it cannot be presented in the Appendix of the manuscript, so we opted to add it as Supplementary Information, and we point to it from within the text. We hope this can facilitate that users with national knowledge about wetland distributions can judge the overall quality of GLWD v2 for their own regional or local assessments.

In total, our major revision related to validation and comparison analyses added 6 new tables (1 in manuscript, 4 in Appendix, 1 in Supplementary Information), 1 new figure, and >3 pages of new explanations and discussions, including statistical performance indicators. We hope these additions will sufficiently improve the presentation of the GLWD v2 database and allow readers to judge its validity.

Major Points:

1. I recommend emphasizing the distinct applications and improvements over other recent global wetland datasets. While the paper touches on this, a more detailed comparative analysis between GLWD v2 and existing databases (e.g., GIEMS, GLOWABO) would strengthen the argument for its uniqueness and applicability in contemporary research.

   RESPONSE: We appreciate this comment. In our major revision, we made several substantial additions to the manuscript which we describe in our **General Responses #1 (applicability) and #3 (comparisons)**. Specifically, comparisons to the GIEMS and GLOWABO datasets are already shown in Fig. 1 and in Table 4. But we would also like to point out that GIEMS and GLOWABO are rather different databases compared to GLWD v2; i.e., GIEMS, which is fully integrated in GLWD v2, only provides inundation extents rather than a wetland classification, and GLOWABO only refers to the single class of lakes (without further distinction into reservoirs or saline lakes).

2. While the authors acknowledge persistent issues in defining and classifying wetlands globally, consider proposing potential solutions or standardization efforts to improve consistency in future wetland mapping initiatives. Since the authors mention that there are very significant differences in the definitional criteria for wetlands used in different data products or studies, are the wetland classification criteria used in this dataset comparable to those used in other studies, and are the wetland products obtained comparable to other products?

   RESPONSE: These comments and suggestions are well taken as the problem of different wetland definitions is at the very core of why wetland mapping products or extent estimates are so difficult to create and compare. We feel, however, that this question is going beyond the scope of our database paper which aims to simply describe a new data product. Unifying or standardizing the definitions of wetlands is a momentous challenge that would require international and authoritative input from many organizations that are not represented by the co-authors of this manuscript. Therefore, we are hesitant to expand our manuscript towards proposing new wetland definitions. Rather, our manuscript aims to be as transparent as possible in the description of our product, and we already propose a crosswalk table to other wetland definitions (Table 5 in original manuscript; now Table 6) in order to make GLWD v2 as useful and clear as possible for future studies.

3. Clarify and potentially expand on the validation methods used to assess the accuracy of the new dataset. Although the area estimates of GLWD V2 was compared with other datasets, please consider comparing results against independent observations or field data where possible. I recommend adding a section that describes field-based or independent validation efforts for other wetland types, especially in regions with significant wetland coverage, such as Southeast Asia or the Amazon basin, to compare GLWD v2 classifications against in-situ observations or higher-resolution local datasets. This would provide empirical validation of the classification system and spatial accuracy.

   RESPONSE: We acknowledge the request for more validation, in particular field-based and in situ comparisons. We would like to refer to our **General Response #3** which outlines the additional assessment that we conducted in the major revision of our manuscript to address this concern. In particular, the comparisons shown in Table 4 already include regional and higher-resolution datasets, such as those for the Amazon Basin or various regions in Southeast Asia (Indonesia,

India), and we hope that our new Table A2 in Appendix A, which lists and briefly describes each of the comparison datasets, will increase clarity about this. In terms of validation, we add a comparison against a validation dataset of ~25,000 individual point locations of wetland presence/absence observations across the world.

4. Are there inconsistencies or conflicts between the 25 major global data products used to generate the GLWD V2 data? What measures have been taken in this work to avoid the impacts on wetland classification when there are inconsistencies between the surface types of the input data (e.g., the HydroLAKES and Global Surface Water dataset, these two estimates are highly inconsistent)? reported by Rajib et al., (2024): A call for consistency and integration in global surface water estimates)? Is it possible to be specific in the section on selection criteria for input data (coherency between datasets)?

RESPONSE: We fully agree with the call made by Rajib et al. (2024) for more consistency and integration in global surface water estimates, and we added this new reference to our manuscript. We believe that the creation of GLWD v2 and its transparent description of input sources and integration techniques follows this call. There are indeed many inconsistencies and conflicts in the original data sources used to create GLWD v2. We aimed to describe those differences succinctly in Table 1 (now with added descriptions of major limitations of the input datasets), and then we explain in the detailed Methods sections how we treated each dataset in the amalgamation process. In particular, section 3.1 provides an overview of the methodology including the main approach that we followed to avoid that inconsistencies in the input data transgress into our results, namely by a) selecting only one input data source per wetland class rather than merging many inconsistent ones; and b) creating a hierarchy of input datasets whereby the higher ranked classes receive priority over (possibly inconsistent) lower ranked datasets. The main selection criteria and ranking decisions of datasets are described in lines 258-279 (of the original manuscript), and the hierarchy is shown in Figure 3 (now updated to add clarity). Despite all attempts to reduce issues of inconsistencies and duplication, we discuss this as a main source of uncertainty of our product in lines 788-803 (original manuscript; now slightly expanded and reworded for clarity).

5. While GLWD v2 is described as a static map representing contemporary conditions, and although they provide more detailed wetland classification information than the previous version of the data, they cannot be used to quantify seasonal fluctuations and inter-annual scales in wetland ecosystems. The importance of the data is diminished by the fact that wetlands can change significantly over relatively short periods of time. The authors may need to go into more depth to explain the critical role of this wetland classification information and potential application scenarios to highlight the importance of this dataset.

RESPONSE: We fully acknowledge the shortcomings that stem from the fact that GLWD v2 is a static database, and we agree that it should not be used as a stand-alone database to quantify seasonal fluctuations and inter-annual trends. See also our **General Response #2** regarding this issue. In fact, we state and discuss this shortcoming in lines 821-830 of the original manuscript, and we further expanded on these explanations in the revision. Please also note our explanation as part of our **General Response #2** regarding seasonal fluctuations in the extent of inundation or saturation, which is not the same as a fluctuation in the extent of the wetland ecosystem itself

as the dry state is part of the ecological condition of a wetland. Finally, please also refer to our **General Response #1** about the applicability of GLWD v2.

6. Offer more detailed guidance on appropriate uses and limitations of the dataset for various applications. This could help users better understand how to effectively utilize the data in different contexts.

    RESPONSE: We would like to refer to our **General Response #1** regarding the applicability of our static but classified global wetland map.

7. Discuss integration with other datasets: Explore how GLWD v2 could be integrated or used in conjunction with other global environmental datasets (e.g., land cover, climate data) to enhance its value for interdisciplinary research.

    RESPONSE: Again, we would like to refer to our **General Response #1** regarding the applicability and integrability of our global wetland map.

Minor Points:

1. Overall, the manuscript is well-written and clear. However, there are instances where technical jargon may impede accessibility for a broader audience. For example, the use of terms like "mosaicking" and "ancillary data" may need more explanation. Consider simplifying or defining these terms more clearly for non-specialist readers.

    RESPONSE: We appreciate this comment, and it is certainly our goal to keep this manuscript accessible to as broad a reader community as possible. That said, the GLWD v2 database is a GIS product and thus descriptions of some specific GIS procedures are necessary. We aimed to keep the terminology simple where possible, using only GIS expressions that are rather common (such as the standard process of combining two raster datasets through 'mosaicking', an approach that is available in virtually all GIS software packages). Our goal of keeping the explanations accessible is exemplified by the fact that already the in the original manuscript we had replaced the GIS term 'mosaicking' with the simpler term 'inserting' which we defined in lines 261-264 (original manuscript). We also consider the expression 'ancillary data' to be a commonly used GIS term for 'supporting data' (though the more precise adjective 'ancillary' being preferred as even core data could be confused to be 'supportive data' within an analysis). In our revision, we carefully inspected the manuscript and made further adjustments to improve accessibility of the explanations as appropriate. For example, we replaced the only other occurrence of the term 'mosaicking' with 'merging'. That said, we also aimed to prioritize precise and correct technical terminology in cases where simplifications would introduce ambiguity.

2. The inclusion of several figures to demonstrate the different stages of data integration and the final wetland classification is excellent. However, Figure 3 could be expanded with more details on the data fusion procedures as the current methods section is complex and is not very clear. A table comparing GLWD v2 with other global wetland maps in terms of resolution, typology, and applications would be a valuable addition.

    RESPONSE: Thank you for this suggestion. In our revision, we modified/expanded Figure 3 to include more of the core processing steps, and we added class numbers to the figure to provide

a more direct link between the descriptions in the text and the location of each step in the figure. In addition, we created an even more elaborate figure that depicts many of the sub-steps of the methodology. Given the high amount of detail on this figure, we opted to place it as Figure B1 in Appendix B and refer to it from the manuscript. We hope these substantial modifications will increase clarity in the Methods section.

As for comparing GLWD v2 with other global wetland maps, we would like to point out that Figure 1 ("Common surface water datasets plotted according to their spatial and temporal resolution") intends to provide exactly this information (including also a temporal component). To make this more evident and to provide even more information, we added a new Table A1 in Appendix A which contains the basic characteristics of each of the 24 existing datasets shown in Figure 1, and we refer to this new table from within the manuscript.

3. Table 1 provides a good overview of the data sources but could be improved by adding information on the temporal coverage of each dataset.

RESPONSE: We appreciate this suggestion. There has already been information on the temporal coverage for some datasets listed in Table 1 (in column 'Description' of the original manuscript). In the revision, we updated Table 1 by adding the temporal information for each dataset in a separate column to provide this information more clearly and more comprehensively.

4. Please consider adding a section to describe all necessary information on the data files provided in the dataset (e.g., data format, layer names and content). This would make it easy for data users to quickly know what information are provided in each data file.

RESPONSE: We fully agree with this suggestion. In fact, all data files have already been described and documented in a dedicated Technical Documentation (in PDF format) which is distributed together with the data files and is available online (see links on the figshare repository). In addition, there is also a table (in CSV format) provided with the datasets which contains the legend information for each wetland class (i.e., a reference table showing wetland class ID and class name). We now placed the Technical Documentation more prominently on the figshare repository and we added the following sentence to section 6 (Data availability): "*The data layers are provided in different formats and are accompanied by a Technical Documentation explaining file names and specifications.*"

With kind regards,

Bernhard Lehner
on behalf of all co-authors

---

## Author Comment (AC3)

**Referee #3**

This article generates GLWD version 2 by integrating the latest ground and satellite data products into one database. GLWD v2 mapped 33 wetland categories worldwide with a minimum resolution of 10m. This database has successfully overcome the differences caused by inconsistent regional or national data sources, filling the gap between field surveys and globally applicable classifications. Beneficial for promoting large-scale hydrological, ecological, biogeochemical, and conservation applications, supporting research and protection of wetland ecosystems around the world.

RESPONSE: We thank the Referee for this positive feedback recognizing the beneficial value of the GLWD v2 database.

Although GLWD v2 has made significant progress in global wetland classification and representation, it may require careful consideration of the following issues:

(1) GLWD v2 combined different datasets to map a total of 33 wetland categories, including wetlands and water body types. However, how to eliminate the temporal differences in wetland types extracted from different datasets? That is to say, the increase or decrease of wetlands may have biases in different statistical time differences, especially in areas where human activities have a significant impact on wetland area disturbance. How to consider this?

RESPONSE: We appreciate the concern raised by the Referee. In fact, in the original manuscript, we comment on this issue in the discussion of limitations (original lines 821-825): "*Rather than being a time-resolved product, GLWD v2 depicts contemporary conditions and limited aspects of inundation periodicity (seasonal, ephemeral, etc.) as a static map. As such, it represents a long-term baseline and should not be used to directly infer or monitor trends over time in global wetland distribution. The input sources are limited to data without explicit temporality, and in many cases there may be mismatches between sources due to different temporal snapshots or time integrated summaries (e.g., flood frequencies).*" Therefore, other than by aiming to select input data sources that represent contemporary situations within a comparable time period, loosely defined as "1984-2020" (original line 169; now reiterated several times in the revised manuscript to add clarity), we have not applied any methods to extrapolate wetland extents to a particular timeline or for pre-anthropogenic conditions.

Nonetheless, in our major revision of the manuscript (and also in response to comments from the other Referees), we expanded Table 1, which describes all input data sources, by adding information on the representative time period of each input dataset. Furthermore, we carefully revisited the Discussion section and made some adjustments to improve the explanations which describe our database as a static product, including the limitations that are caused by this characteristic.

(2) There are still doubts about the clear definition of wetlands, such as whether wetlands in high dynamic change areas can be defined as wetlands? Is the statistical analysis of the high dynamic change area accurate? And this is difficult to accurately model and obtain for wetlands that exist intermittently or in the short term.

RESPONSE: We agree with the Referee that there are particular challenges in accurately defining and mapping highly dynamic wetland areas. Due to the high uncertainties in interpreting inundation frequencies (which are mostly derived from the two input datasets of GIEMS-D3 and GSW; Table 1), we chose to simplify the classification of dynamic wetland types into only 4 categories: regularly flooded,

seasonally flooded, seasonally saturated, and ephemeral (Fig. 2). To be transparent, we also summarized all frequency thresholds that we applied in Table 2.

However, we admit that in the Discussion section we were not clearly expressing our own concerns regarding the limited accuracy of these particular class distinctions related to temporality. Therefore, in our revision we expanded the discussion of uncertainties regarding inundation frequencies in section 5.3 (Limitations and uncertainties), now ending in this statement: "*Overall, we expect that the sub-class distinctions derived from the connectivity and flood frequency analyses for riverine, lacustrine, palustrine, and ephemeral categories are the most uncertain within GLWD v2, and caution should be exercised in applications that rely on their individual characteristics.*" Finally, we would like to note that in the major revision, we added several validation and comparison assessments (new sections 4.3.3 to 4.3.5) in which we discuss the reliability of the highly dynamic wetland classes and their observed uncertainties in more detail.

(3) What is the significance of distinguishing lakes, saltwater lakes, and reservoirs based on the classification criteria for 33 wetland types? Is the source of the third-party dataset used reliable? How are subcategories specifically classified?

RESPONSE: Thank you for this comment. As cited in our manuscript (original lines 175 ff.), the Ramsar Convention on Wetlands (1971) adopted a broad definition of wetlands, comprising nearly all types of aquatic ecosystems including "*areas of marsh, fen, peatland or water, whether natural or artificial, permanent or temporary, with water that is static or flowing, fresh, brackish, or salt ...*" According to this definition, lakes and reservoirs are part of wetlands. In GLWD v2, we follow this definition, and we break out freshwater lakes, saline lakes, and artificial reservoirs as their own subcategories because a) they are ecologically distinct aquatic systems; and b) because there are readily available datasets and methods which allowed us to make this differentiation globally. The classification of these three subcategories is explained in its own section 3.2.1 in the manuscript (original lines 285-300), and all data sources are referenced in Table 1.

Natural freshwater lakes are depicted from the HydroLAKES dataset, which has been evaluated to be near complete for lakes that exceed 10 ha in size (Messager et al. 2016; Lehner et al. 2022). The distinction of saline lakes from HydroLAKES has been performed using methods that are described in Ding et al. (2024); the results have additionally been verified in our manuscript (see original lines 297-300). Reservoirs are depicted from the GDW database which has just been released as the most comprehensive global database with mapped reservoir extents, mostly complete for reservoirs larger than 10 km$^2$ and containing many smaller ones (we now updated the citation to *Lehner et al., 2024*). Given the validation results in their respective publications, we believe that these three distinct wetland categories are depicted in reliable quality in their sources and are thus also reasonably represented in GLWD v2. This, however, also depends on the size of the waterbodies as reservoirs below 10 km$^2$ may be falsely classified as lakes, and lakes or reservoirs below the 10-ha threshold (fresh or saline) are only represented in the lumped class 6 of GLWD v2 as 'Other permanent waterbody'.

Given the comments from other Referees as well, we conducted a major revision of the manuscript in which the explanations of the input datasets in Table 1 were extended (including main shortcomings and time periods) and several validation and comparison assessments were made. We hope that these additions help to sufficiently clarify the quality of the lake, saline lake, and reservoir classes in GLWD v2.

(4) The article uses a hierarchical data processing approach to separately process wetlands, such as high-resolution coastal wetland types, urban and glaciated areas, peatlands, paddy rice class, peatland classes. However, for the extraction of wetland types, only the image data or dataset used to obtain it is introduced, and the specific method needs to be declared or referenced, which further verifies the effectiveness and accuracy of the results. This question also applies to the extraction of lake water bodies.

RESPONSE: If we understand this comment correctly, the Referee wishes to see more explanations regarding how each of the 25 input datasets that are listed in Table 1 (and also mentioned throughout the Methods section) have been generated. In terms of the accuracy (or inaccuracy) that is introduced from these source datasets to GLWD v2, we state in section 5.3 on limitations and uncertainties (original lines 788-790): "*As a composite mapping product, GLWD v2 inherits the uncertainties and shortcomings of its data sources. Given the large diversity of input datasets, we refrain from discussing the quality of each source and instead refer the reader to their original publications.*" In the revision, we slightly modified this explanation which now ends in "*… we refer the reader to the original publications of the source datasets (see Table 1)*" to add clarity on where to find these publications.

As a response to this Referee request, we feel it is going beyond the scope of our manuscript to provide a full review of the 25 input datasets that we used in the production of GLWD v2, in part as some of these products are themselves based on a variety of original data sources with their own characteristics and uncertainties (e.g., derived from multiple remote sensing sources). That said, we agree with the Referee that the explanations provided in Table 1 were, at least in part, too limited to be appreciated by readers who are less familiar with each data source. For that reason, we expanded the explanations of the input datasets presented in Table 1 to briefly describe the main characteristics related to the generation, individual reliability, and/or shortcomings of each source dataset, as well as the time period which they represent.

I think if the author can explain or handle the above issues well, the paper can be published after major revision.

RESPONSE: Besides our direct responses the comments made by Referee #3, we would also like point out the replies that we submitted to the other two Referees, in which we explain how we conducted a major revision of the manuscript. As we substantially expanded on several aspects related to the validation and comparison of GLWD v2, we hope that our updates will also serve to address the raised concerns on data accuracy as stated by Referee #3 above.

With kind regards,

Bernhard Lehner
on behalf of all co-authors

---

## Author Response (AR2)

**Dear Dr. Gelfan,**

We are very pleased to learn that you consider our revisions to be sufficiently addressing the concerns of the reviewers and that you therefore recommend publication of our manuscript entitled "Mapping the world's inland surface waters: an upgrade to the Global Lakes and Wetlands Database (GLWD v2)." Please find below our brief responses to the remaining issues raised in the "Notification to the authors." All our responses are in blue.

With kind regards,

Bernhard Lehner (on behalf of all co-authors)

**Dear Authors:**

I have received three reviews of your manuscript. All referees provided extensive comments and recommended reconsidering the manuscript after major revisions. Criticisms of the referees were addressed in your responses. The revised version of the manuscript addresses the comments that resulted in improvements to the manuscript in comparison with the previous version. In the current version, the manuscript addresses relevant scientific questions within the scope of ESSD, and I believe that it is not necessary to send the manuscript to referees for a further review. The only corrections that need to be made are listed in "Notification to the authors from review file validation." Thus, I recommend the manuscript for publication after these technical corrections.

Alexander Gelfan (ESSD Topical Editor)

**Notification to the authors:**

1. Please ensure that the colour schemes used in your maps and charts allow readers with colour vision deficiencies to correctly interpret your findings. Please check your figures using the Coblis – Color Blindness Simulator (https://www.color-blindness.com/coblis-color-blindness-simulator/) and revise the colour schemes accordingly.

Response: We checked all figures regarding the guidelines to accommodate colour vision deficiencies. We believe that our color schemes are adequate for all readers to interpret our findings correctly.

2. Works "in review" can be cited upon submission with entry in the reference list, as long as these works are available to the reviewers (either on an external server or as review asset). Such works should not be cited in the final, accepted manuscript, unless published, accepted for publication, or available as preprint with a DOI.

Response: We verified that all our cited works comply with this request. There is only one citation "in review" but it is available as a preprint with a permanent DOI.

3. Before the next revision, please adjust the copyright statement in Figure 7 as follows "© Google Maps".

Response: We added the copyright statement to Figure 7, as requested.

4. Checking your paper, I noticed that your Table A3 contains grey shadings. Please note that this will not be possible in the final revised version of the paper due to HTML conversion of the paper. When revising the final version, you can use footnotes or italic/bold font.

Response: We removed all shadings from all tables (and added bold fonts in Table A3).